# How Far Can Unsupervised RLVR Scale LLM Training?

**Bingxiang He**[*1], **Yuxin Zuo**[*†1,2], **Zeyuan Liu**[*1], **Shangziqi Zhao**[*3], **Zixuan Fu**[1], **Junlin Yang**[1], **Cheng Qian**[4], **Kaiyan Zhang**[1,5], **Yuchen Fan**[6], **Ganqu Cui**[2], **Xiusi Chen**[4], **Youbang Sun**[1], **Xingtai Lv**[1], **Xuekai Zhu**[6], **Li Sheng**[1], **Ran Li**[1], **Huan-ang Gao**[1], **Yuchen Zhang**[7], **Lifan Yuan**[4], **Bowen Zhou**[‡1,2], **Zhiyuan Liu**[‡1], **Ning Ding**[‡1,2]

[1]Tsinghua University    [2]Shanghai AI Lab    [3]Xi'an Jiaotong University    [4]UIUC
[5]Frontis.AI    [6]Shanghai Jiao Tong University    [7]Peking University
`hebx24@mails.tsinghua.edu.cn`, `dingning@mail.tsinghua.edu.cn`

## Abstract

Unsupervised reinforcement learning with verifiable rewards (URLVR) offers a pathway to scale LLM training beyond the supervision bottleneck by deriving rewards without ground truth labels. Recent works leverage model intrinsic signals, showing promising early gains, yet their potential and limitations remain unclear. In this work, we revisit URLVR and provide a comprehensive analysis spanning taxonomy, theory and extensive experiments. We first classify URLVR methods into intrinsic versus external based on reward sources, then establish a unified theoretical framework revealing that all intrinsic methods converge toward sharpening the model's initial distribution This sharpening mechanism succeeds when initial confidence aligns with correctness but fails catastrophically when misaligned. Through systematic experiments, we show intrinsic rewards consistently follow a rise-then-fall pattern across methods, with collapse timing determined by model prior rather than engineering choices. Despite these scaling limits, we find intrinsic rewards remain valuable in test-time training on small datasets, and propose Model Collapse Step to measure model prior, serving as a practical indicator for RL trainability. Finally, we explore external reward methods that ground verification in computational asymmetries, showing preliminary evidence they may escape the confidence-correctness ceiling. Our findings chart boundaries for intrinsic URLVR while motivating paths toward scalable alternatives. Code is available at https://github.com/PRIME-RL/TTRL.

## 1 Introduction

Reinforcement learning with verifiable rewards (RLVR) has been central to recent breakthroughs in enhancing reasoning capability in large language models (LLMs). In RLVR, models learn from rewards that can be verified against ground truth, such as correctness in mathematics or successful code execution. Recent leading models including DeepSeek-R1 (Guo et al., 2025), Gemini 2.5 (Comanici et al., 2025) and the Qwen3 series (Yang et al., 2025; Team, 2025) have achieved remarkable performance on mathematics, coding and science benchmarks by scaling supervised RLVR. However, on the path toward superintelligence, this approach faces a crucial limitation: scaling supervision requires prohibitively high human costs, and as models reach or surpass human expertise in specialized domains, obtaining reliable ground truth supervision becomes increasingly infeasible (Burns et al., 2023; Silver & Sutton, 2025).

This supervision bottleneck has spurred growing interest in Unsupervised RLVR (URLVR) (Zuo et al., 2025; Zhao et al., 2025a), which derives rewards without ground truth labels. Just as pretraining scaling laws (Brown et al., 2020; Raffel et al., 2020) transformed large-scale computation into

---

[*]Equal Contribution. Orders are determined randomly.
[†]Project Lead.
[‡]Corresponding Authors.

intelligence on vast amounts of unlabeled data, URLVR promises to extend this paradigm to post-training, scaling reinforcement learning beyond reliance on human-provided labels.

Current URLVR methods have primarily relied on leveraging the model's intrinsic signals as rewards. Many works have proposed intrinsic reward methods, from majority voting across multiple rollouts (Zuo et al., 2025) to entropy-based metrics (Agarwal et al., 2025) as rewards, reporting encouraging early training gains. Yet these gains come with growing concerns, as several studies highlight critical failure modes including reward hacking and model collapse (Shafayat et al., 2025; Agarwal et al., 2025; Zhang et al., 2025c). Moreover, diverse methodologies have been applied across different model families and evaluation settings without systematic comparison or consensus on what constitutes reliable unsupervised rewards. This raises a fundamental question for the field: **Can intrinsic rewards truly scale LLM training?**

To answer this, we conduct a comprehensive study of URLVR, spanning taxonomy, theory and extensive experiments. We begin by classifying URLVR methods into intrinsic and external based on the source of rewards (Section 2). We then theoretically analyze the underlying mechanism of intrinsic URLVR, revealing that despite diverse design choices, all intrinsic methods converge toward sharpening the model's initial distribution, amplifying existing preferences rather than discovering new knowledge (Section 3). This sharpening mechanism has both strengths and limitations, depending on the model prior: whether the model's confidence aligns with correctness. To validate this, we implement intrinsic methods and show that intrinsic URLVR consistently follows a rise-then-fall pattern across all methods, differing only in when rather than whether collapse occurs (Section 4).

Despite these scaling limitations, intrinsic rewards remain valuable on small and domain-specific datasets, avoiding collapse even when all initial preferences are wrong, making it well-suited for test-time training (Section 5). Furthermore, we can leverage the rise-then-fall pattern to measure model prior, proposing Model Collapse Step to serve as a practical model prior indicator, predicting RL trainability without expensive training runs (Section 6). Finally, since intrinsic rewards face fundamental scalability limits rooted in sharpening mechanism, we investigate the scalability of external URLVR methods, taking self-verification as an example, which generate verifiable rewards through generation-verification asymmetries rather than internal model states, showing that it exhibits sustained improvement without the collapse patterns inherent to intrinsic methods (Section 7).

Overall, our analysis charts a clear path **from understanding why current intrinsic URLVR fails to identifying how future methods can scale**. Our findings reveal that intrinsic rewards work within well-defined boundaries determined by the model prior, enabling efficient and safe gains in test-time training while risking reward hacking when confidence misaligns with correctness. These limitations motivate exploration of external reward methods, from generation-verification asymmetries in structured domains to self-supervised signals from vast unlabeled corpora, which offer pathways toward more robust and scalable improvement.

## 2 RELATED WORK

**Reinforcement Learning with Verifiable Rewards.** Recent advances in language model reasoning leverage reinforcement learning with verifiable rewards (RLVR) (Lambert et al., 2024), where models receive binary rewards based on answer correctness verified against ground truth. Leading systems including OpenAI's o1 and o3 (Jaech et al., 2024; OpenAI, 2025), DeepSeek-R1 (Guo et al., 2025), Gemini 2.0 (Comanici et al., 2025) and Qwen3 (Yang et al., 2025; Team, 2025) have achieved remarkable performance through scaling supervised RLVR. However, this approach faces a fundamental bottleneck: as models approach human expertise in specialized domains, obtaining reliable ground-truth supervision becomes prohibitively expensive (Burns et al., 2023).

**URLVR with Intrinsic Rewards.** To address the supervision bottleneck, URLVR extends RL scalability beyond labeled data by using self-generated proxy rewards. We distinguish two intrinsic paradigms: (1) *Certainty-based methods* derive rewards from a single policy's confidence, employing metrics such as self-certainty (Zhao et al., 2025b), token-level entropy (Agarwal et al., 2025; Prabhudesai et al., 2025), trajectory entropy (Agarwal et al., 2025), raw probabilities (Li et al., 2025a) or attention patterns (Kiruluta et al., 2025b;a). (2) *Ensemble-based methods* leverage agreement across multiple rollouts, typically through majority voting (Zuo et al., 2025) or semantic clustering (Zhang et al., 2025b). Subsequent works have refined this via reward-hacking analysis (Shafayat et al., 2025),

tree search (Liu et al., 2025), paraphrasing (Zhang et al., 2025d), multi-model collectives (Yuan et al., 2025) and intermediate reasoning consistency (Zhang et al., 2025a). Crucially, these methods "sharpen" existing preferences rather than discovering new knowledge.

**URLVR with External Rewards.** An alternative paradigm within URLVR escapes the intrinsic knowledge ceiling by deriving rewards from external mechanisms. One approach leverages *unlabeled data*, where the corpus provides latent ground truth for next-token prediction (Dong et al., 2025; Li et al., 2025b), reasoning-step prediction (Wang et al., 2025; Hatamizadeh et al., 2025), dual reconstruction objectives (She et al., 2025) or autonomous QA generation (Zweiger et al., 2025; Akter et al., 2025). A second approach exploits *generation-verification asymmetries*, where verifying a solution is computationally cheaper than generating it. This principle scales via deterministic verifiers in mathematics (Simonds & Yoshiyama, 2025; Simonds et al., 2025), code execution (Zhao et al., 2025a) and formal theorem proving (Shao et al., 2025b; Hubert et al., 2025). Unlike intrinsic signals, these external rewards are grounded in objective computation or vast corpora, ensuring reward quality does not degrade as model capability scales.

## 3   THE SHARPENING MECHANISM OF INTRINSIC REWARDS

We start from intrinsic reward methods in URLVR, investigating their potential in scaling LLM training. Existing studies (Zhang et al., 2025b; Zuo et al., 2025; Agarwal et al., 2025; Shafayat et al., 2025) have empirically assessed the strengths and weaknesses of these intrinsic methods, but the underlying mechanisms of how and why they lead to the observed effects, have been underexplored.

**One-Step Update Dynamics.** Training with intrinsic rewards involves moving toward an optimal policy $\pi_\theta^*$ defined by the KL-regularized objective: $\max_{\pi_\theta} \mathbb{E}_{y \sim \pi_\theta}[r(x,y)] - \beta D_{\mathrm{KL}}(\pi_\theta \| \pi_{\mathrm{ref}})$. For a binary majority voting reward $r(x,y) = \mathbf{1}[\mathrm{ans}(y) = \mathrm{maj}(Y)]$, the optimal probability mass on the majority answer $p_{\mathrm{maj}}^*$ is given by:

$$p_{\mathrm{maj}}^{*,(k+1)} = \frac{p_{\mathrm{maj}}^{(k)} \cdot e^{1/\beta}}{p_{\mathrm{maj}}^{(k)} \cdot e^{1/\beta} + (1 - p_{\mathrm{maj}}^{(k)})}. \tag{1}$$

In practice, a single gradient update does not reach the optimum but moves partway toward it. The actual probability mass after one update $p_{\mathrm{maj}}^{(k+1)}$, satisfies the ordering $p_{\mathrm{maj}}^{*,(k+1)} \geq p_{\mathrm{maj}}^{(k+1)} \geq p_{\mathrm{maj}}^{(k)}$. This holds because policy gradients increase mass on rewarded trajectories (lower bound) but are constrained by the maximum expected reward of the optimal policy (upper bound). See Appendix A.1 for detailed derivations and empirical validation.

**Convergence to Sharpening.** The one-step update creates a "rich-get-richer" dynamic that trajectories leading to the majority have their probabilities consistently increased. Iterating this process, the policy converges toward a deterministic policy concentrated on the initial majority answer:

**Theorem 1.** *(Geometric Convergence). Consider the training process where at each iteration $k$, we: (1) sample $N$ rollouts $Y_k$ from $\pi_\theta^{(k)}$, (2) compute majority $\mathrm{maj}_k(Y_k)$, (3) perform one-step update with reward $r_k(x,y) = \mathbf{1}[\mathrm{ans}(y) = \mathrm{maj}_k(Y_k)]$ to obtain $\pi_\theta^{(k+1)}$. Let $p_{\mathrm{maj}}^{(k)} = \sum_{y:\mathrm{ans}(y)=\mathrm{maj}_k(Y_k)} \pi_\theta^{(k)}(y|x)$ denote the probability mass on majority trajectories at iteration $k$.*

*Suppose the following assumptions hold, validated empirically in Appendix A.2:*

   (A1) *Majority stability:* $\mathrm{maj}_k(Y_k) = \mathrm{maj}_0(Y_0)$ *for all $k$ (holds for sufficiently large $N$);*

   (A2) *Effective learning:* $p_{\mathrm{maj}}^{(k+1)} > p_{\mathrm{maj}}^{(k)}$ *for all $k$ (a standard assumption in policy gradient methods).*

*Then $p_{\mathrm{maj}}^{(k)}$ converges geometrically to 1 with rate $\rho = e^{-1/\beta}$, and the policy converges to*

$$\lim_{k \to \infty} \pi_\theta^{(k)}(y \mid x) = \begin{cases} \dfrac{\pi_{\mathrm{ref}}(y \mid x)}{\sum_{y':\mathrm{ans}(y')=\mathrm{maj}_0(Y_0)} \pi_{\mathrm{ref}}(y' \mid x)}, & \text{if } \mathrm{ans}(y) = \mathrm{maj}_0(Y_0), \\ 0, & \text{otherwise}. \end{cases} \tag{2}$$

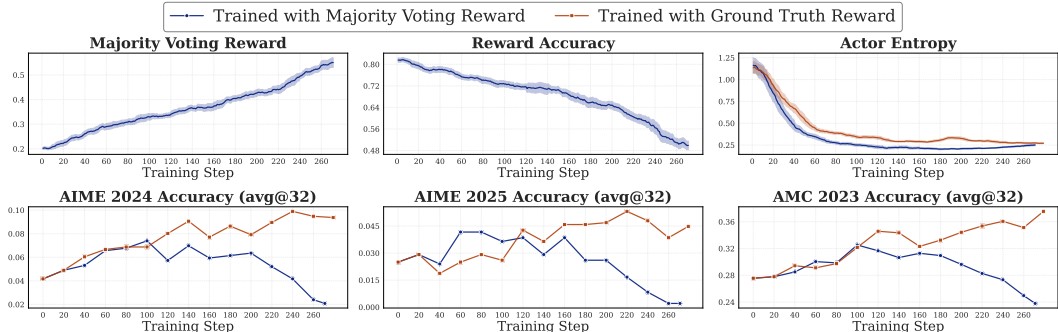

Figure 1: Training dynamics comparing majority-voting training and ground-truth training. Intrinsic rewards initially match supervised performance but eventually collapse: the proxy reward rises while validation accuracy falls, revealing divergence between optimizing confidence and optimizing correctness.

Complete proof is provided in Appendix A.3. For other URLVR intrinsic rewards, we leverage the sharpening intuition and propose a unified reward framework in Appendix A.4, showing that all intrinsic rewards can be understood through a single lens: manipulating cross-entropy between carefully chosen distributions. Built upon this unified perspective, we generalize this convergence analysis in Appendix A.5, and provide optimal policies induced respectively in Appendix A.6.

This convergence behavior has profound implications depending on the model prior. When the model's confidence (reflected in $\text{maj}_0$) aligns with correctness, convergence reinforces good solutions. Conversely, if confidence is poorly aligned, the same mechanism amplifies errors, leading to model collapse. Next, we will evidence this from an empirical perspective.

## 4   WHEN DOES INTRINSIC URLVR WORK?

> **Takeaways**
>
> Intrinsic URLVR universally follows a rise-then-fall pattern across all methods. Early gains reflect confidence-correctness alignment in the model's prior, while eventual collapse is inevitable when this alignment breaks down.

As derived in Section 3, intrinsic rewards sharpen distributions by amplifying the model's initial preferences. Here, we empirically investigate the practical boundaries of this mechanism. We first trace the lifecycle of intrinsic URLVR (Section 4.1), finding early gains followed by collapse across all methods. We then conduct a fine-grained per-problem analysis (Section 4.2), revealing that training primarily amplifies existing preferences rather than correcting them. These results confirm that success depends fundamentally on whether initial confidence aligns with correctness.

### 4.1   THE RISE AND FALL OF INTRINSIC URLVR

In this section, we conduct experiments across different hyperparameters and methods for intrinsic rewards, revealing that intrinsic URLVR follows a consistent rise-then-fall pattern.

**Setup.** In main text, we take majority voting reward from TTRL (Zuo et al., 2025) as an example to compare intrinsic reward training against standard RLVR using ground truth labels. We train Qwen3-1.7B-Base on DAPO-17k (Yu et al., 2025) using default hyperparameters from Table 6, and evaluate on AIME 2024 (Li et al., 2024), AIME 2025 (Balunović et al., 2025) and AMC 2023 (Li et al., 2024). Following standard practice, we generate 32 solutions per problem at temperature 0.6 with top-p 0.95, reporting average accuracy (avg@32). Beyond validation performance, we track three training dynamics: *Majority Voting Reward* (the intrinsic reward signal), *Reward Accuracy* (whether pseudo-reward matches ground truth reward), and *Actor Entropy*. See Appendix B.1 for complete details. Unless stated otherwise, we utilize the default setup above for later experiments.

**Results.** Intrinsic rewards initially match ground-truth training but eventually collapse, and this pattern exists across all hyperparameter settings we tested. Figure 1 shows that majority-voting training matches or even exceeds ground-truth training on three benchmarks during the early phase. But continued training reveals that while the proxy *Majority Voting Reward* keeps rising, both *Reward Accuracy* and validation performance decline, exhibiting obvious reward hacking. We also observe that training with majority voting reward drives down *Actor Entropy* faster than ground-truth training, establishing a connection between reduced uncertainty and improved performance.

We further thoroughly tune four key hyperparameters across five intrinsic reward methods, including training temperature, mini-batch size, KL regularization and rollout number (Appendix B.3). We find that some choices matter significantly (e.g. mini-batch size and rollout number), but nearly all settings eventually degrade, differing only in when rather than whether collapse occurs. Even with the most stable settings, collapse occurs roughly 1,000 steps, suggesting that this rise-then-fall pattern may reflect a fundamental limitation rather than an engineering problem.

While this pattern is universal, the specific pathologies of collapse differ by reward formulation. We find that *certainty-based* methods like raw probability tend to collapse toward brevity, whereas *entropy-based* methods often pad sequences with repetitive tokens to drive down entropy. We provide a detailed comparison of these failure modes and their associated scaling limits in Appendix B.4.

## 4.2 FINE-GRAINED PER-PROBLEM ANALYSIS

Next, we examine individual training samples in detail. Our analysis reveals that training amplifies initial preferences rather than correcting errors within the same problem. Surprisingly, this amplification can still generalize to unseen problems and improve performance.

**Setup.** We train Qwen3-1.7B-Base on 25 randomly sampled individual problems from MATH500 using REINFORCE with Trajectory-Level Entropy as the intrinsic reward. Each problem trains for 100 epochs with batch size 1 and 8 rollouts for baseline estimation.

**Results.** Figure 2 shows 8 representative cases and the others appear in Figure 8. We track greedy decoding correctness (heatmap) and whether the highest-reward sample is correct (green binary square-wave indicator). We observe four distinct patterns:

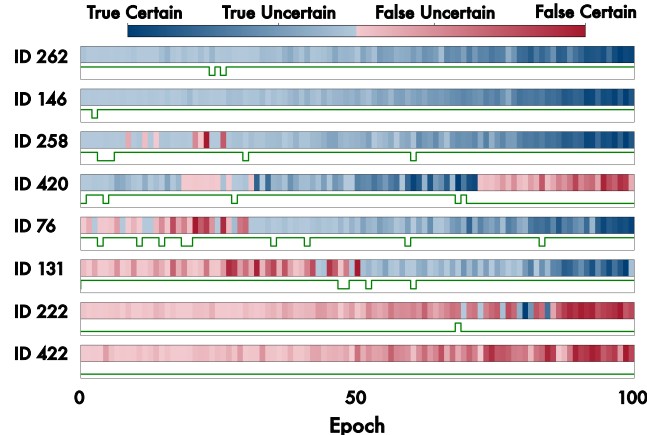

• **Amplifying success** (ID 262, 146, 258): These problems start with correct greedy decoding (blue at epoch 0), and the highest-reward sample remains correct throughout training (green wave stays at 1). Training simply increases the model's confidence around this already-correct solution, shown by the deepening blue color.

Figure 2: Training dynamics on individual representative problems. For each problem, the heatmap shows greedy decoding correctness across epochs (blue = correct, red = wrong; darker = higher confidence), and the green wave indicates whether the highest-reward rollout is correct.

• **Amplifying failure** (ID 222, 422): The highest-reward sample is almost wrong throughout (green wave mainly stays at 0). Training amplifies the model's confidence in these incorrect answers, shown by the deepening red color.

• **Wrong → Correct** (ID 76, 131): Greedy decoding initially produces wrong answers (red), but the highest-reward sample is usually correct during training (green wave mostly at 1). This guides the model from wrong to correct, as shown by the transition (red → blue).

• **Correct → Wrong** (ID 420): The model starts correct (blue), but the highest-reward sample fluctuates between correct and wrong (green wave alternates between 1 and 0). This inconsistency causes the model to gradually degrade from correct to wrong (blue → red).

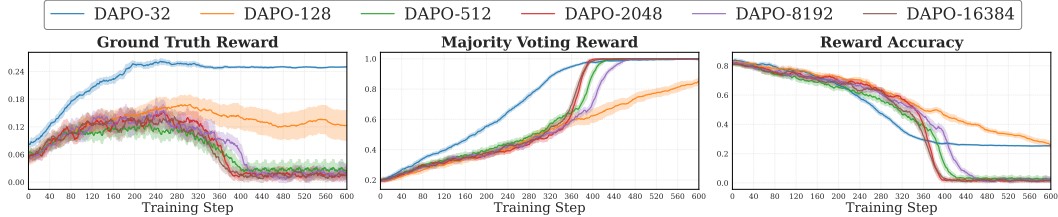

Figure 3: Effect of training dataset size from $\{32, 128, 512, 2048, 8192, 16384\}$. Training with 32 or 128 samples maintains stable performance without model collapse.

Among the 25 problems, training flipped greedy correctness in just 3 cases (12%). The remaining 22 simply sharpened the model's initial preference regardless of whether correct or wrong. Even for the 3 that flipped, the color still deepens over training, showing that sharpening happens regardless of whether correctness changes. This reveals training as amplification rather than correction.

The key factor appears to be whether the highest-reward sample is mostly correct. Surprisingly, even when intrinsic rewards amplify failures on a training problem, the resulting sharpening can still generalize and improve performance on unseen out-of-distribution (OOD) problems. This suggests that the "knowledge" gained is not necessarily problem-specific but reflects a broader alignment between the model's confidence and correctness that can be exploited in certain regimes. Detailed results and analysis of this OOD cross-problem generalization are provided in Appendix B.5.

## 5 How Can Sharpening from Intrinsic URLVR Be Applied Safely?

> **Takeaways**
>
> Small datasets induce localized rather than systematic policy shift, even training on wrong problems can yield gains, making test-time training a safe and practical application.

The previous section establishes that intrinsic URLVR drive convergence to deterministic policies and may result in reward hacking, limiting its application in scaling LLM training. But do intrinsic URLVR always lead to model collapse, or can they be safely applied under specific conditions? In this section, we demonstrate that model collapse can be prevented when training data is sufficiently small and domain-specific, making intrinsic URLVR particularly suitable for test-time training scenarios.

### 5.1 Small Datasets Prevent Model Collapse

In Section 4.2, we have seen how training on one problem works. Now we want to see whether the training dataset size influences the intrinsic URLVR performance.

**Setup.** We train Qwen3-1.7B-Base using majority voting reward on randomly sampled training subsets from DAPO-17k with $\{32, 128, 512, 2048, 8192, 16384\}$ samples. To ensure fair comparison, we fix global batch size to 32 and adjust epochs so all settings complete exactly 600 optimization steps. We monitor *Ground Truth Reward*, *Majority Voting Reward*, and *Reward Accuracy* to detect reward hacking. Results are verified across 3 random seeds for subset sizes $\{32, 128, 512\}$.

**Results.** Figure 3 reveals that training with $\leq 128$ samples maintains stable performance without collapse, while larger datasets ($\geq 512$) consistently exhibit reward hacking. Notably, DAPO-32 achieves rapid consensus (*Majority Voting Reward* $\rightarrow 1$) while preserving high *Ground Truth Reward*, demonstrating that the model converges on these problems without collapsing. This distinction holds across all three seeds, where DAPO-32 never collapses, while DAPO-512 always does.

**Analysis.** We hypothesize that small datasets induce localized overfitting rather than systematic policy shift. With only 32 problems, even perfect overfitting learns isolated facts rather than generalizable patterns. To test this, we measure the KL divergence from the reference model at each training step:

$$D_{\mathrm{KL}}^{(t)}(\pi_\theta^{(t)} \| \pi_{\mathrm{ref}}) = \mathbb{E}_{x \sim \mathcal{D}_{\mathrm{train}}} \left[ \mathbb{E}_{y \sim \pi_\theta^{(t)}(\cdot|x)} \left[ \log \frac{\pi_\theta^{(t)}(y|x)}{\pi_{\mathrm{ref}}(y|x)} \right] \right] \tag{3}$$

As shown in Figure 4, smaller subsets induce far smaller distributional shifts. DAPO-32 reaches only 0.057 KL after 600 steps, while DAPO-512 reaches 2× higher. This aligns with findings that catastrophic forgetting correlates with distribution shift between fine-tuned and base policies (Shenfeld et al., 2025). The limited drift suggests the model sharpens confidence on specific samples through localized parameter updates (Carlsson et al., 2024) without altering its global policy, thus preserving general reasoning on AIME24/AMC23. In contrast, large datasets require dense parameter updates that cause global policy shift, causing model collapse.

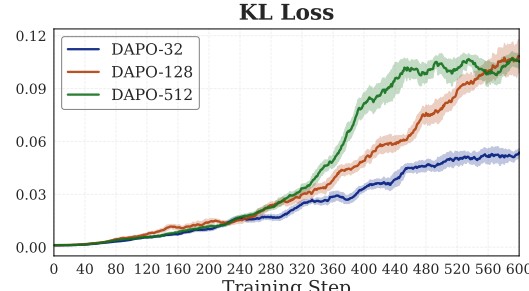

Figure 4: KL divergence at each training step.

Remarkably, this localized effect is so robust that it holds even in extreme cases where the initial majority votes are entirely incorrect. We find that training on 32 samples with near-zero label accuracy still produces gains on OOD benchmarks like AIME24. We provide the detailed dynamics of this "incorrect-majority" experiment in Appendix C.1.

## 5.2 TEST-TIME TRAINING AS A SAFE APPLICATION

Building on the small dataset insights, we examine test-time training where models are adapted directly on the target evaluation domains without ground truth labels, which may fit the small dataset size constraint. We investigate whether it can bring improvement while not leading to collapse.

**Setup.** We train Qwen3-1.7B-Base using majority voting reward on two settings: AMC23 (40 problems, test-time) and DAPO-17k (~17,000 problems, train-time). Both use batch size 40. We track *Ground Truth Reward*, *Majority Voting Reward*, and performance on both AMC23 and AIME24.

**Results.** Figure 5 shows that test-time training on AMC23 avoids collapse. Both *Ground Truth Reward* and *Majority Voting Reward* rise and stabilize, with performance improving on both AMC23 and AIME24. In contrast, training on DAPO-17k shows the familiar rise-then-fall pattern. This indicates that intrinsic URLVR may be safely applied in test-time training. And this explains why many recent works using intrinsic rewards focus on test-time rather than train-time settings (Zuo et al., 2025; Prabhudesai et al., 2025).

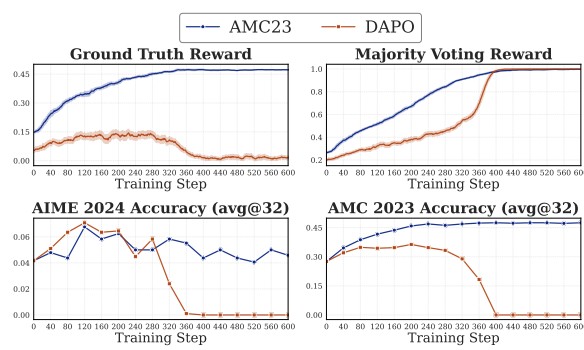

Figure 5: Comparison between training and test-time.

## 6 HOW CAN WE MEASURE MODEL PRIOR?

> **Takeaways**
>
> We propose the *Model Collapse Step* as a novel indicator of model priors, which measures standard RL trainability by tracking reward accuracy collapses during intrinsic URLVR. This indicator achieves accuracy in assessing trainability on par with running standard RL itself, but with higher efficiency; it outperforms pass@k, requires no ground-truth labels and remains robust to multiple-choice problems.

Previous sections suggest that intrinsic URLVR is effective only when the model's initial confidence is aligned with correctness. This raises a practical question: **Can the strength of this alignment**

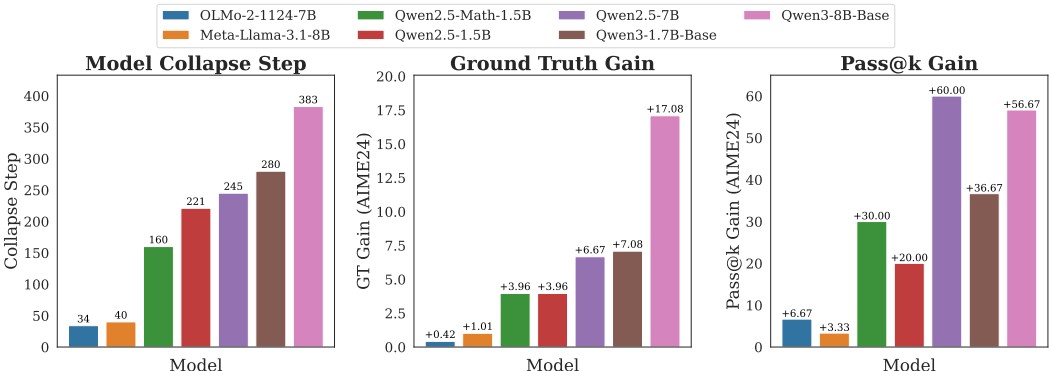

Figure 6: Comparison between Model Collapse Step (left) and **Pass**@k Gain (right) as predictors of RL trainability, with GT Gain (middle) as reference. Later collapse strongly predicts better RL performance.

**be used to estimate the model prior rather than running expensive RL training to select the base model?** The community typically selects by running full RL training on multiple checkpoints and considers such models to have a stronger prior. However, this method is costly and inefficient. Another standard approach is $pass@k$ (Wu et al., 2025a), measuring by sampling $k$ solutions without training. This works but also has limitations. First, this method has a significantly lower accuracy. And it also struggles with multiple-choice questions, where $pass@k \rightarrow 1$ when $k$ is sufficiently large.

In this section, we leverage the rise-then-fall pattern to propose **Model Collapse Step**, defined as the training step where *Reward Accuracy* drops below 1%, measuring how long a model can sustain intrinsic URLVR before it starts to collapse. We find that different base models lead to different outcomes (see Appendix C.2 for a pilot study on Qwen and LLaMA families). Models with stronger priors remain stable for longer before collapsing, indicating higher suitability as base models for RL.

## 6.1 MODEL COLLAPSE STEP ACCURATELY PREDICTS RL GAINS

To validate Model Collapse Step as an accurate predictor of RL trainability, we compare it against the widely-used $pass@k$ metric and standard RLVR training.

**Setup.** We evaluate 7 models from 3 families (OLMo, LLaMA, Qwen) on AIME24, using three indicators: (1) **Ground Truth (GT) Gain**, measuring performance improvement from standard supervised RLVR on 1 epoch DAPO-17k training with ground truth rewards; (2) **Pass**@k **Gain**, calculated as the difference between $pass@256$ and $pass@1$ on AIME24; and (3) **Model Collapse Step**, measured as the training step when *Reward Accuracy* falls below 1% during intrinsic URLVR training with majority voting reward with default hyperparameters. We evaluate the correlation between these indicators and use them to measure the model prior.

**Results.** As shown in Figure 6, Model Collapse Step correlates strongly with GT Gain. Models that survive longer with a larger Model Collapse Step consistently yield better results in standard supervised RL. This metric matches and even surpasses $pass@k$'s predictive power and it cannot be gamed by random guessing on multiple-choice questions, demonstrating its reliability as an indicator.

## 6.2 MODEL COLLAPSE STEP RAPIDLY PREDICTS RL GAINS

While Model Collapse Step provides an accurate prediction, can it be computed efficiently? Based on insights from the hyperparameter tuning (Appendix B.3), we hypothesize that certain hyperparameters, such as mini-batch size and rollout number, can accelerate convergence without sacrificing the accuracy of the model prior indicator for RL trainability. This is valuable for practitioners evaluating multiple model candidates before committing to expensive RL training.

**Setup.** We test whether it remains consistent across different training hyperparameters by varying mini-batch size $\in \{1, 8, 64\}$ and rollout count $\in \{8, 16, 32\}$. For each setting, we measure for same 7 models and assess whether we can preserve predictive power while reducing computation time.

**Results.** Figure 33 shows that while aggressive hyperparameters ($MBS = 1$ or $N = 32$) accelerate collapse in absolute steps, the ranking of models remains relatively stable. This enables more rapid assessment for models with strong prior ($\geq 50$ steps less than before), effectively shortening the time required to measure model priors, making Model Collapse Step not only accurate but also rapid.

Notably, using optimized hyperparameters ($MBS = 1, N = 8$), the Model Collapse Step requires only 1.19B inference tokens, **5.6× fewer** than a standard RL training run (6.66B tokens). Detailed computation costs are provided in Table 7. While $pass@k$ offers an alternative static metric, our experiments show it correlates less reliably with actual RL gains (Figure 6). By measuring training dynamics rather than pre-training performance, Model Collapse Step captures the interaction between model priors and the learning process itself. Moreover, it requires no ground truth labels, making it applicable even when verification is unavailable. For practitioners evaluating multiple model candidates for RL, this provides a more reliable filter before committing to expensive RL training.

The predictive power of Model Collapse Step is grounded in Theorem 1. It captures the point where geometric amplification shifts from beneficial (reinforcing correct) to harmful (amplifying errors). In practice, this enables efficient pre-RLVR assessment of trainability via short diagnostic runs with aggressive hyperparameters and calibrated early stopping, capturing gains while avoiding collapse.

## 7 DISCUSSION

> **Takeaways**
>
> Intrinsic rewards are fundamentally bounded by what the model already knows. External rewards grounded in unlabeled data or generation-verification asymmetry provide signals that scale with data and computation rather than saturating with model capacity, offering a more promising path towards scalable URLVR.

Preceding sections reveal that intrinsic URLVR faces fundamental scalability limits rooted in confidence-correctness alignment. When this alignment is weak, intrinsic methods amplify existing biases rather than discovering new knowledge. This limitation is not merely a hyperparameter issue but stems from the nature of the reward signal itself, which derives entirely from the model's internal state and therefore cannot consistently push the model beyond what it already knows.

This motivates exploring alternative URLVR methods that can escape this ceiling. Recent works have explored external reward methods that generate verifiable signals through mechanisms independent of the model's internal state. As surveyed in Section 2, we identify two promising directions that leverage unlabeled data structures to derive rewards from the corpus and exploit generation-verification asymmetries. Both paradigms offer rewards that scale with data or computation rather than saturating with model capacity, providing complementary paths toward scalable self-evolution.

To understand how external rewards differ in practice, we examine self-verification as a concrete case study, which exploits a key asymmetry in data that generation is hard but verification is easy. This asymmetry allows models to provide accurate rewards by themselves without ground truth labels, which is also validated in DeepSeekMath-V2 (Shao et al., 2025b) [1].

Specifically, we train Qwen3-1.7B-Base and 4B-Base on a subset of the Countdown task[2], where models must form arithmetic expressions that reach a target value. For Countdown, generating correct expressions is challenging, but checking if an expression equals the target is trivial. We sample 4k problems for training and 1k for validation. For self-verification, the model generates solutions and evaluates them using a verification prompt that outputs binary correctness (see Prompt 2 in Appendix C.3). For reward score computation, we use the ground truth scoring function[3] to determine whether expressions correctly evaluate to the target and compare: (1) **Self-Verification** (our method), (2) **Trajectory-Level Entropy** (reward from Table 1), and (3) **Oracle Supervision** (training with ground truth reward). We track validation accuracy, *Ground Truth Reward* and *Reward Accuracy*.

---

[1]Same method name but different implementation.
[2]https://huggingface.co/datasets/Jiayi-Pan/Countdown-Tasks-3to4
[3]https://github.com/Jiayi-Pan/TinyZero/blob/main/verl/utils/reward_score/countdown.py

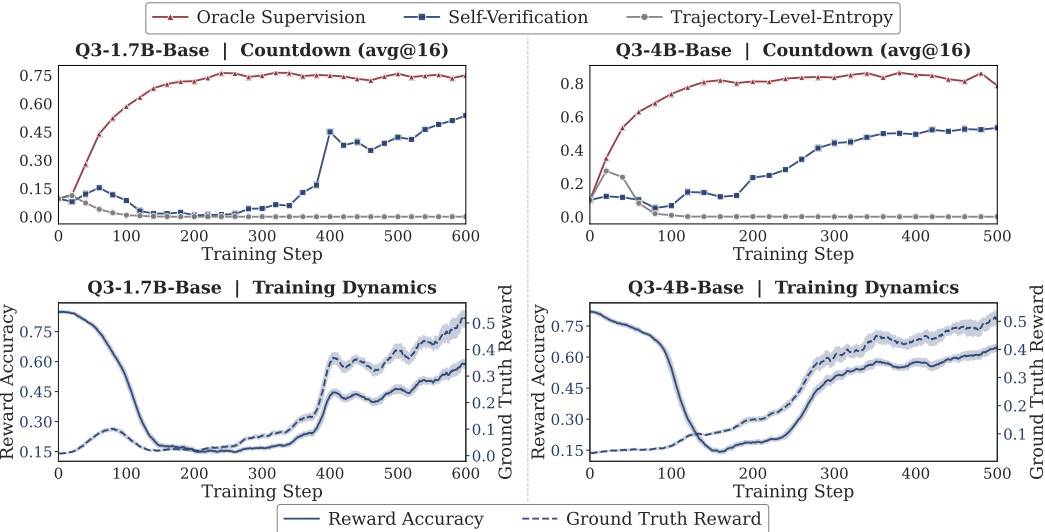

Figure 7: Top: Validation accuracy comparing Self-Verification with Trajectory-Level Entropy and Oracle Supervision. Bottom: Training dynamics for Self-Verification.

Our experiments show that Self-Verification works much better than Trajectory-Level Entropy (Figure 7, top). Both models reach higher validation accuracy with Self-Verification. The bottom figure shows an interesting pattern. *Reward Accuracy* initially drops around step 200 as the policy tries to exploit the verifier, then recovers and stabilizes above 0.5. Meanwhile, *Ground Truth Reward* keeps rising. This recovery shows the model is genuinely learning to solve problems, validating that it supplies stronger signals than previous intrinsic rewards while resisting reward hacking, supporting generation-verification asymmetry as a promising direction beyond intrinsic methods.

We find that a model's ability to follow verification instructions is a key determinant of success. Instruction-aligned models demonstrate higher starting performance and significantly greater robustness to different verification prompts compared to base models. We provide a comprehensive analysis of prompt sensitivity and the scaling benefits of instruction alignment in Appendix C.3.

These observations point to why external rewards offer a more promising path for scaling URLVR. **First, external rewards are grounded in procedures that do not degrade as the model improves.** A verifier that checks arithmetic expressions, executes code, or validates proofs against formal specifications remains equally reliable regardless of how sophisticated the model's outputs become. This contrasts with intrinsic rewards, which shift as the model's distribution changes and can amplify arbitrary patterns. **Second, external rewards can be generated at scale without human annotation.** Verification procedures are often cheap to run, and the unlabeled-data paradigm shows that even tasks without native verification asymmetries can be converted into reward-based learning by leveraging structure in existing text. The quantity of available data and the affordability of verification computation are both scalable resources, unbounded by human labeling capacity. Where intrinsic rewards are bounded by what the model already knows, external rewards can provide a fresh signal drawn from data and computation.

## 8 CONCLUSION

This work explores how Unsupervised RLVR scales LLMs via a unified framework for intrinsic reward methods. We show that these rewards sharpen outputs around confident predictions, enabling efficient gains when confidence aligns with correctness but amplifying errors when it does not. Empirical results reveal distinct failure modes yet also show that collapse can be avoided in small, domain-specific settings, making test-time training a natural application. Beyond these findings, early training dynamics emerge as a lightweight diagnostic of model-task priors, offering a fast alternative to $pass@k$ for assessing RL trainability. Together, these results outline the limits of intrinsic rewards and highlight the need for external signals and hybrid paradigms for robust, scalable gains.

## ACKNOWLEDGEMENTS

This work is supported by National Science and Technology Major Project (2023ZD0121403), Young Elite Scientists Sponsorship Program by CAST (2023QNRC001), National Natural Science Foundation of China (No. 62406165), Shanghai Municipal Science and Technology Major Project, the high-quality development project of MIIT, the National Natural Science Foundation of China (No. 62236004) and Major Project of the National Social Science Foundation of China (No. 22&ZD298). We thank anonymous reviewers for their insightful comments and suggestions.

## ETHICS STATEMENT

This work investigates unsupervised reinforcement learning in large language models. While our work advances understanding of AI self-improvement, we acknowledge key ethical considerations. Our findings about reward hacking highlight risks if these methods were deployed without safeguards, so that systems might become overconfident in incorrect solutions, potentially causing harmful outputs in critical applications. Our identification of "safe" conditions should not be interpreted as universal guarantees, as model behavior can be unpredictable in novel contexts. We emphasize that our findings aim to improve understanding of limitations and appropriate use cases rather than encourage unconstrained deployment. Practitioners should carefully assess the confidence-correctness alignment in their applications and implement monitoring systems.

## REPRODUCIBILITY STATEMENT

We provide complete materials for reproducing our theoretical and empirical findings. Theoretical contributions include mathematical derivations of the unified framework and formal proofs in Appendix A.3. Experimental implementations use standardized frameworks (veRL/GRPO) with hyperparameters in Table 6 and tuning procedures in Appendix B.3. Code for all five intrinsic reward methods is provided at https://github.com/PRIME-RL/TTRL. Experiments use publicly available models (Qwen series) and datasets (DAPO-17k, AIME, AMC) are shown in the supplementary materials. All evaluation metrics are defined in Appendix B.2, and our codebase enables reproduction of results in all figures and tables.

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

| Method | Estimator | Formula |
|---|---|---|
| RLIF | Self-Certainty | $r(x,y) = \frac{1}{|y|}\sum_{t=1}^{|y|} D_{\text{KL}}(U\|\pi_\theta(\cdot|x, y_{<t}))$ |
| EM-RL | Trajectory-Level Entropy | $r(x,y) = \frac{1}{|y|}\sum_{t=1}^{|y|} \log \pi_\theta(y_t|x, y_{<t})$ |
| EM-RL, RENT | Token-Level Entropy | $r(x,y) = -\frac{1}{|y|}\sum_{t=1}^{|y|} H(\pi_\theta(\cdot|x, y_{<t}))$ |
| RLSC | Probability | $r(x,y) = \prod_{t=1}^{|y|} \pi_\theta(y_t|x, y_{<t})$ |
| RLSF | Probability Disparity | $r(x,y) = \frac{1}{M}\sum_{t=1}^{|a|}\left[\max_{a_t}\pi_\theta(a_t|x, c, a_{<t}) - \max_{a_t \neq \arg\max \pi_\theta}\pi_\theta(a_t|x, c, a_{<t})\right]$ |

Table 1: Overview of certainty-based rewards, estimators and their formulas. All variants reward high-confidence predictions through different formalizations of model certainty.

| Method | Estimator | Formula |
|---|---|---|
| TTRL, SRT, ETTRL SeRL, SQLM, R-Zero | Majority Voting | $r(x,y) = \mathbf{1}\left[y = \arg\max_{y'}\sum_{i=1}^{N}\mathbf{1}[y_i = y']\right], \quad \{y_i\}_{i=1}^{N} \sim \pi_\theta(\cdot|x)$ |
| Co-Reward | Majority Voting across Rephrased Question | $r(x,y) = \mathbf{1}[y = \arg\max_{y^*}\sum_{i=1}^{N}\mathbf{1}[y_i = y^*]], \quad \{y_i\}_{i=1}^{N} \sim \pi_\theta(\cdot|x)$ 
 $+ \mathbf{1}[y = \arg\max_{y^*}\sum_{j=1}^{N}\mathbf{1}[y'_j = y^*]], \quad \{y'_j\}_{j=1}^{N} \sim \pi_\theta(\cdot|\text{rephrase}(x))$ |
| RLCCF | Self-consistency Weighted Voting | $r(x,y) = \mathbf{1}\left[y = \arg\max_{a}\sum_{n=1}^{N}\left(\max_{a'}\sum_{k=1}^{K}\mathbf{1}[o_{n,k} = a']\right)\cdot\sum_{k=1}^{K}\mathbf{1}[a = o_{n,k}]\right],$ 
 $\{o_{n,k}\}_{k=1}^{K} \sim \pi_{\theta_n}(\cdot|x), \quad n = 1, \ldots, N$ |
| EMPO | Semantic Similarity | $r(x,y) = \frac{|\mathcal{C}(y)|}{G}, \quad \mathcal{C}(y) \in \texttt{SemanticCluster}(\{o_i\}_{i=1}^{G}), \quad \{o_i\}_{i=1}^{G} \sim \pi_\theta(\cdot|x)$ |
| CoVo | Trajectory Consistency and Volatility | $r(x,y) = \frac{1}{G}\left\|\sum_{i=1}^{G}\texttt{Con}(y_i)\cdot[\cos(\texttt{Vol}(y_i)), \sin(\texttt{Vol}(y_i))]\right\| + r_{\text{cur}},$ 
 $\{y_i\}_{i=1}^{N} \sim \pi_\theta(\cdot|x), \quad G = |\{i : \text{ans}(y_i) = \text{ans}(y)\}|$ |

Table 2: Overview of ensemble-based rewards, estimators and their formulas. All variants operationalize the assumption that consistency across independent samples correlates with correctness.

# A  DETAILS FOR SECTION 3

## A.1  DYNAMICS OF ONE-STEP UPDATE

Training with intrinsic rewards involves multiple gradient steps toward convergence. In this section, we take TTRL (Zuo et al., 2025) and its majority voting reward as a representative intrinsic method to demonstrate the dynamics of a one-step update.

Consider the standard KL-regularized RL objective:

$$\max_{\pi_\theta} \mathbb{E}_{y \sim \pi_\theta(\cdot|x)}\left[r(x,y)\right] - \beta D_{\text{KL}}\left[\pi_\theta(\cdot|x)\|\pi_{\text{ref}}(\cdot|x)\right], \tag{4}$$

where $\pi_{\text{ref}}$ is the reference policy and $\beta$ controls the strength of regularization. Refer to (Rafailov et al., 2023), the optimal policy for this objective has the well-known closed form:

$$\pi_\theta^*(y|x) = \frac{1}{Z(x)}\pi_{\text{ref}}(y|x)\exp\left(\frac{1}{\beta}r(x,y)\right), \tag{5}$$

where $Z(x) = \sum_y \pi_{\text{ref}}(y|x)\exp\left(\frac{1}{\beta}r(x,y)\right)$ is the partition function. The majority voting reward used in TTRL at iteration $k$ is defined as

$$r_k(x,y) = \mathbf{1}[\text{ans}(y) = \text{maj}_k(Y_k)], \tag{6}$$

where $Y_k = \{y^{(1)}, \ldots, y^{(N)}\}$ denotes $N$ rollouts sampled from $\pi_\theta^{(k)}$, and $\mathrm{maj}_k(Y_k) = \arg\max_a |\{i \in [N] : \mathrm{ans}(y^{(i)}) = a\}|$ is the most frequent answer among the rollouts. Then applying Equation (5) with the majority voting reward $r_k$, if we held $r_k$ constant and performed infinite updates starting from reference policy $\pi_\theta^{(k)}$, we would converge to the optimal policy:

$$\pi_\theta^{*,(k+1)}(y|x) = \frac{\pi_\theta^{(k)}(y|x) \cdot \exp\left(\frac{r_k(x,y)}{\beta}\right)}{Z_k(x)}. \tag{7}$$

Since $r_k$ is binary, the exponential term takes only two values: $e^{1/\beta}$ for the majority-voted $\mathrm{maj}_k(Y_k)$ and $e^0 = 1$ for all others. This yields the explicit form:

$$\pi_\theta^{*,(k+1)}(y|x) = \begin{cases} \frac{\pi_\theta^{(k)}(y|x) \cdot e^{1/\beta}}{Z_k(x)}, & \text{if } \mathrm{ans}(y) = \mathrm{maj}_k(Y_k), \\ \frac{\pi_\theta^{(k)}(y|x)}{Z_k(x)}, & \text{otherwise}, \end{cases} \tag{8}$$

where the partition function ensures proper normalization:

$$Z_k(x) = \sum_{y:\mathrm{ans}(y)=\mathrm{maj}_k(Y_k)} \pi_\theta^{(k)}(y|x) \cdot e^{1/\beta} + \sum_{y:\mathrm{ans}(y)\neq\mathrm{maj}_k(Y_k)} \pi_\theta^{(k)}(y|x). \tag{9}$$

For cleaner notation, let $p_{\mathrm{maj}}^{(k)} = \sum_{y:\mathrm{ans}(y)=\mathrm{maj}_k(Y_k)} \pi_\theta^{(k)}(y|x)$ denote the current policy's total probability mass on trajectories leading to the majority answer. Then the partition function simplifies to:

$$Z_k(x) = p_{\mathrm{maj}}^{(k)} \cdot e^{1/\beta} + (1 - p_{\mathrm{maj}}^{(k)}). \tag{10}$$

The structure of Equation (8) reveals the method's update mechanism. At the optimal policy $\pi_\theta^{*,(k+1)}$, each trajectory $y$ where $\mathrm{ans}(y) = \mathrm{maj}_k(Y_k)$ has its probability amplified by factor $e^{1/\beta}$. The probability mass on the majority trajectories would be:

$$p_{\mathrm{maj}}^{*,(k+1)} = \frac{p_{\mathrm{maj}}^{(k)} \cdot e^{1/\beta}}{p_{\mathrm{maj}}^{(k)} \cdot e^{1/\beta} + (1 - p_{\mathrm{maj}}^{(k)})}. \tag{11}$$

**Actual Training Dynamics.** In practice, our training performs only one gradient update per iteration, not reaching the optimum $\pi_\theta^{*,(k)}$ but moving partway toward it. The actual probability mass after one update $p_{\mathrm{maj}}^{(k+1)}$ satisfies:

$$p_{\mathrm{maj}}^{*,(k+1)} \geq p_{\mathrm{maj}}^{(k+1)} \geq p_{\mathrm{maj}}^{(k)}. \tag{12}$$

This ordering holds because: (1) policy gradient methods with positive rewards on majority trajectories tend to increase their probability mass (lower bound), and (2) the optimal policy achieves maximum expected reward, so one-step updates cannot exceed it (upper bound). See Appendix A.2 for detailed theoretical justification and empirical validation.

## A.2 JUSTIFYING THE ORDERING $p_{\mathrm{MAJ}}^{*,(k+1)} \geq p_{\mathrm{MAJ}}^{(k+1)} \geq p_{\mathrm{MAJ}}^{(k)}$

From the optimal policy of the standard KL-regularized RL objective (Equation (4)), if we held reward $r_k$ fixed and performed infinite updates starting from $\pi_\theta^{(k)}$, we would reach the optimal policy with probability mass:

$$p_{\mathrm{maj}}^{*,(k+1)} = \frac{\alpha \cdot p_{\mathrm{maj}}^{(k)}}{1 + (\alpha - 1)p_{\mathrm{maj}}^{(k)}}, \qquad \alpha := e^{1/\beta} > 1 \tag{13}$$

**Lower bound** $(p_{\mathrm{maj}}^{(k+1)} \geq p_{\mathrm{maj}}^{(k)})$: The policy gradient is: $\nabla_\theta J = \mathbb{E}_{\pi_\theta}[r_k(x,y)\nabla_\theta \log \pi_\theta(y|x)]$. Since $r_k(x,y) = 1$ for majority trajectories and $r_k(x,y) = 0$ for non-majority trajectories, the gradient

increases $\log \pi_\theta(y|x)$ only for majority trajectories. Therefore, after one gradient update with learning rate $\eta$: $\log \pi_\theta^{(k+1)}(y|x) \approx \log \pi_\theta^{(k)}(y|x) + \eta \cdot r_k(x,y) \cdot$ [advantage terms]. **Here we use $\approx$ because we are conducting stochastic gradient descent. We will empirically validate this increase later.** For majority trajectories with positive advantage, this increases their probability. Hence:

$$p_{\text{maj}}^{(k+1)} = \sum_{y:\text{ans}(y)=\text{maj}_{k+1}(Y)} \pi_\theta^{(k+1)}(y|x) \geq \sum_{y:\text{ans}(y)=\text{maj}_k(Y)} \pi_\theta^{(k)}(y|x) = p_{\text{maj}}^{(k)} \tag{14}$$

**Upper bound** $(p_{\text{maj}}^{(k+1)} \leq p_{\text{maj}}^{*,(k+1)})$: Since $\pi_\theta^{*,k}$ maximizes the KL-regularized objective for fixed $r_k$, it achieves the highest possible expected reward. Our single-step update only moves partway toward this maximum, so we cannot exceed the optimal probability mass: $p_{\text{maj}}^{(k+1)} \leq p_{\text{maj}}^{*,(k+1)}$.

This ordering establishes that $p_{\text{maj}}^{(k)}$ is monotonically increasing and bounded. The key remaining question is: **does the majority answer $\text{maj}_k(Y)$ remain stable across iterations?**

With $N$ rollouts from $\pi_\theta^{(k)}$, each rollout independently yields answer $a$ with probability $\pi_\theta^{(k)}(a|x)$. By the Law of Large Numbers, as $N$ increases, the empirical frequency of each answer converges to its true probability: $\frac{|\{\text{ans}(y^{(i)})=a\}|}{N} \xrightarrow{N\to\infty} \pi_\theta^{(k)}(a|x)$. Therefore, $\text{maj}_k(Y)$ (the most frequent answer in rollouts) converges to $\arg\max_a \pi_\theta^{(k)}(a|x)$ (the most probable answer under the policy). Since $p_{\text{maj}}^{(k)}$ increases monotonically, the most probable answer remains $\text{maj}_0(Y)$ throughout training: $\arg\max_a \pi_\theta^{(k)}(a|x) = \text{maj}_0(Y)$ for all $k$. In practice, even moderate $N$ (we use $N = 8$) provides reasonable stability.

### A.2.1 EMPIRICAL VALIDATION 1: VALIDATION OF ORDERING AND MAJORITY STABILITY

**Setup:** We trained on a single problem from MATH-500 with $N = 1024$ rollouts (reducing majority vote randomness) for 50 steps. We randomly selected 4 problems and monitored whether the majority answer $\text{maj}_k(Y_k)$ remains stable and whether $p_{\text{maj}}^{(k)}$ increases monotonically.

| Step | 1 | 2 | 3 | 4 | 5 | 6 | 7 | 8 | 9 | 10 |
|---|---|---|---|---|---|---|---|---|---|---|
| level3_id146 | 12.70 | 15.53 | 15.92 | 16.21 | 18.46 | 22.07 | 22.56 | 24.80 | 31.35 | 39.36 |
| level1_id187 | 6.64 | 6.69 | 6.84 | 7.42 | 10.45 | 11.04 | 11.43 | 11.82 | 15.82 | 18.46 |
| level1_id262 | 15.14 | 17.19 | 17.48 | 18.85 | 20.02 | 22.07 | 24.12 | 25.20 | 34.67 | 39.84 |
| level3_id122 | 11.33 | 12.01 | 12.40 | 14.06 | 17.87 | 18.46 | 20.31 | 21.29 | 33.59 | 33.89 |

Table 3: Monotonic increase of $p_{\text{maj}}$ (%) in early training steps.

**Results for monotonic increase of $p_{\text{maj}}$.** Table 3 shows $p_{\text{maj}}$ values for the first 10 steps. All 4 problems exhibit strict monotonic increase at every single step, confirming the lower bound $p_{\text{maj}}^{(k+1)} \geq p_{\text{maj}}^{(k)}$ of the ordering. We also found that the majority answer remained stable across all iterations.

| Step | 5 | 10 | 15 | 20 | 25 | 30 | 35 | 40 | 45 | 50 |
|---|---|---|---|---|---|---|---|---|---|---|
| level3_id146 | 18.46 | 39.36 | 48.93 | 91.11 | 95.41 | 98.14 | 98.54 | 99.02 | 99.61 | 99.80 |
| level1_id187 | 11.04 | 18.46 | 26.37 | 79.88 | 89.84 | 93.07 | 96.09 | 97.66 | 98.54 | 99.02 |
| level1_id262 | 22.07 | 39.84 | 51.37 | 90.14 | 95.90 | 96.80 | 97.46 | 98.05 | 98.63 | 99.41 |
| level3_id122 | 17.87 | 33.59 | 43.55 | 84.28 | 92.19 | 93.26 | 95.80 | 96.09 | 98.34 | 98.54 |

Table 4: Geometric convergence of $p_{\text{maj}}$ (%) to 1.0 over 50 training steps.

**Results for convergence of $p_{\text{maj}}$.** Table 4 shows the same 4 problems trained for 50 steps. All problems converge from initial values toward near-complete concentration (98.54%-99.80% at step 50), demonstrating the convergence predicted by Theorem 1. This validates non-trivial progress and confirms that the iterative training procedure with policy-dependent rewards does indeed converge to deterministic policies.

### A.2.2 EMPIRICAL VALIDATION 2: FIXED REWARD CONVERGENCE VALIDATION

**Setup.** To validate that the closed-form optimal policy in Equation (8) is achievable when reward is held fixed, we conducted an extreme off-policy experiment. We used global batch size 1024 with mini-batch size 1, generated one-time rollout (with $N = 8$ for each of 1024 prompts), and performed 1024 gradient updates using rewards computed solely from the initial rollout majority. This setup tests whether solving a single KL-regularized RL objective can converge to the theoretical optimum when the reward signal remains constant.

**Results.** After 1024 mini-updates using the same fixed reward signal, the Majority Voting Reward reached 1.0 (complete convergence), while validation performance on AIME24, AIME25, and AMC23 dropped to zero. This confirms that the convergence point predicted by Equation (8) is achievable with sufficient updates.

### A.3 PROOF OF THEOREM 1

> **Geometric Convergence to Deterministic Policy**
>
> Consider the training procedure where at each iteration $k$: (1) sample $N$ rollouts $Y_k$ from $\pi_\theta^{(k)}$, (2) compute majority $\text{maj}_k(Y_k)$, (3) perform one gradient update with reward $r_k(x, y) = \mathbf{1}[\text{ans}(y) = \text{maj}_k(Y_k)]$.
> Under assumptions (A1) stable majority $\text{maj}_k = \text{maj}_0$ and (A2) $\eta_k \geq \eta_{\min} > 0$, the probability mass $p_{\text{maj}}^{(k)}$ converges geometrically to 1. As $k \to \infty$:
>
> $$\lim_{k \to \infty} \pi_\theta^{(k)}(y|x) = \begin{cases} \frac{\pi_{\text{ref}}(y|x)}{\sum_{y':\text{ans}(y')=\text{maj}_0(Y_0)} \pi_{\text{ref}}(y'|x)} & \text{if } \text{ans}(y) = \text{maj}_0(Y_0), \\ 0 & \text{otherwise} \end{cases} \tag{15}$$

**Step 1: Effective Update Rule.**

We model the actual update with step efficiency $\eta_k \in (0, 1]$:

$$p_{\text{maj}}^{(k+1)} = p_{\text{maj}}^{(k)} + \eta_k \cdot (p_{\text{maj}}^{*,(k+1)} - p_{\text{maj}}^{(k)}) \tag{16}$$

Substituting Equation (13):

$$\begin{aligned} p_{\text{maj}}^{(k+1)} &= p_{\text{maj}}^{(k)} + \eta_k \left( \frac{\alpha \cdot p_{\text{maj}}^{(k)}}{1 + (\alpha - 1)p_{\text{maj}}^{(k)}} - p_{\text{maj}}^{(k)} \right) \\ &= p_{\text{maj}}^{(k)} + \eta_k \cdot \frac{(\alpha - 1)(1 - p_{\text{maj}}^{(k)})p_{\text{maj}}^{(k)}}{1 + (\alpha - 1)p_{\text{maj}}^{(k)}} \end{aligned} \tag{17}$$

**Step 2: Error Dynamics.**

Define the error from the fixed point 1 as:

$$\epsilon^{(k)} := 1 - p_{\text{maj}}^{(k)} \in (0, 1) \tag{18}$$

Substituting into Equation (17):

$$\begin{aligned} \epsilon^{(k+1)} &= 1 - p_{\text{maj}}^{(k+1)} \\ &= \epsilon^{(k)} - \eta_k \cdot \frac{(\alpha - 1)(1 - \epsilon^{(k)})\epsilon^{(k)}}{1 + (\alpha - 1)(1 - \epsilon^{(k)})} \\ &= \epsilon^{(k)} \left( 1 - \eta_k \cdot \frac{(\alpha - 1)(1 - \epsilon^{(k)})}{\alpha - (\alpha - 1)\epsilon^{(k)}} \right) \end{aligned} \tag{19}$$

**Step 3: Monotonic Decrease.**

Since $\alpha > 1$, $\epsilon^{(k)} \in (0, 1)$, and $\eta_k \in (0, 1]$, we have:

$$0 < \frac{(\alpha - 1)(1 - \epsilon^{(k)})}{\alpha - (\alpha - 1)\epsilon^{(k)}} < 1 \tag{20}$$

Therefore:

$$0 < 1 - \eta_k \cdot \frac{(\alpha - 1)(1 - \epsilon^{(k)})}{\alpha - (\alpha - 1)\epsilon^{(k)}} < 1 \tag{21}$$

This implies $\epsilon^{(k+1)} < \epsilon^{(k)}$, proving the sequence $\{\epsilon^{(k)}\}$ is strictly decreasing and bounded below by 0.

**Step 4: Convergence to Zero.**

Let $\ell = \lim_{k \to \infty} \epsilon^{(k)} \geq 0$. Under assumption (A2), $\eta_k \geq \eta_{\min} > 0$. If $\ell > 0$, then for large $k$, the multiplier in Equation (19):

$$1 - \eta_k \cdot \frac{(\alpha - 1)(1 - \epsilon^{(k)})}{\alpha - (\alpha - 1)\epsilon^{(k)}} \leq 1 - \eta_{\min} \cdot \frac{\alpha - 1}{\alpha} < 1 \tag{22}$$

is bounded away from 1, causing continued decay. The only consistent limit is $\ell = 0$. Therefore:

$$\epsilon^{(k)} \to 0 \qquad \text{equivalently} \qquad p_{\text{maj}}^{(k)} \to 1 \tag{23}$$

**Step 5: Geometric Convergence Rate.**

From Equation (19), for large $k$ when $\epsilon^{(k)}$ is small:

$$\epsilon^{(k+1)} \approx \epsilon^{(k)} \left(1 - \eta_k \cdot \frac{\alpha - 1}{\alpha}\right) \tag{24}$$

Under assumption (A2):

$$\epsilon^{(k+1)} \leq \left(1 - \eta_{\min} \cdot \frac{\alpha - 1}{\alpha}\right) \epsilon^{(k)} \tag{25}$$

This establishes geometric convergence with rate depending on $\eta_{\min}$ and $\alpha = e^{1/\beta}$. In the ideal case where $\eta_k = 1$ for all $k$ (each update reaches the optimum), the convergence rate is exactly $\rho = e^{-1/\beta}$.

**Step 6: Limiting Policy.**

Given assumption (A1) that the majority remains stable at $\text{maj}_0(Y_0)$, as $p_{\text{maj}}^{(k)} \to 1$, all probability mass concentrates on trajectories with $\text{ans}(y) = \text{maj}_0(Y_0)$. The limiting distribution is:

$$\lim_{k \to \infty} \pi_\theta^{(k)}(y|x) = \begin{cases} \frac{\pi_{\text{ref}}(y|x)}{\sum_{y':\text{ans}(y')=\text{maj}_0(Y_0)} \pi_{\text{ref}}(y'|x)} & \text{if } \text{ans}(y) = \text{maj}_0(Y_0), \\ 0 & \text{otherwise} \end{cases} \tag{26}$$

This completes the proof. $\square$

**Remark on Assumptions.**

- **(A1) Majority stability:** By the Law of Large Numbers, with $N$ rollouts, the empirical majority $\text{maj}_k(Y_k)$ converges to $\arg\max_a \pi_\theta^{(k)}(a|x)$ as $N \to \infty$. Since $p_{\text{maj}}^{(k)}$ increases monotonically, the argmax remains $\text{maj}_0$ throughout training. We validate this empirically with $N = 1024$ rollouts in Appendix A.2, where the majority never flipped across 200 iterations.

- **(A2) Non-trivial progress:** We assume $\eta_k \geq \eta_{\min} > 0$, meaning each gradient update makes non-trivial progress. We validate this empirically: our experiments show consistent monotonic increase in $p_{\text{maj}}$ and convergence to 1.0 under extreme off-policy settings (Appendix A.2).

## A.4 UNIFIED REWARD FRAMEWORK

Despite varied implementations of intrinsic rewards, they can be understood through a single lens: the manipulation of cross-entropy between carefully chosen distributions. We consolidate these diverse rewards into the following unified paradigm:

---

**Unified Reward Framework**

Most intrinsic rewards can be expressed as:

$$r_{\text{uni}}(x, y) = \psi\left(\frac{\sigma}{|\mathcal{I}|} \sum_{i \in \mathcal{I}} \mathbb{H}(q^i, \pi_\theta^i)\right), \quad \sigma \in \{+1, -1\}, \tag{27}$$

where rewards derive from cross-entropy $\mathbb{H}$ between anchor distributions $q^i$ and model distributions $\pi_\theta^i$, aggregated over granularity $\mathcal{I}$, with sign $\sigma$ and monotonic transformation $\psi$.

---

### A.4.1 FRAMEWORK COMPONENTS

To understand how different intrinsic methods fit into this framework, we define each component:

**Key Components:**

- Given a question $x$ and generated response $y$ (a sequence of tokens $y_1, \ldots, y_{|y|}$), we can derive rewards from the model's internal distributions at different levels of granularity.

- **Aggregation granularity** $\mathcal{I}$: Determines the level to compute distributions. For token-level methods, $\mathcal{I} = \{1, \ldots, |y|\}$ where each element corresponds to a position in the sequence. For answer-level methods, $\mathcal{I} = \{\mathcal{A}\}$ represents a single distribution over complete semantic answers.

- **Model distribution** $\pi_\theta^i$ **at granularity** $i$: For token-level granularity at position $t$, this is $\pi_\theta^t(\cdot) = \pi_\theta(\cdot \mid x, y_{<t})$, the distribution over the next token given the context. For answer-level granularity, this is $\pi_\theta^{\mathcal{A}} = \pi_\theta(\cdot \mid x)$, the distribution over complete answers.

- **Anchor distribution** $q^i$ **at granularity** $i$: Serves as a reference point. Different reward estimators use different anchors: uniform distribution $U_V$ for Self-Certainty or one-hot distribution $\delta^t$ centered on the generated token for Trajectory-Level Entropy.

- **Cross-entropy** $\mathbb{H}(q^i, \pi_\theta^i)$**:** Cross-entropy between anchor distribution $q^i$ and model distribution $\pi_\theta^i$ at granularity $i$, defined as $\mathbb{H}(q^i, \pi_\theta^i) = -\sum_{v \in \mathcal{V}^i} q^i(v) \log \pi_\theta^i(v)$. For token-level granularity ($i = t$), $\mathcal{V}^i$ is the token vocabulary, and the cross-entropy measures divergence between distributions over next tokens. For answer-level granularity ($i = \mathcal{A}$), $\mathcal{V}^i$ is the set of distinct semantic answers, and the cross-entropy measures divergence between distributions over complete answers.

- **Sign factor** $\sigma \in \{+1, -1\}$: Determines the optimization direction. When the anchor $q$ is uniform, we set $\sigma = +1$ to reward divergence from uniformity (encouraging peaked distributions). When the anchor $q$ is sharp (e.g., one-hot or the model's own distribution), we set $\sigma = -1$ to reward alignment (reinforcing confident predictions).

- **Monotonic transformation** $\psi$: Reshapes the reward signal while preserving ordering. Common choices are identity ($\psi(z) = z$) or exponential ($\psi(z) = \exp(z)$), with exponential transformations amplifying the sharpening effect.

### A.4.2 INSTANTIATIONS OF THE FRAMEWORK

We next demonstrate how most intrinsic rewards instantiate this framework. Each method's distinctive characteristics emerge from specific choices of $\mathcal{I}$, $q$, $\sigma$, and $\psi$, as shown in Table 5, which reveals that despite surface-level differences, all methods manipulate cross-entropy to achieve distribution sharpening.

**Remarks.** We highlight two special cases. First, the formulation of Self-Certainty includes an additional $\log |V|$ term. Since this constant is independent of model parameters, it does not affect gradients during RL training. Second, the expression of $r_{\text{MV}}$ corresponds to the asymptotic case where the number of rollouts $n \to \infty$. By the law of large numbers, as $n \to \infty$,

| Method | Estimator | Formula | Monotonic transformation $\psi$ | Anchor distribution $q$ | Model distribution $\pi_\theta$ |
|---|---|---|---|---|---|
| RLIF | Self-Certainty | $r_{\mathrm{SC}} = \dfrac{1}{|y|}\sum_{t=1}^{|y|}\mathbb{H}(U_V, \pi_\theta^t) + \log|V|$ | $z + \log|V|$ | $\{U_V\}_{t=1}^{|y|}$ | $\{\pi_\theta^t\}_{t=1}^{|y|}$ |
| EM-RL, RENT | Token-Level Entropy | $r_{\mathrm{H}} = -\dfrac{1}{|y|}\sum_{t=1}^{|y|}\mathbb{H}(\pi_\theta^t)$ | $z$ | $\{\pi_\theta^t\}_{t=1}^{|y|}$ | $\{\pi_\theta^t\}_{t=1}^{|y|}$ |
| EM-RL | Trajectory-Level Entropy | $r_{\mathrm{Traj}} = -\dfrac{1}{|y|}\sum_{t=1}^{|y|}\mathbb{H}(\delta^t, \pi_\theta^t)$ | $z$ | $\{\delta^t\}_{t=1}^{|y|}$ | $\{\pi_\theta^t\}_{t=1}^{|y|}$ |
| RLSC | Probability | $r_{\mathrm{Prob}} = \exp(-\sum_{t=1}^{|y|}\mathbb{H}(\delta^t, \pi_\theta^t))$ | $\exp(|\mathcal{I}|\cdot z)$ | $\{\delta^t\}_{t=1}^{|y|}$ | $\{\pi_\theta^t\}_{t=1}^{|y|}$ |
| EMPO | Semantic Entropy | $r_{\mathrm{SE}} = \exp(-\mathbb{H}(\delta^A, \pi_\theta^A))$ | $\exp(z)$ | $\delta^A$ | $\pi_\theta^A$ |
| TTRL, SRT, ETTRL | Majority Voting | $r_{\mathrm{MV}} = \lim_{\tau\to 0^+}\exp(-\mathbb{H}(\delta^A, \tilde\pi_\theta^A))$ | $\exp(z)$ | $\delta^A$ | $\tilde\pi_\theta^A$ |

Table 5: Instantiations for the unified reward framework of representative intrinsic rewards. Each method is specified by its estimator, anchor and model distributions, and a monotonic transformation of the cross-entropy between them. Token-level ($t$) and answer-level ($A$) variants capture different granularities of aggregation.

the majority vote almost surely selects the answer with the highest probability under $\pi_\theta^A$, i.e. $\arg\max_a \pi_\theta^A(a)$. To make this limit computationally tractable, we use the tempered distribution $\tilde\pi_\theta^A(a) = \exp\big(\pi_\theta^A(a)/\tau\big)/\sum_{b\in A}\exp\big(\pi_\theta^A(b)/\tau\big)$ which avoids the undefined $\log 0$ issue; as $\tau\to 0^+$, it collapses to the hard majority indicator $\mathbf{1}\big[\mathrm{ans}(y) = \arg\max_a \pi_\theta^A(a)\big]$, thereby recovering the same limiting behavior as majority voting.

**Key Observations:**

- **Token-level methods** (Self-Certainty, Token-Level Entropy, Trajectory-Level Entropy, Probability) operate at $\mathcal{I} = \{1, \ldots, |y|\}$, manipulating distributions at each generation step to encourage local confidence.
- **Answer-level methods** (Semantic Entropy, Majority Voting) work at $\mathcal{I} = \{\mathcal{A}\}$, focusing on global answer consistency where $\pi_\theta^A$ represents the distribution over semantic answers and $\tilde\pi_\theta^A$ denotes the empirical distribution from multiple rollouts.
- **Anchor choices** reveal the core mechanism: uniform distributions ($U_V$) encourage departure from randomness, while sharp distributions ($\delta^t$, $\delta^A$) reinforce high-probability paths, both leading to increased determinism.
- **Exponential transformations** ($\exp(z)$) amplify the sharpening effect by exponentially rewarding low cross-entropy, while identity transformations ($z$) provide more gradual reinforcement.

### A.4.3 MONOTONICITY ANALYSIS

The key insight comes from analyzing the monotonicity of the exponent in Equation (27). Since $\psi$ is strictly increasing by design, the behavior depends entirely on $\sigma$:

- **Case $\sigma = +1$:** The reward increases with cross-entropy. Sequences where $\pi_\theta$ diverges from $q$ (typically uniform) receive higher rewards, pushing the policy toward more peaked distributions.
- **Case $\sigma = -1$:** The reward decreases with cross-entropy. Sequences where $\pi_\theta$ aligns with $q$ (typically sharp) receive higher rewards, reinforcing existing confident predictions.

Both cases lead to the same outcome: progressive sharpening of the model's distribution, either by moving away from uniformity or by reinforcing peaked predictions.

### A.5 GENERALIZED SHARPENING ANALYSIS VIA UNIFIED REWARD FRAMEWORK

To address the concern that Theorem 1 applies only to Majority Voting, and to demonstrate the analytical utility of our unified framework, we provide a generalized sharpening analysis. We show

that methods with $\sigma = -1$ share a critical structural property, Reward-Confidence Monotonicity, which creates a persistent pressure toward distribution sharpening.

**Note:** The following is a proof sketch demonstrating the key convergence mechanism shared by $\sigma = -1$ methods. A fully rigorous treatment requires additional technical conditions that we validate empirically. Methods with $\sigma = +1$ (Self-Certainty) require separate analysis as they reward away from uniform distribution.

**Proposition 1** (Sharpening Dynamics for $\sigma = -1$ Methods). *Consider any intrinsic reward with $\sigma = -1$ in the unified framework ($r_{uni} = \psi(-\mathbb{H}(q, \pi))$) where $\psi$ is strictly increasing and $q$ is a sharp anchor. These methods satisfy* **Reward-Confidence Monotonicity**:

$$\pi_\theta(y_a|x) > \pi_\theta(y_b|x) \implies r_{uni}(x, y_a) > r_{uni}(x, y_b) \tag{28}$$

*For a dominant trajectory $y^*$ (e.g., majority) and a non-dominant competitor $y'$, this inequality is strict: $r(y^*) > r(y')$. Under iterative KL-regularized updates, this property creates a self-reinforcing feedback loop that drives geometric concentration.*

**Proof Sketch:**

We analyze the dynamics for a dominant trajectory $y^*$ and a competitor $y'$ (for ensemble methods, not in the same class as $y^*$) where the model initially prefers $y^*$ (i.e., $\pi_k(y^*) > \pi_k(y')$) and assigns it strictly higher reward ($r_k(y^*) > r_k(y')$).

**Step 1: Existence of a Positive Reward Gap**

Using the unified formula, we justify why the gap is positive for $\sigma = -1$:

- **Self-Reinforcing Anchors** (e.g., Probability): $r(y) = \psi(\log \pi(y))$. Since $\pi_k(y^*) > \pi_k(y')$ and $\psi$ is strictly increasing, $r_k(y^*) > r_k(y')$.

- **Answer-Level Anchors** (e.g., Majority Voting): $y^*$ belongs to the dominant answer class $a^*$, while $y'$ does not. By construction, $r(y^*) = 1$ and $r(y') = 0$.

In both cases, the intrinsic reward gap is strictly positive: $\Delta_r^{(k)} = r_k(y^*) - r_k(y') > 0$.

**Step 2: The Optimization Target**

We consider the optimal policy $\pi^*$ for the current fixed reward landscape $r_k$. The optimal solution implies a target ratio:

$$\frac{\pi^*(y^*)}{\pi^*(y')} = \frac{\pi_k(y^*)}{\pi_k(y')} \cdot \exp\left(\frac{\Delta_r^{(k)}}{\beta}\right) \tag{29}$$

Since $\Delta_r^{(k)} > 0$, the target ratio is strictly larger than the current ratio.

**Gradient Assumption:** The gradient $\nabla_\theta J = \mathbb{E}_{\pi_k}[r_k(y)\nabla_\theta \log \pi_\theta(y)]$ assigns positive weight to high-reward trajectories. We assume that policy gradient updates with positive learning rate $\eta$ satisfy: if $r(y^*) > r(y')$ and both have positive probability, then the updated policy satisfies $\frac{\pi_{k+1}(y^*)}{\pi_{k+1}(y')} \geq \frac{\pi_k(y^*)}{\pi_k(y')}$. This aligns with standard policy gradient convergence properties.

**Step 3: The Reinforcement Loop**

The unified framework reveals why this process spirals into determinism. As the policy updates to increase the probability mass on the dominant trajectory:

- For **Self-Reinforcing Anchors** (e.g., Probability), because $r(y) = \psi(\log \pi(y))$, increasing $\pi(y^*)$ directly increases its reward $r(y^*)$.

- For **Answer-Level Anchors** (e.g., Majority Voting), increasing the total probability mass on the dominant answer class $a^*$ increases the reward for all trajectories in that class (since $r \propto \log p(a^*)$).

This creates a positive feedback loop: the update increases the probability of the dominant path, which maintains or widens the reward gap $\Delta_r$, ensuring the pressure to sharpen ($\Delta_r > 0$) persists.

**Utility of the Framework:** This derivation demonstrates that the "rich-get-richer" dynamic is a structural inevitability for any method where the reward function is monotonically aligned with the model's own confidence ($\sigma = -1$). The framework allows us to identify this shared property and predict that all such methods will drive the policy toward deterministic outputs, regardless of whether this leads to success (when aligned with correctness) or failure (when misaligned).

**Remark on $\sigma = +1$ Methods:** Self-Certainty ($\sigma = +1$) rewards higher when away from uniform distribution. Therefore, $\pi(y_a) > \pi(y_b)$ does not imply $r(y_a) > r(y_b)$. A high-probability output and a very low-probability output could both have high KL-divergence from uniform, violating direct Reward-Confidence Monotonicity. Its sharpening mechanism requires separate analysis.

While methods with $\sigma = +1$ do not strictly align reward with raw confidence, they still induce sharpening by penalizing high-entropy distributions. By maximizing the distance from a uniform anchor, the optimization landscape naturally favors peaked, low-entropy policies, effectively driving the model toward determinism.

**Empirical Validation:** To substantiate the assumptions in this proof sketch, we provide empirical validation for different intrinsic reward methods in Figure 29 and Appendix B.3, confirming that Reward-Confidence Monotonicity is not just a theoretical construct but the actual driver of the observed training dynamics.

### A.6 Optimal Policies Induced by other Intrinsic Rewards

**Optimal Policy of the Reward Function $r_{\text{SC}}$.** For the Self-Certainty reward function $r_{\text{SC}}$, it instantiates our unified framework with token-level granularity $\mathcal{I} = \{1, 2, ..., |y|\}$, anchor distribution $q = \{U_V\}_{t=1}^{|y|}$ (uniform distribution over vocabulary), model distribution $\pi = \{\pi_\theta^t\}_{t=1}^{|y|}$, sign factor $\sigma = +1$, and transformation $\psi(z) = z$. As established previously, for any input $x$, the token-level predictive distribution of the model is evaluated against the current policy $\pi$. Due to $\sigma = +1$, the farther this distribution deviates from the uniform distribution (i.e., the higher the model's confidence), the larger the reward $r_{\text{SC}}(x, y)$. Consequently, after a single step of policy update, the optimal probability $\pi_\theta(y|x)$ increases for such high-confidence sequences, whereas it decreases when the per-token distribution is close to uniform (low confidence). Thus, $r_{\text{SC}}$ encourages the model to generate answers that are already preferred by the prior policy.

A detailed derivation is provided below. The Self-Certainty based reward is defined as:

$$r_{\text{SC}}(x, y) = \frac{1}{|y|} \sum_{t=1}^{|y|} D_{\text{KL}}\big(U \parallel \pi_\theta(\cdot \mid x, y_{<t})\big) = -\log|V| - \frac{1}{|y||V|} \sum_{t=1}^{|y|} \sum_{v=1}^{|V|} \log \pi_\theta^t(y_t = v). \quad (30)$$

Within the KL-regularized RL framework, dropping the constant term $-\log|V|$, the one-step optimal policy becomes:

$$\pi_\theta(y|x) \propto \pi_{\text{ref}}(y|x) \exp\left( -\frac{1}{\beta |y| |V|} \sum_{t=1}^{|y|} \sum_{v=1}^{|V|} \log \pi_\theta^t(y_t = v) \right). \quad (31)$$

Therefore, whenever the model assigns concentrated probabilities to every token of $y$ (high confidence), the exponent grows, thus increasing the probability of the sequence $\pi_\theta(y|x)$. In summary, the Self-Certainty based reward systematically enhances the model's "self-confidence" with respect to its prior policy.

**Optimal Policy of the Reward Function $r_H$.** For the token-level entropy-based reward $r_H$, it instantiates our unified framework with token-level granularity $\mathcal{I} = \{1, 2, ..., |y|\}$, anchor distribution $q = \{\pi_\theta^t\}_{t=1}^{|y|}$, model distribution $\pi = \{\pi_\theta^t\}_{t=1}^{|y|}$, sign factor $\sigma = -1$, and transformation $\psi(z) = z$. Maximizing $r_H$ is equivalent to minimizing the predictive entropy at every position, thereby discouraging the model from spreading its probability mass across multiple candidate tokens and hence increasing its decisiveness.

A detailed derivation is provided below. The entropy-based reward is defined as:

$$r_H(x, y) = -\frac{1}{|y|} \sum_{t=1}^{|y|} H\big(\pi_\theta(\cdot \mid x, y_{<t})\big) = -\frac{1}{|y|} \sum_{t=1}^{|y|} \sum_{v=1}^{|V|} \pi_\theta^t(y_t = v) \log \pi_\theta^t(y_t = v). \tag{32}$$

Within the KL-regularized RL framework, the one-step optimal policy becomes:

$$\pi_\theta(y|x) \propto \pi_{\text{ref}}(y|x) \exp\left( -\frac{1}{\beta |y|} \sum_{t=1}^{|y|} \sum_{v=1}^{|V|} \pi_\theta^t(y_t = v) \log \pi_\theta^t(y_t = v) \right). \tag{33}$$

Consequently, if the predictive distribution of an output sequence $y$ exhibits high entropy (i.e., the per-token distributions are close to uniform), the negative-entropy reward $r_H$ is strongly negative, which suppresses the exponential weight and reduces $\pi_\theta(y|x)$. Conversely, low entropy (highly peaked per-token distributions) yields $r_H \approx 0$, thus the sequence probability is enhanced after normalization. Therefore, the entropy-based reward $r_H$ encourages the model to generate answers whose token-level distributions are sharply concentrated, effectively boosting its "self-confidence" under the prior policy.

**Optimal Policy of the Reward Function $r_{\text{Traj}}$.** For the trajectory-level entropy-based reward $r_{\text{Traj}}$, it instantiates our unified framework with token-level granularity $\mathcal{I} = \{1, 2, ..., |y|\}$, anchor distribution $q = \{\delta^t\}_{t=1}^{|y|}$, model distribution $\pi = \{\pi_\theta^t\}_{t=1}^{|y|}$, sign factor $\sigma = -1$, and transformation $\psi(z) = z$. For a given input $x$, the model's predictive distribution is evaluated at every token. With $\sigma = -1$, the closer the distribution is to the one-hot reference $\delta^t$ (i.e., the higher the model's confidence in each ground-truth token), the larger the reward $r_{\text{Traj}}(x, y)$. Hence, after one policy-update step, the optimal probability $\pi_\theta(y|x)$ increases for such high-confidence trajectories, and decreases otherwise. Thus, $r_{\text{Traj}}$ encourages the model to generate sequences that already enjoy high probability under the prior policy.

The trajectory-level reward is defined as:

$$r_{\text{Traj}}(x, y) = \frac{1}{|y|} \sum_{t=1}^{|y|} \log \pi_\theta(y_t \mid x, y_{<t}) = \frac{1}{|y|} \log \pi_\theta(y \mid x). \tag{34}$$

Within the KL-regularized RL framework, the one-step optimal policy becomes:

$$\pi_\theta(y|x) \propto \pi_{\text{ref}}(y|x) \exp\left( \frac{1}{\beta |y|} \log \pi_\theta(y \mid x) \right) = \pi_{\text{ref}}(y|x) \cdot \big[\pi_\theta(y \mid x)\big]^{\frac{1}{\beta |y|}}. \tag{35}$$

Consequently, whenever the model assigns a higher prior probability to a sequence $y$, the weighted product term is amplified, thereby increasing its normalized probability $\pi_\theta(y|x)$. Therefore, the trajectory-level entropy reward boosts the probability of sequences that are already likely under the current policy $\pi_\theta$.

**Optimal Policy of the Reward Function $r_{\text{Prob}}$.** For the probability-based reward function $r_{\text{Prob}}$, it instantiates our unified framework with token-level granularity $\mathcal{I} = \{1, 2, ..., |y|\}$, anchor distribution $q = \{\delta^t\}_{t=1}^{|y|}$, model distribution $\pi = \{\pi_\theta^t\}_{t=1}^{|y|}$, sign factor $\sigma = -1$, and transformation $\psi(z) = \exp(|\mathcal{I}| \cdot z)$. For a given input $x$, the model's predictive distribution is evaluated at every token. With $\sigma = -1$, the closer the distribution is to the one-hot reference $\delta^t$ (i.e., the higher the model's confidence in each ground-truth token), the larger the reward $r_{\text{Prob}}(x, y)$ will be. Hence, after one policy-update step, the optimal probability $\pi_\theta(y|x)$ increases for such high-confidence trajectories, and decreases otherwise. Thus, $r_{\text{Prob}}$ encourages the model to generate sequences that already enjoy high probability under the prior policy.

The probability-based reward is defined as:

$$r_{\text{Prob}}(x, y) = \prod_{t=1}^{|y|} \pi_\theta(y_t \mid x, y_{<t}) = \pi_\theta(y \mid x). \tag{36}$$

Table 6: Default hyperparameters for training.

| Advantage Estimator | Training Temperature | Global Batch Size | Mini Batch Size | Rollout Number | Regularization | Max Prompt Length | Max Response Length | Learning Rate | Epoch |
|---|---|---|---|---|---|---|---|---|---|
| GRPO | 1.0 | 64 | 64 | 8 | w/o KL/Entropy | 1024 | 7168 | 1e-6 | 1 |

Within the KL-regularized RL framework, the one-step optimal policy becomes:

$$\pi_\theta(y|x) \propto \pi_{\text{ref}}(y|x) \exp\left(\frac{1}{\beta}\pi_\theta(y \mid x)\right). \tag{37}$$

Consequently, whenever the model assigns a high joint probability to a sequence $y$, the exponential weight is amplified, thereby increasing its normalized probability $\pi_\theta(y|x)$. The probability-product reward thus directly reinforces sequences that are already likely under the current policy, enhancing the model's preference for "high-likelihood" trajectories.

**Optimal Policy of the Reward Function $r_{\text{EMPO}}$.** For the answer-space probability-distribution reward $r_{\text{EMPO}}$ employed by the EMPO algorithm, it instantiates our unified framework with answer-level granularity $\mathcal{I} = \{\mathcal{A}\}$, anchor distribution $q = \delta^A$, model distribution $\pi = \pi_\theta^A$, $\sigma = -1$, and transformation $\psi(z) = \exp(z)$. For a given input $x$, multiple roll-outs are used to estimate the current policy's distribution over the answer space. With $\sigma = -1$, the closer this distribution is to the one-hot reference $\delta^A$ (i.e., the more probability mass is assigned to the extracted answer), the larger the reward $r_{\text{EMPO}}(x, y)$ will be. Hence, after one policy-update step, the optimal probability $\pi_\theta(y|x)$ increases for sequences that endorse the high-probability answer, while it decreases for all others. Maximizing $r_{\text{EMPO}}$ is therefore equivalent to driving the model to become more decisive at the answer level, thereby improving the consistency and determinism of the generated outputs.

Formally, the reward is defined as:

$$r_{\text{EMPO}}(x, y) = \pi_\theta(\text{ans}(y) \mid x), \quad \text{where } \pi_\theta(\text{ans}(y) \mid x) = \sum_{\text{ans}(y')=\text{ans}(y)} \pi_\theta(y' \mid x). \tag{38}$$

Within the KL-regularised RL framework, the one-step optimal policy is:

$$\pi_\theta(y \mid x) \propto \pi_{\text{ref}}(y \mid x) \exp\left(\frac{\pi_\theta(\text{ans}(y) \mid x)}{\beta}\right). \tag{39}$$

As evidenced by Equation (39), a single EMPO update re-weights each sequence by a factor of $\exp\left(\pi_\theta(\text{ans}(y) \mid x)/\beta\right)$ that depends on the current answer-level probability. After normalization, answers that already enjoy high probability under the prior policy gain additional mass, whereas low-probability answers suffer a decrease. Consequently, the optimal policy at each step systematically shifts the overall probability mass toward the high-probability region of the prior policy.

## B DETAILS FOR SECTION 4

### B.1 EXPERIMENTAL SETUP

**Implementation Details.** All experiments are conducted using the veRL framework (Sheng et al., 2025) with the GRPO algorithm. Unless stated otherwise, we utilize the default configuration outlined in Table 6. We implement five representative intrinsic rewards by customizing the `RewardManager` module of VeRL, following the reward formulations in Table 1 and Table 2:

- **Ensemble-Based Reward Estimators:** Majority Voting
- **Certainty-Based Reward Estimators:** Self-Certainty, Token-Level Entropy, Trajectory-Level Entropy, and Probability

**Training Dynamics Monitoring.** To monitor reward hacking and validate our theoretical predictions from Section 3, we implement specialized metrics to track the evolution of pseudo-rewards and their

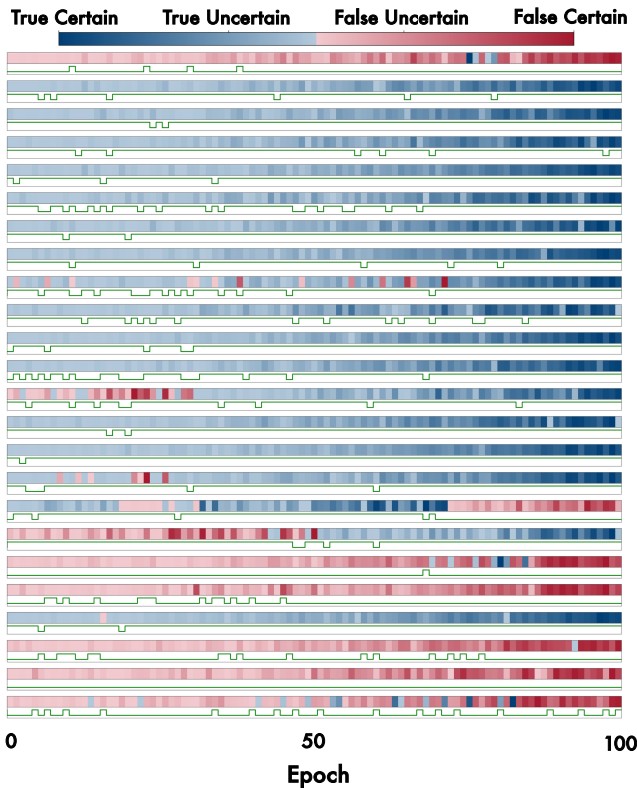

Figure 8: Examples of per-problem training dynamics from MATH-500.

alignment with ground truth. These metrics help identify when and how these intrinsic methods transition from beneficial sharpening to pathological collapse.

- **Ensemble-Based Metrics:** For methods using majority voting, we separately track the accuracy of the chosen label and the accuracy of the rewards it generates.
  - *Label Accuracy*: Prompt-level accuracy of majority-voted answers against ground truth, measuring ensemble quality
  - *Reward Accuracy*: Sample-level agreement between pseudo rewards and oracle rewards, capturing "lucky hits" (Zuo et al., 2025) where individual rewards align despite incorrect majority votes
  - *Ground Truth Reward*: Average oracle reward (supervised baseline), computed using actual correctness
  - *Majority Voting Reward*: Average pseudo reward from majority voting, the divergence from *Ground Truth Reward* indicates reward hacking

- **Certainty-Based Metrics:** For certainty-based methods, we measure the correlation between this proxy reward and the actual correctness.
  - *Label Accuracy*: Ground-truth accuracy of the highest-confidence response per prompt, testing whether maximum certainty implies correctness

These metrics collectively diagnose (1) pseudo-label quality degradation via *Label Accuracy* and (2) reward signal corruption via the gap between *Majority Voting Reward* and *Ground Truth Reward*. Mathematical definitions and implementation details are provided in Appendix B.2.

## B.2   Calculation of Training Dynamics

We provide mathematical definitions for the metrics used to monitor training dynamics. These metrics diagnose reward hacking and validate theoretical predictions about distribution sharpening.

### B.2.1 NOTATION

Let $\mathcal{D} = \{(x_i, a_i^*)\}_{i=1}^M$ denote the training dataset with $M$ prompts, where $x_i$ is the $i$-th prompt and $a_i^*$ is its ground-truth answer. For each prompt $x_i$, we generate $N$ rollout responses $\{y_{i,j}\}_{j=1}^N$ from the current policy $\pi_\theta$, where each response $y_{i,j}$ contains a trajectory and an extracted answer $\text{ans}(y_{i,j})$.

Define the following:

- $\mathbf{1}[\cdot]$: Indicator function returning 1 if the condition is true, 0 otherwise
- $\text{maj}(x_i)$: Majority-voted answer for prompt $x_i$, computed as $\arg\max_a \sum_{j=1}^N \mathbf{1}[\text{ans}(y_{i,j}) = a]$
- $r_{\text{gt}}(y_{i,j})$: Ground-truth reward for response $y_{i,j}$, equals $\mathbf{1}[\text{ans}(y_{i,j}) = a_i^*]$
- $r_{\text{mv}}(y_{i,j})$: Majority-voting pseudo-reward, equals $\mathbf{1}[\text{ans}(y_{i,j}) = \text{maj}(x_i)]$
- $r_{\text{cert}}(y_{i,j})$: Certainty-based reward (e.g., self-certainty, entropy) for response $y_{i,j}$

### B.2.2 ENSEMBLE-BASED METRICS

**Label Accuracy**    Measures the prompt-level accuracy of majority-voted answers:

$$\text{Label Accuracy} = \frac{1}{M} \sum_{i=1}^M \mathbf{1}[\text{maj}(x_i) = a_i^*]. \tag{40}$$

This metric ranges from 0 to 1, where 1 indicates perfect pseudo-label generation.

**Reward Accuracy**    Quantifies sample-level agreement between pseudo-rewards and oracle rewards:

$$\text{Reward Accuracy} = \frac{1}{M \cdot N} \sum_{i=1}^M \sum_{j=1}^N \mathbf{1}[r_{\text{mv}}(y_{i,j}) = r_{\text{gt}}(y_{i,j})]. \tag{41}$$

This captures "lucky hits" where individual rewards are correct even when the majority vote is wrong. For example, if the majority vote is incorrect but a minority response is correct, that response still receives the appropriate (zero) pseudo-reward.

**Ground Truth Reward**    Average oracle reward across all generated responses:

$$\text{Ground Truth Reward} = \frac{1}{M \cdot N} \sum_{i=1}^M \sum_{j=1}^N r_{\text{gt}}(y_{i,j}). \tag{42}$$

This represents the true quality of generated responses and serves as the supervised baseline.

**Majority Voting Reward**    Average pseudo-reward from majority voting:

$$\text{Majority Voting Reward} = \frac{1}{M \cdot N} \sum_{i=1}^M \sum_{j=1}^N r_{\text{mv}}(y_{i,j}). \tag{43}$$

The divergence between this metric and Ground Truth Reward indicates reward hacking: when the model learns to maximize pseudo-rewards at the expense of actual correctness.

### B.2.3 CERTAINTY-BASED METRICS

**Label Accuracy**    For certainty-based methods, we identify the highest-confidence response per prompt and measure its accuracy:

$$\text{Label Accuracy} = \frac{1}{M} \sum_{i=1}^M \mathbf{1}[\text{ans}(y_{i,j_i^*}) = a_i^*], \tag{44}$$

where $j_i^* = \arg\max_{j \in \{1,\dots,N\}} r_{\text{cert}}(y_{i,j})$ is the index of the highest-confidence response for prompt $x_i$.

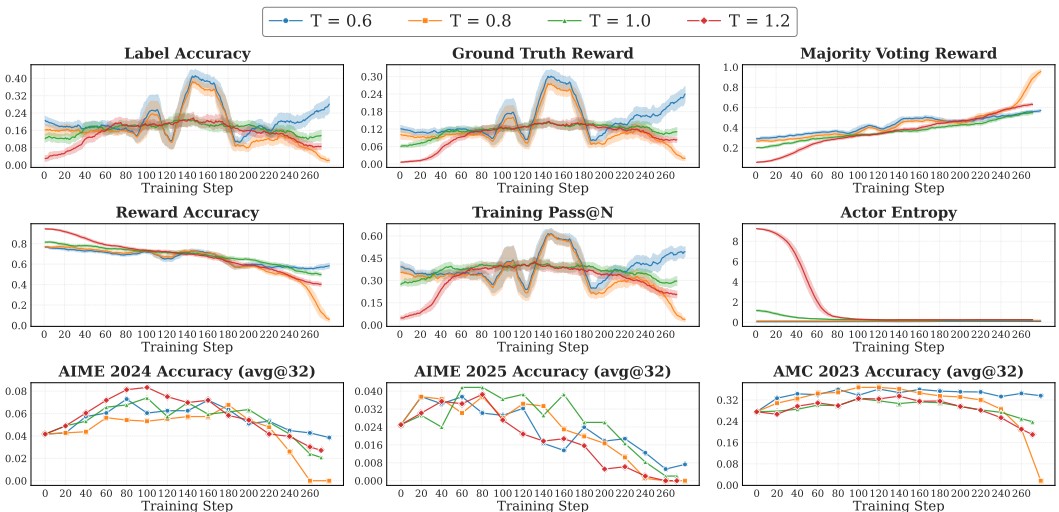

Figure 9: Effect of training temperature for Majority Voting method.

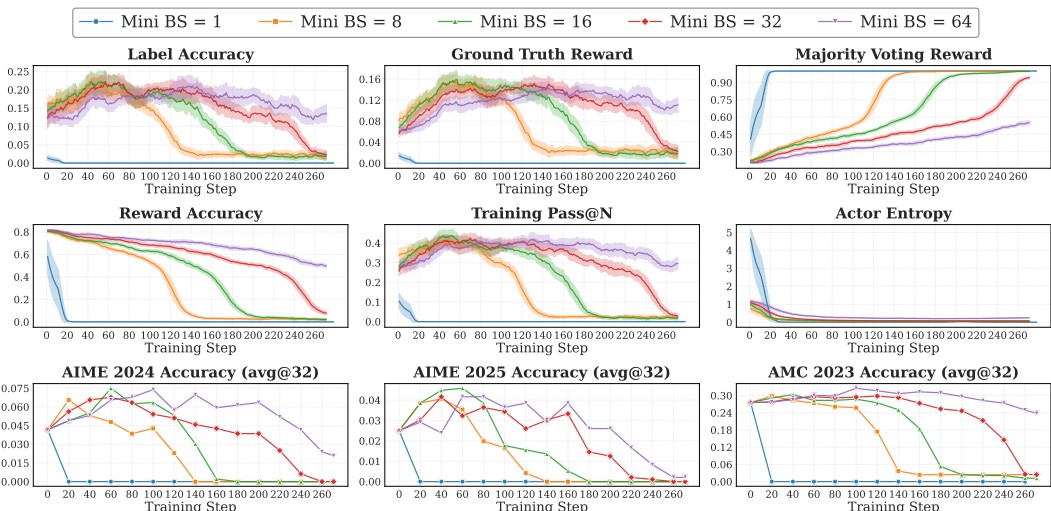

Figure 10: Effect of mini-batch size for Majority Voting method.

## B.3 HYPERPARAMETER TUNING

**Setup.** We study four hyperparameters, including training temperature, mini-batch size, KL divergence regularization, and rollout count, that directly influence performance of intrinsic reward training. We vary one parameter at a time while keeping others fixed at baseline values (see Table 6).

### B.3.1 MAJORITY VOTING

**Training Temperature.** Temperature directly controls exploration during rollout generation and affects the quality of pseudo-labels via voting diversity. From our convergence analysis in Theorem 1, lower temperature reduces the effective $\beta$ in the KL regularization term, accelerating convergence. As shown in Figure 9, low $T \in \{0.6, 0.8\}$ quickly sharpens logits, causing unstable *Label Accuracy*, consistent with premature convergence to an early majority that may be incorrect. Higher temperature ($T = 1.2$) maintains stability longer by preserving exploration, but the increased noise reduces peak performance. We find $T = 1.0$ provides optimal balance, showing steady early gains with delayed degradation.

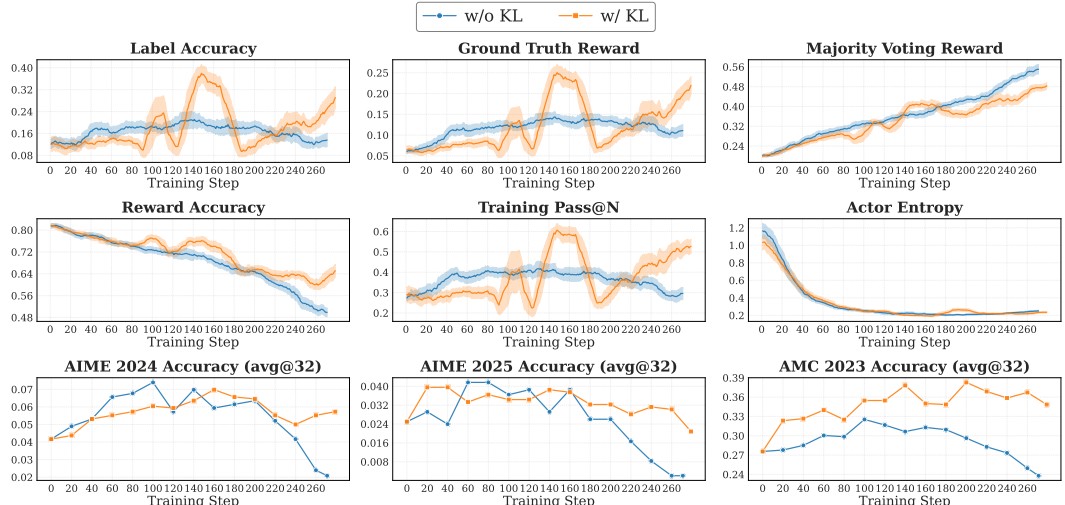

Figure 11: Effect of KL divergence regularization for Majority Voting method.

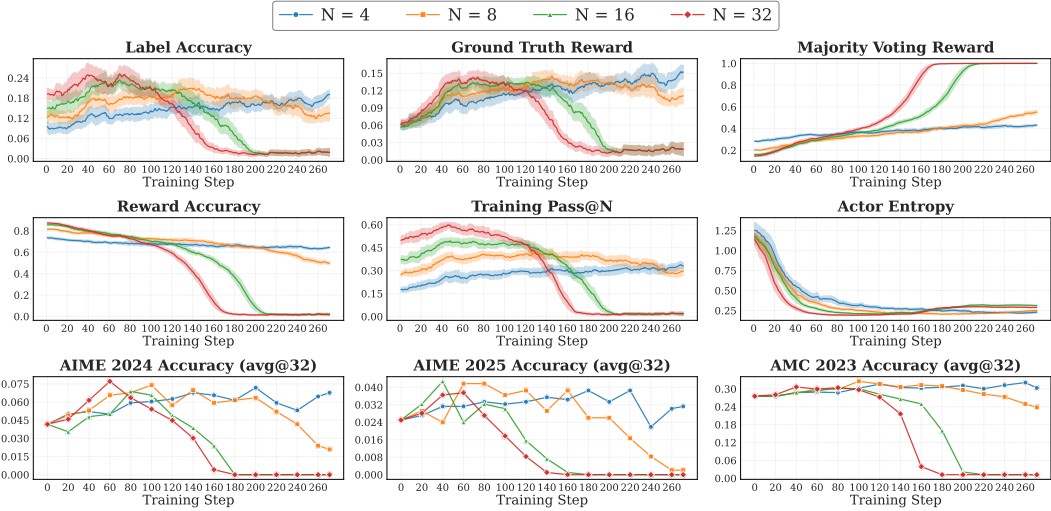

Figure 12: Effect of rollout number for Majority Voting method.

**Mini-batch Size.** This parameter controls the on-policy nature of updates, directly affecting the validity of our optimal policy assumptions. Our theoretical derivation in Equation (8) assumes rewards are computed under the current policy $\pi_\theta$. Small mini-batches violate this assumption through reward staleness: pseudo-rewards computed under $\pi_\theta$ become misaligned when applied to samples from $\pi_{\theta_{\text{old}}}$. As shown in Figure 10, mini-batch size 1 drives rapid collapse within 20 steps, while pure on-policy training (mini-batch $= 64$, matching global batch size) provides maximum stability. The intermediate sizes (16–32) show gradual improvement, confirming that maintaining policy-reward alignment is crucial for stable convergence.

**KL Regularization.** Our theoretical analysis suggests that KL regularization should slow convergence by increasing the effective $\beta$ parameter in Equation (5). However, empirical results in Figure 11 show that adding KL regularization ($\beta = 0.005$) yields only marginal benefits: small early gains but increased training variance and minimal delay in collapse ($\sim 40$ steps). This discrepancy arises because intrinsic rewards create competing optimization pressures, where the intrinsic signal drives sharpening while KL pulls toward the reference policy. Rather than smoothly balancing these forces, the optimization oscillates between them, increasing variance without providing durable stability. The marginal gains do not justify the additional memory overhead and training instability.

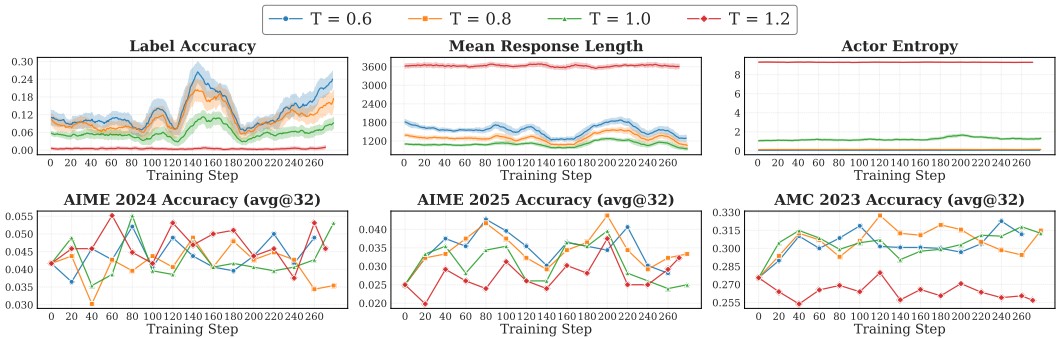

Figure 13: Effect of training temperature on Self-Certainty performance. Note that Point-Biserial Correlation is replaced with Mean Response Length due to Self-Certainty's scoring characteristics.

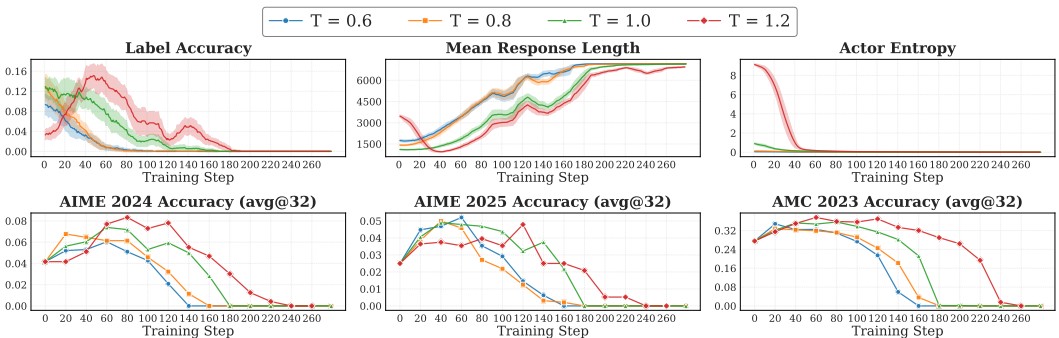

Figure 14: Effect of training temperature on Token-Level Entropy performance.

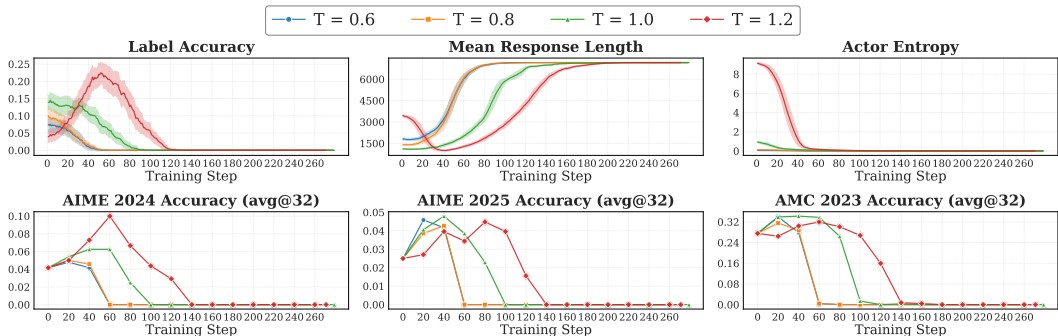

Figure 15: Effect of training temperature on Trajectory-Level Entropy performance.

**Number of Rollouts.** The rollout count $N$ affects both vote reliability and signal strength. While more rollouts improve statistical reliability of the majority vote, they also amplify the majority signal strength. From Equation (8), each update amplifies majority probability by factor $e^{1/\beta}$. With more rollouts, this majority becomes more confident, accelerating convergence. Figure 12 shows this effect: $N = 32$ collapses within 180 steps, $N = 16$ within 220 steps, while $N \leq 8$ remains stable over the full epoch. Although $N = 4$ shows competitive performance in some metrics, we recommend $N = 8$ as it provides better statistical reliability for the voting mechanism while maintaining reasonable convergence control. The slight performance difference suggests that for this specific experimental setup, the trade-off between reliability and stability favors slightly smaller $N$, but $N = 8$ offers more robust behavior across diverse problem types.

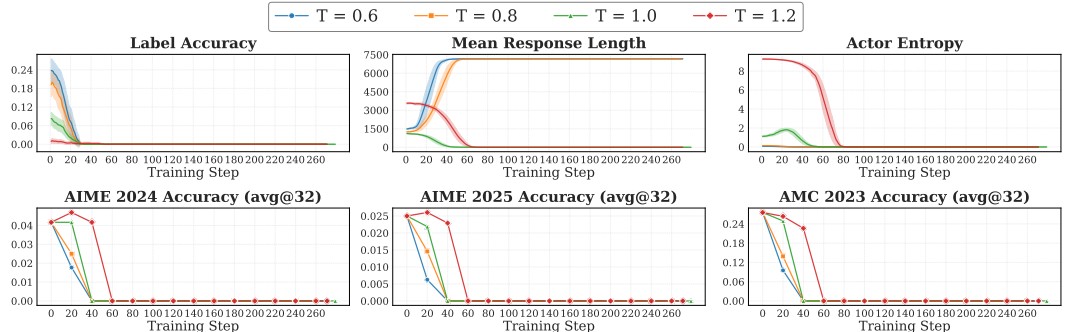

Figure 16: Effect of training temperature on Probability-based certainty performance.

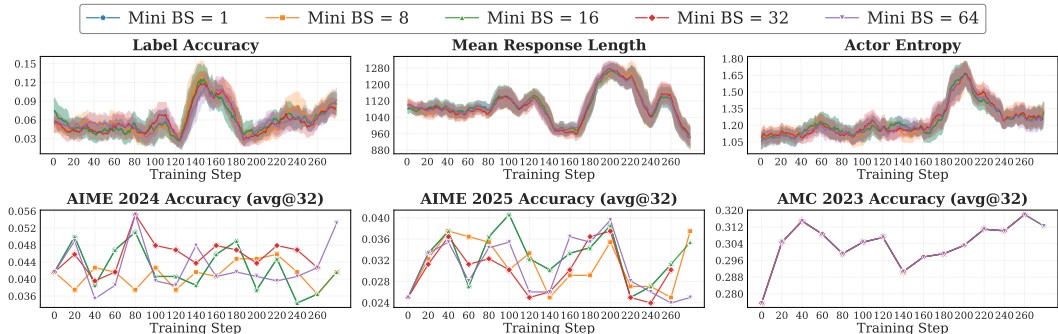

Figure 17: Effect of mini-batch size on Self-Certainty performance.

### B.3.2 CERTAINTY-BASED METHODS

**Training Temperature.** Temperature effects on certainty-based methods reveal distinct behavioral patterns compared to ensemble-based approaches. Unlike Majority Voting, certainty-based methods generally benefit from higher exploration temperatures, with notable method-specific variations in optimal configurations and convergence characteristics.

Results in Figures 14 to 16 demonstrate that higher temperature ($T = 1.2$) significantly delays model collapse across Token-Level Entropy, Trajectory-Level Entropy, and Probability methods. Higher temperatures initially maintain elevated **Actor Entropy**, facilitating extended exploration phases with gradual improvements across validation benchmarks. Importantly, these methods also exhibit relatively higher **Point-Biserial Correlation** values at $T = 1.2$, indicating stronger alignment between certainty estimates and actual correctness—a crucial property for effective uncertainty-based reward assignment.

However, Figure 13 reveals that Self-Certainty exhibits contrasting behavior. Higher temperature ($T = 1.2$) leads to excessive exploration without convergence, maintaining persistently high **Actor Entropy** while achieving lower validation scores and **Label Accuracy**. The moderate temperature $T = 1.0$ provides more stable and superior performance for Self-Certainty. This divergence suggests that while different certainty-based methods converge toward similar sharp distributions, they exhibit distinct convergence rates requiring method-specific temperature tuning. Among all certainty-based approaches, Token-Level and Trajectory-Level Entropy methods demonstrate the greatest benefits from higher temperature exploration, likely due to their more robust entropy-based uncertainty estimation mechanisms.

**Mini-Batch Size.** Mini-batch size effects on certainty-based methods largely parallel those observed in Majority Voting, confirming that on-policy ratio critically affects training stability regardless of the underlying reward computation mechanism. However, method-specific sensitivities reveal important distinctions in robustness to off-policy learning.

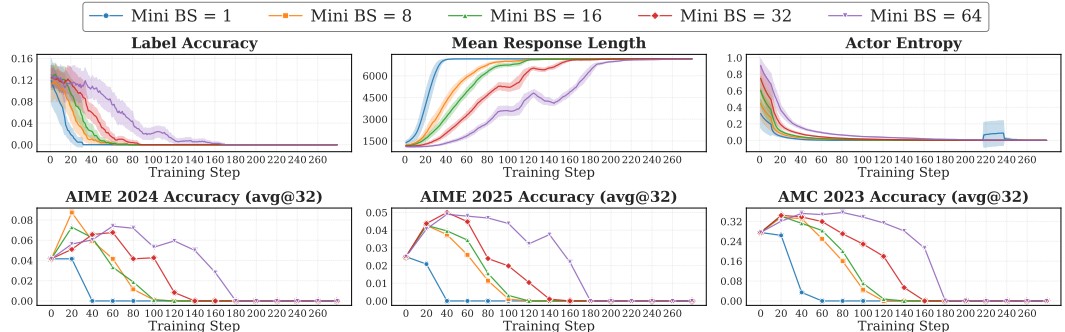

Figure 18: Effect of mini-batch size on Token-Level Entropy performance.

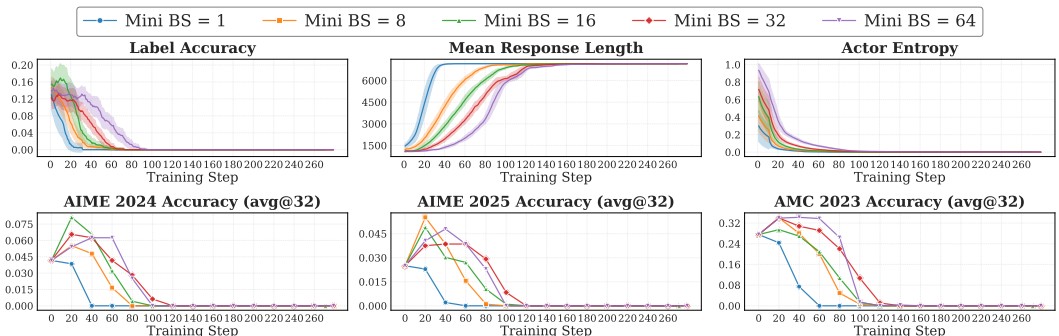

Figure 19: Effect of mini-batch size on Trajectory-Level Entropy performance.

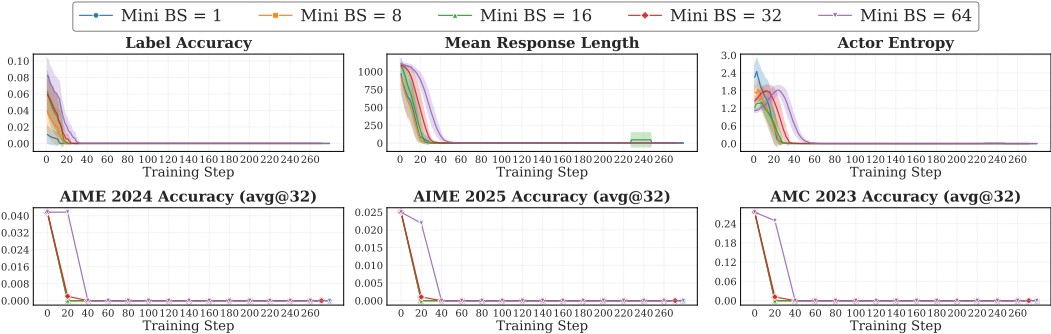

Figure 20: Effect of mini-batch size on Probability performance.

Figures 18 to 20 consistently demonstrate that larger mini-batch sizes prevent premature model collapse across Token-Level Entropy, Trajectory-Level Entropy, and Probability methods. This pattern mirrors Majority Voting behavior, where pure on-policy training (mini-batch size = 64) maintains optimal coupling between samples and their corresponding certainty-based rewards. The underlying mechanism remains consistent: certainty estimates computed from current policy states become unreliable when applied to samples generated from earlier policy iterations.

Notably, Self-Certainty exhibits exceptional robustness to mini-batch size variations, as shown in Figure 17. This method demonstrates minimal sensitivity to on-policy ratio changes, suggesting that KL divergence-based certainty computation may be inherently more stable across different temporal policy alignments. This robustness likely stems from Self-Certainty's reliance on logit distribution comparisons rather than explicit probability estimates, making it less susceptible to the temporal inconsistencies that destabilize other certainty-based approaches. Among the certainty-based methods, Self-Certainty thus offers superior stability but at the cost of lower overall performance improvements.

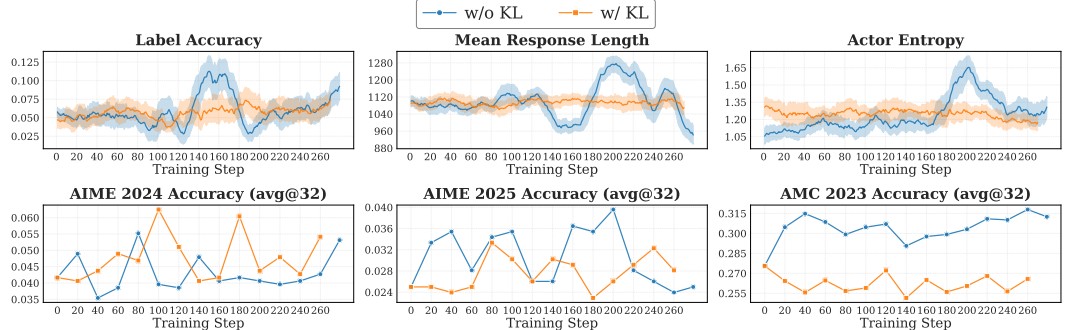

Figure 21: Effect of KL divergence regularization on Self-Certainty performance.

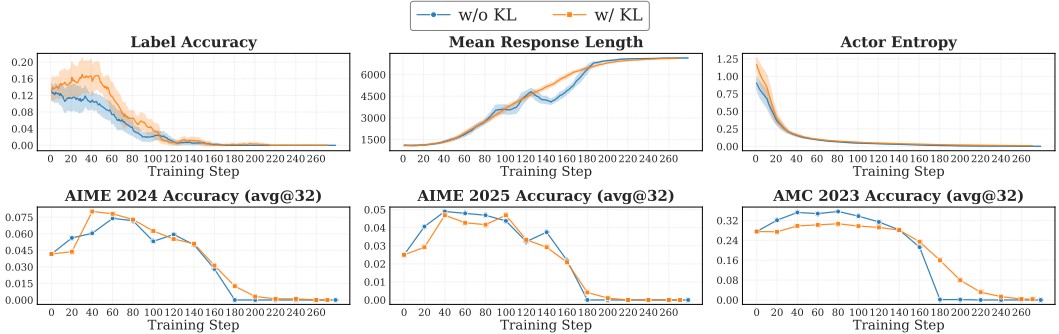

Figure 22: Effect of KL divergence regularization on Token-Level Entropy performance.

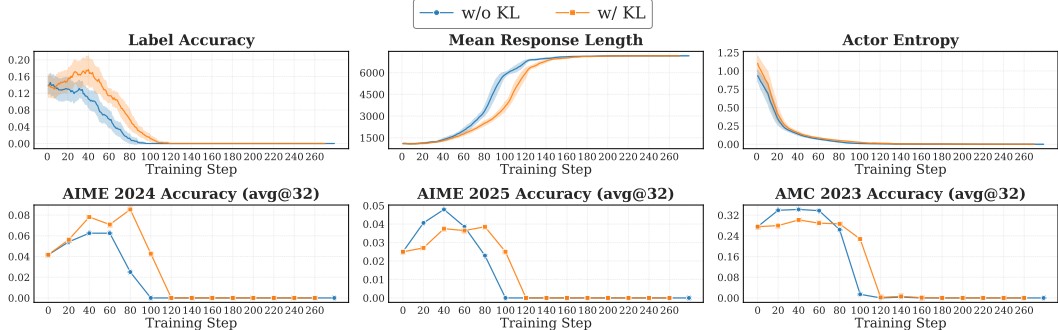

Figure 23: Effect of KL divergence regularization on Trajectory-Level Entropy performance.

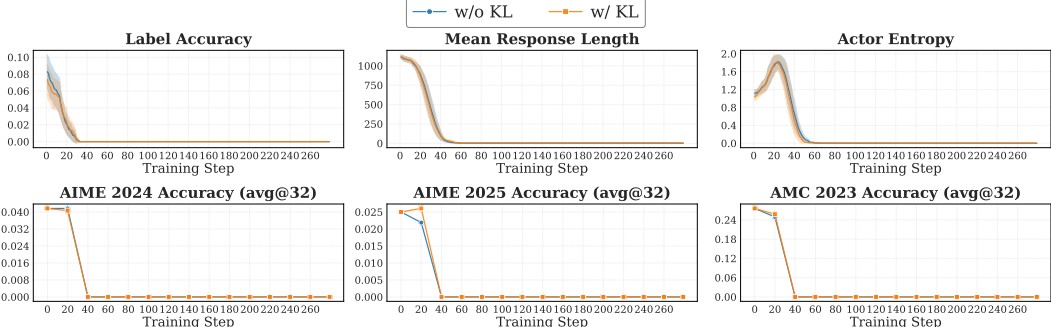

Figure 24: Effect of KL divergence regularization on Probability performance.

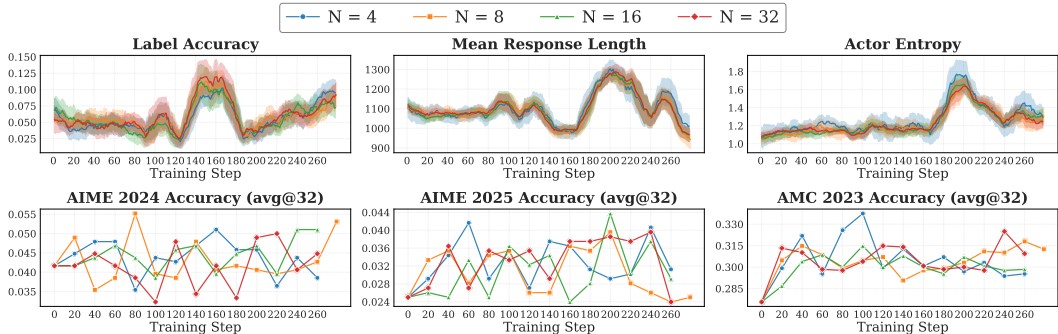

Figure 25: Effect of rollout number on Self-Certainty performance.

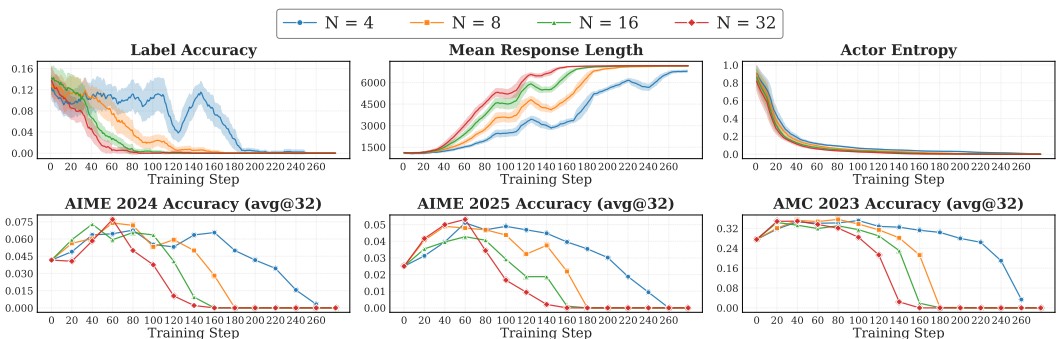

Figure 26: Effect of rollout number on Token-Level Entropy performance.

**KL Divergence Regularization.** KL regularization effects on certainty-based methods mirror the limited impact observed in Majority Voting, confirming that this regularization technique fails to address the fundamental instabilities inherent in training. However, subtle differences in method responses provide insights into the interaction between regularization and different uncertainty estimation approaches.

Results across all certainty-based methods (Figures 21 to 24) show minimal impact on both training dynamics and downstream performance. KL regularization neither prevents eventual model collapse (except for Self-Certainty) nor significantly improves validation scores, consistent with our findings for Majority Voting. The underlying issue persists: regularization techniques designed for fixed reward signals cannot effectively stabilize systems where rewards themselves evolve with policy changes.

Interestingly, Token-Level and Trajectory-Level Entropy methods exhibit slightly more pronounced benefits from KL regularization, as evidenced by modest improvements in **Label Accuracy** curves. While these improvements remain insufficient to prevent collapse, they suggest that entropy-based certainty estimation may have marginally better compatibility with KL-based stabilization approaches. This observation aligns with the superior temperature robustness of these methods, indicating that entropy-based uncertainty measures may be inherently more amenable to regularization techniques than probability-based or KL-based certainty estimates.

**Number of Rollouts.** Rollout count effects reveal consistent patterns across most certainty-based methods, with one notable exception that highlights fundamental differences in underlying reward computation mechanisms. These findings provide crucial insights into the sample size requirements for reliable uncertainty estimation.

Figures 26 to 28 demonstrate behavior parallel to Majority Voting: larger rollout counts ($N \geq$ 16) accelerate model convergence and premature collapse, as evidenced by rapid degradation in validation benchmarks and **Label Accuracy**. This pattern suggests that the self-reinforcing dynamics observed in ensemble voting also manifest in certainty-based reward assignment, where higher sample

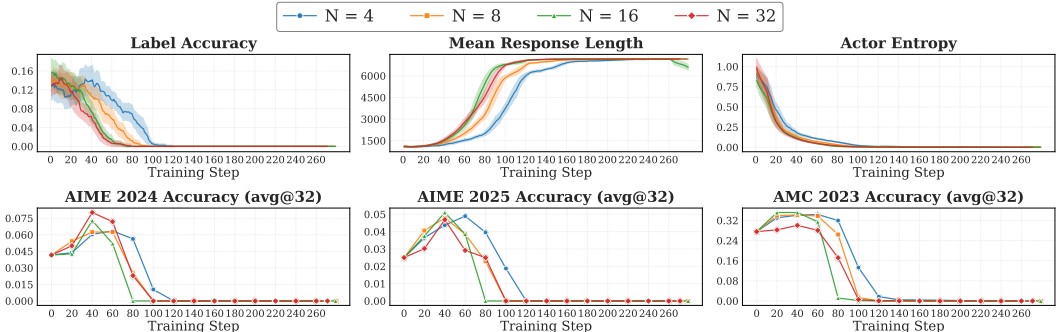

Figure 27: Effect of rollout number on Trajectory-Level Entropy performance.

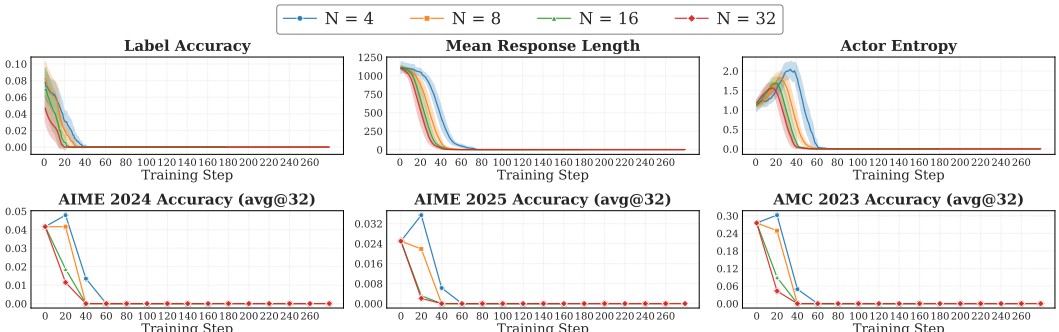

Figure 28: Effect of rollout number on Probability performance.

sizes amplify confidence in potentially incorrect assessments, leading to faster convergence toward suboptimal solutions.

However, Self-Certainty exhibits markedly different behavior, as shown in Figure 25. This method demonstrates remarkable stability across all rollout configurations, maintaining consistent performance without collapse or significant improvement. This unique characteristic stems from Self-Certainty's reliance on KL divergence between uniform and logit distribution. This fundamental difference in reward computation makes Self-Certainty inherently more robust to sample size variations, though at the cost of limited performance improvements throughout training.

## B.4 DIFFERENT METHODS, DIFFERENT FAILURES

We've shown that intrinsic rewards eventually collapse regardless of hyperparameter tuning. But do all methods fail the same way? We find that not all failures are equal. In this section, we reveal three distinct failure patterns, each exposing different weaknesses in how rewards reinforce confidence.

**Setup.** We compare five intrinsic rewards on Qwen3-1.7B-Base trained on DAPO-17k, each with separately tuned hyperparameters (Appendix B.3). For ensemble-based rewards we use the Majority Voting estimator, and for certainty-based we test Self-Certainty, Trajectory-Level Entropy, Token-Level Entropy, and Probability. We use the formulas in Tables 1 and 2 and evaluate on three benchmarks, tracking *Label Accuracy* (whether pseudo-label matches ground truth label), *Actor Entropy* and *Mean Response Length*. The calculation is detailed in Appendix B.1.

**Results.** Figure 29 shows that different methods lead to different failure modes:

- **Gradual degradation:** Self-Certainty and Majority Voting degrade most slowly, maintaining higher validation performance and higher *Label Accuracy* without collapsing within one epoch. Self-Certainty sharpens against a uniform distribution at each position (Table 1), making it less aggressive than direct probability maximization. Majority Voting operates at the answer level rather than the token level (Table 2), avoiding token-level artifacts.

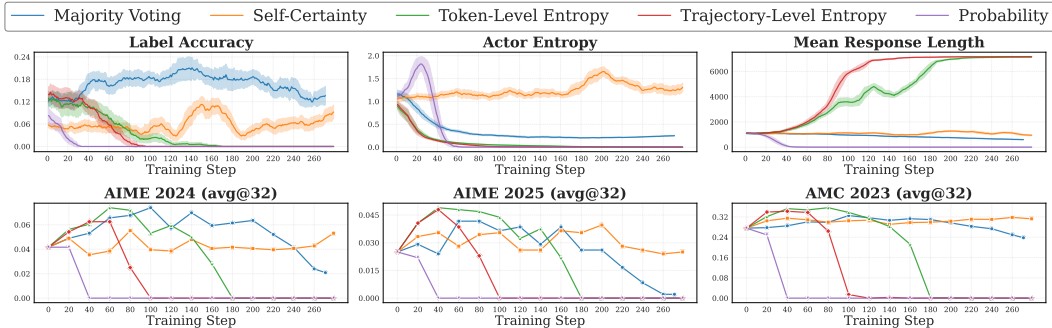

Figure 29: Five intrinsic reward methods exhibit distinct failure patterns. Self-Certainty and Majority Voting degrade gradually while maintaining label accuracy. Probability collapses toward brevity and entropy-based methods drive entropy down through repetition rather than correctness.

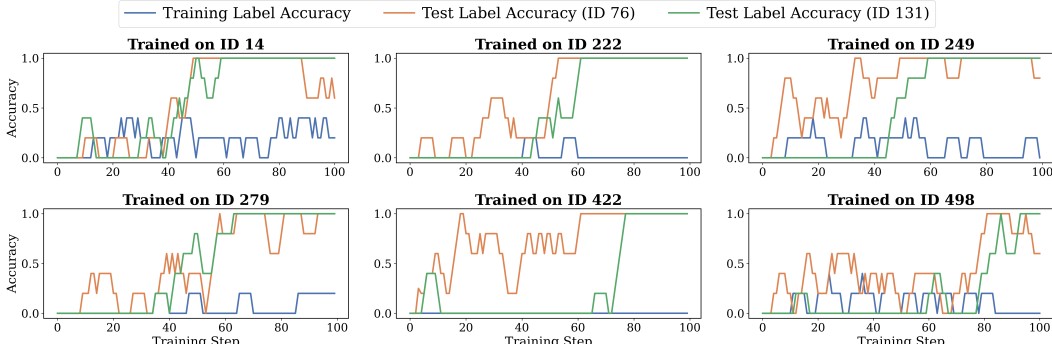

Figure 30: *Training Label Accuracy* (blue) on six MATH500 problems and *Test Label Accuracy* on two OOD problems: ID 76 (orange) and ID 131 (green).

- **Length collapse:** Probability rewards brevity because it multiplies token probabilities, which naturally favors shorter sequences. The model becomes confident (lower *Actor Entropy*) but produces overly brief answers (shorter *Mean Response Length*), creating a distinct reward hacking pattern focused on length rather than content quality. Length normalization like using geometric mean or average log-probability would likely mitigate this bias.

- **Repetition collapse:** Both entropy-based methods (Token-Level Entropy, Trajectory-Level Entropy) average entropy across tokens, which is minimized not only by confident predictions but also by repeating high-probability tokens. Unlike probability multiplication (which rewards brevity), averaging across sequences encourages the model to pad sequences with repetitive text.

These patterns reveal that while all methods sharpen distributions in theory, their practical failures diverge substantially.

## B.5 OUT-OF-DISTRIBUTION CROSS-PROBLEM GENERALIZATION

To investigate the generalization of intrinsic URLVR, we design a targeted experiment focused on out-of-distribution (OOD) cross-problem generalization.

**Setup.** We train Qwen3-1.7B-Base on 6 problems from MATH500 using the same setting as in Section 4.2, representing a constrained one-shot RL scenario. For each problem, the highest-reward sample is mostly wrong, and the model is trained on it individually for 100 epochs with batch size 1 and 8 rollouts for baseline estimation. For evaluation, we selected two unseen problems (ID 76 and ID 131), ensuring no overlap with the training set. We track *Training Label Accuracy* (the correctness of the highest-reward sample) and *Test Label Accuracy* (the correctness of greedy decoding test problems). All results are smoothed using a moving average with a rolling window of 5.

**Results.** Figure 30 shows that all training problems have a low *Train Label Accuracy* but for unseen test problems, they successfully turn from wrong to correct, with *Test Label Accuracy* increasing steadily from 0 to 1. This indicates that even though all training problems have wrong initial answers

and intrinsic URLVR amplifying their failures, the sharpening can still generalize to OOD problems. As long as the confidence aligns well with correctness on these unseen problems, sharpening still works.

## C  OTHER EXPERIMENTAL DETAILS

### C.1  INCORRECT MAJORITY VOTES STILL IMPROVE REASONING

Following Section 5.1, we now consider an extreme case where the initial majority votes in training are incorrect on a non-trivial proportion of examples. We want to see at this extreme case, shouldn't the small subset also collapse? Surprisingly, we observe that even when training amplifies errors on this incorrect subset, test-time training can still safely yield gains on OOD benchmarks, aligned with our cross-problem generalization findings in Appendix B.5.

**Setup.** We first perform offline filtering on DAPO-17k by sampling 64 responses per prompt from the base model and computing maj@64. Note that during training, we actually use maj@8 with temperature 1.0, which introduces some randomness. To ensure that the majority vote is incorrect in a non-negligible portion, we deliberately filter offline with maj@64 and control for higher majority ratios (>40%). We then trained on 32 filtered samples using the same setting as DAPO-32. We monitor *Label Accuracy* (measure whether maj@8 during training match ground truth label), *Majority Voting Reward* and the actual performance on validation benchmarks.

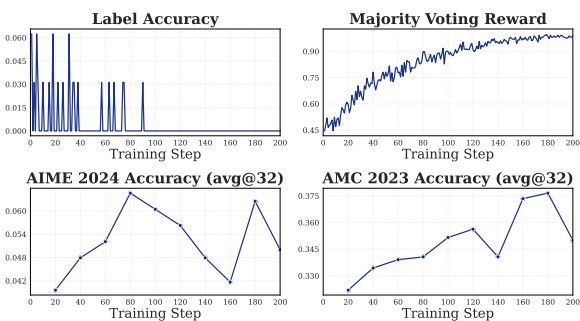

Figure 31: Training dynamics of extreme DAPO-32 setting, where almost all initial majority votes are incorrect.

**Results.** As shown in Figure 31, we observe non-zero accuracy at a few early steps, with only 1-2 maj@8 matching ground truth labels. After 100 steps, it consistently shows zero *Label Accuracy* with a convergent trend in *Majority Voting Reward*. Even when almost all 32 samples have incorrect initial majority votes, training still produces effective learning without catastrophic collapse. The gains observed on AIME24 and AMC23 demonstrate that small-scale training operates under fundamentally different dynamics than large-scale training. This validates that small subsets may avoid collapse through localized overfitting rather than systematic policy shift.

### C.2  PILOT STUDY OF DIFFERENT MODELS

We first observe that intrinsic URLVR behaves differently across model families and training stages.

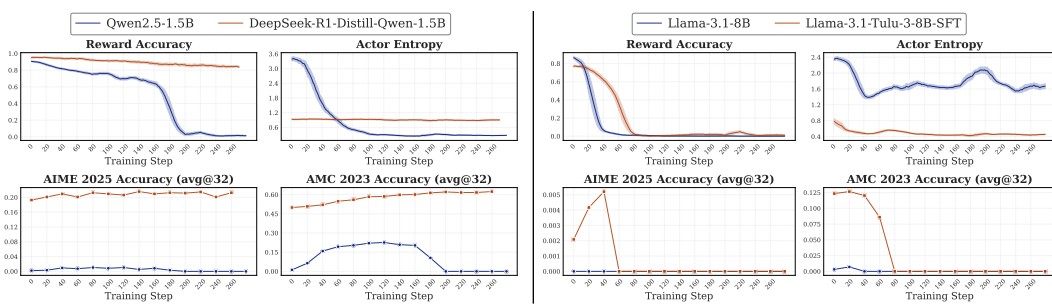

Figure 32: RL Training dynamics after different training stages in Qwen (left) and LLaMA (right) family. In Qwen, SFT enables stable training while base collapses by step 200. In LLaMA, both eventually collapse but SFT delays failure.

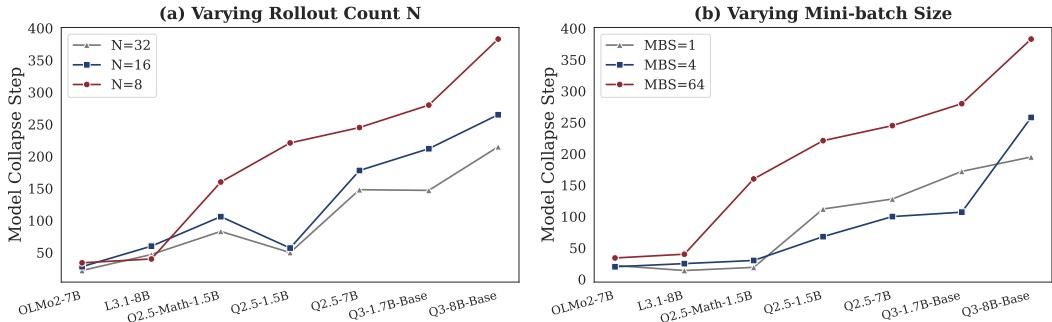

Figure 33: Consistency across different rollout counts (left) and mini-batch sizes (right). Aggressive hyperparameters accelerate collapse but preserve relative model rankings and predictive power.

Table 7: Computation cost comparison between Model Collapse Step and the gold standard (GT Gain) for assessing RL trainability across 7 models.

| Indicator | Computation Cost | Total Tokens | Requires GT |
|---|---|---|---|
| GT Gain | $7k \times 8 \times 17k \times 7$ (response $\times$ rollouts $\times$ problems $\times$ models) | 6.66B (baseline) | Yes |
| **Model Collapse Step** | $7k \times 8 \times 662 \times 32$ (response $\times$ rollouts $\times$ total steps $\times$ batch) | **1.19B** (**5.6$\times$ faster**) | **No** |

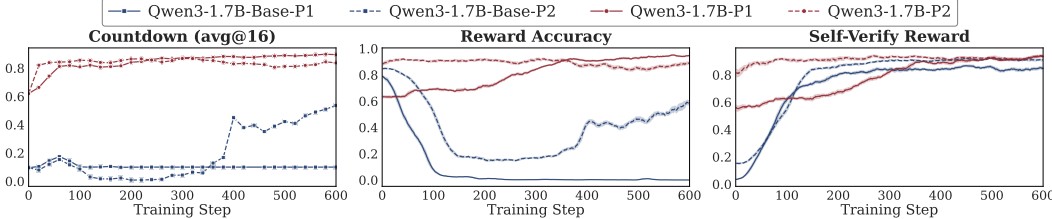

Figure 34: Prompt sensitivity across base and instruction-aligned models.

**Setup.** We compare four models: Qwen2.5-1.5B and DeepSeek-R1-Distill-Qwen-1.5B from the Qwen family; Llama-3.1-8B and Llama-3.1-Tulu-3-8B-SFT from the LLaMA family. This lets us separate architectural effects from training stage effects. All models are trained on DAPO-17k with majority voting reward. We track *Reward Accuracy*, *Actor Entropy*, and validation performance.

**Results.** The two families show strikingly different patterns (Figure 32). In Qwen family (left), the SFT variant maintains *Reward Accuracy* above 0.8 throughout training while the base model drops to near zero by step 200, despite starting with higher *Actor Entropy*. In LLaMA family (right), both variants eventually collapse but at different rates. The base model fails by step 40, while the SFT variant shows an initial performance rise before collapsing later. This architectural difference highlights that Qwen models appear fundamentally more stable, aligned with Shao et al. (2025a).

Zhang et al. (2025c) explains these dynamics through entropy minimization that early gains come from learning correct output formats, while later degradation stems from suppressing high-entropy transitional words that support reasoning. While this view explains how intrinsic rewards sharpen distributions, it cannot predict when sharpening helps versus hurts. Figure 32 reveals the missing piece that base models from both families start with higher *Actor Entropy* yet perform worse, with lower *Reward Accuracy* and faster collapse. If high entropy enables better reasoning, base models should outperform their SFT models. The opposite pattern suggests entropy is a consequence of the sharpening process, not its determinant and not enough to predict RL trainability and model prior.

## C.3 INSTRUCTION ALIGNMENT ENHANCES VERIFIER ROBUSTNESS

---

**Prompt 1: Adapted from RLSR (Simonds et al., 2025))**

Verify:
1. "{expr}" only contains numbers from {nums}.
2. Every number from {nums} is used in "{expr}" and is used only once.
3. "{expr}" is a valid arithmetic expression and not an equation.
4. "{expr}" equals {target}.
If **all** checks pass, return \boxed{True}; otherwise, return \boxed{False}.

---

**Prompt 2**

You are a strict mathematical verifier. Your task is to check whether the given expression correctly solves the arithmetic puzzle.

Do NOT attempt to find or generate a new expression yourself. You must only analyze and evaluate the provided expression "{expr}".

Verification steps:
1. If "{expr}" is missing, empty, or not a valid arithmetic expression (for example, if it contains words instead of numbers and operators), immediately output \boxed{False} and your task is over.
2. Check that "{expr}" only uses numbers from {nums}.
3. Each number from {nums} must appear exactly once in "{expr}".
4. The expression must contain only valid arithmetic operators: +, -, *, /, and parentheses.
5. Evaluate "{expr}" numerically. If the computed result equals {target} (within a tolerance of 1e-6), it passes this check.

If and only if all checks pass, output \boxed{True}. Otherwise, output \boxed{False}.

---

We find that the ability of instruction following is the key to the success of self-verification.

**Setup.** We compare Qwen3-1.7B-Base against its instruction-aligned version (Qwen3-1.7B) to see how model capability affects self-verification. We test both with two verification prompts (Prompt 1 and 2) to measure prompt sensitivity. We track the validation accuracy, *Reward Accuracy* and *Self-Verify Reward* (proxy reward used during training).

**Results.** From Figure 34, we find that instruction alignment helps with higher starting performance and robustness to prompt choice. The instruction model starts above 60% accuracy (surpassing the base model's final performance) and improves further to over 80%. More importantly, it succeeds with both prompts, while the base model only works with P2. The middle figure shows instruction models maintain stable *Reward Accuracy* with both prompts, while base models are highly sensitive. These results suggest self-verification can scale beyond test-time training (Section 5) when combined with instruction alignment, offering a robust path for scaling RL.

## C.4 IMPACT OF BACKBONE MODEL

We investigate how backbone models influence training stability and performance across three key dimensions: training stage, model size, and architectural generation. Our analysis employs 11 models from Qwen and Llama families (detailed configurations in Table 8), selected to provide systematic coverage of these factors. This selection is motivated by recent findings showing distinct architectural behaviors (Gandhi et al., 2025) and potential data contamination concerns (Wu et al., 2025b), making cross-architecture comparison essential. All models are trained on DAPO-17k using optimal hyperparameters from Section B.3 with Majority Voting as the representative intrinsic reward.

### C.4.1 HORIZONTAL ANALYSIS: TRAINING STAGE IMPACT

Training stage progression reveals distinct stability patterns between architectures. For the **Qwen family** (Figure 36), math-specialized and SFT models demonstrate superior stability, maintaining

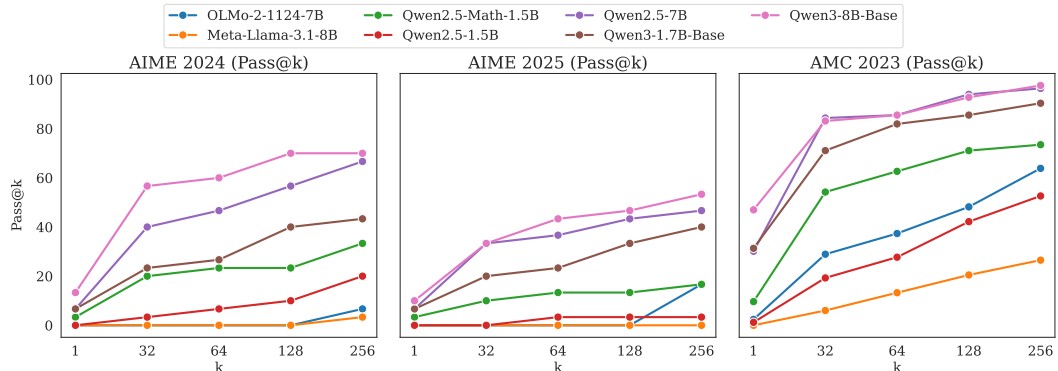

Figure 35: $Pass@k$ performance across different validation benchmarks and models, showing how trends vary as $k$ increases from 1 to 256.

Table 8: Model configurations for backbone experiments. Models are categorized by family, training stage, and size.

| Family | Model | Abbrev. | Stage | Size |
|--------|-------|---------|-------|------|
| Qwen | Qwen2.5-1.5B | Q2.5-1.5B | Base | 1.5B |
| | Qwen2.5-Math-1.5B | Q2.5-Math-1.5B | Math Base | 1.5B |
| | DeepSeek-R1-Distill-Qwen-1.5B | DS-R1-1.5B | SFT | 1.5B |
| | Qwen2.5-1.5B-Instruct | Q2.5-1.5B-Inst | Instruct | 1.5B |
| | Qwen3-1.7B-Base | Q3-1.7B | Base | 1.7B |
| | Qwen3-4B-Base | Q3-4B | Base | 4B |
| Llama | Meta-Llama-3.1-8B | L3.1-8B | Base | 8B |
| | OctoThinker-8B-Short-Base | Octo-8B | Math Base | 8B |
| | OctoThinker-3B-Short-Base | Octo-3B | Math Base | 3B |
| | Llama-3.1-Tulu-3-8B-SFT | L3.1-8B-Tulu-SFT | SFT | 8B |
| | Meta-Llama-3.1-8B-Instruct | L3.1-8B-Inst | Instruct | 8B |

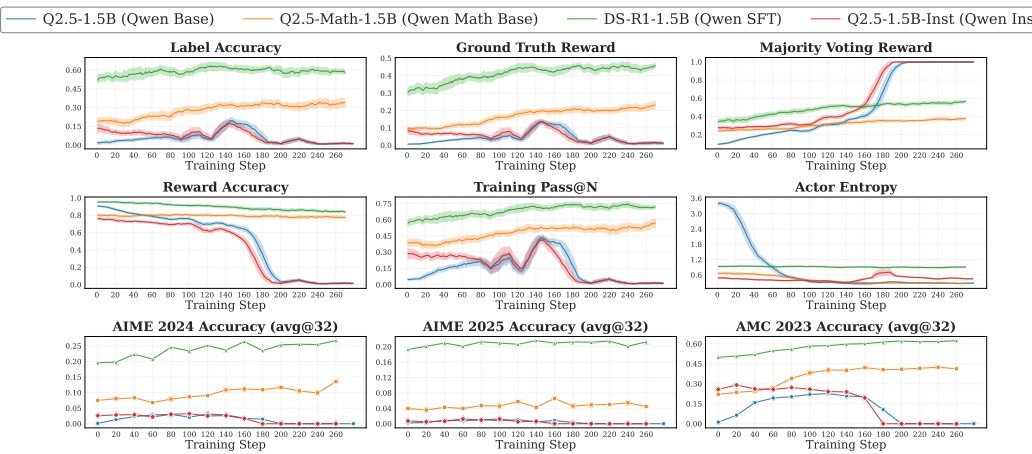

Figure 36: Training dynamics across different training stages in Qwen family models.

**Majority Voting Reward** within 0.3-0.6 while base and instruct variants reach saturation (1.0) by step 180. Math specialization and strong supervised fine-tuning (DS-R1-1.5B) create robust foundations for optimization compared to raw base models or non-math aligned instruct variants.

The **Llama family** exhibits contrasting behavior: all variants eventually succumb to reward hacking with different collapse timing, where base models fail earliest (step 40), followed by math-specialized,

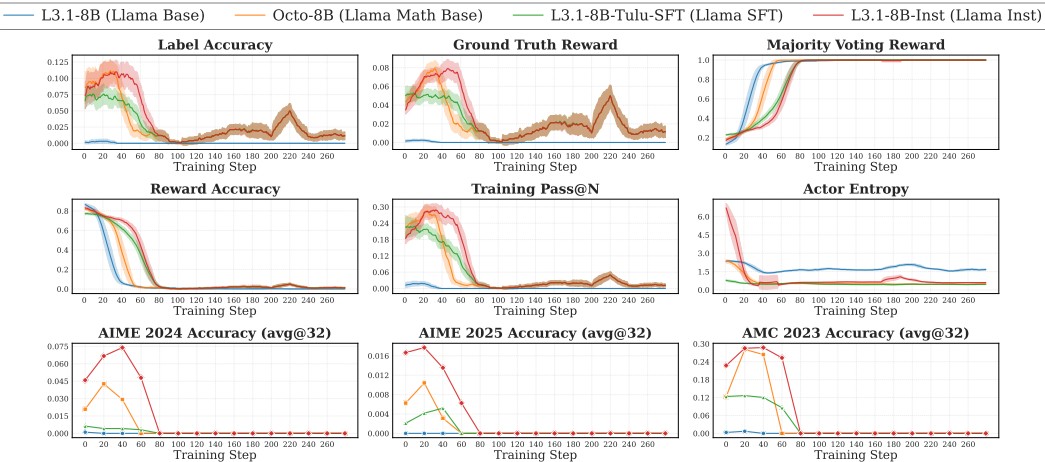

Figure 37: Training dynamics across different training stages in Llama family models.

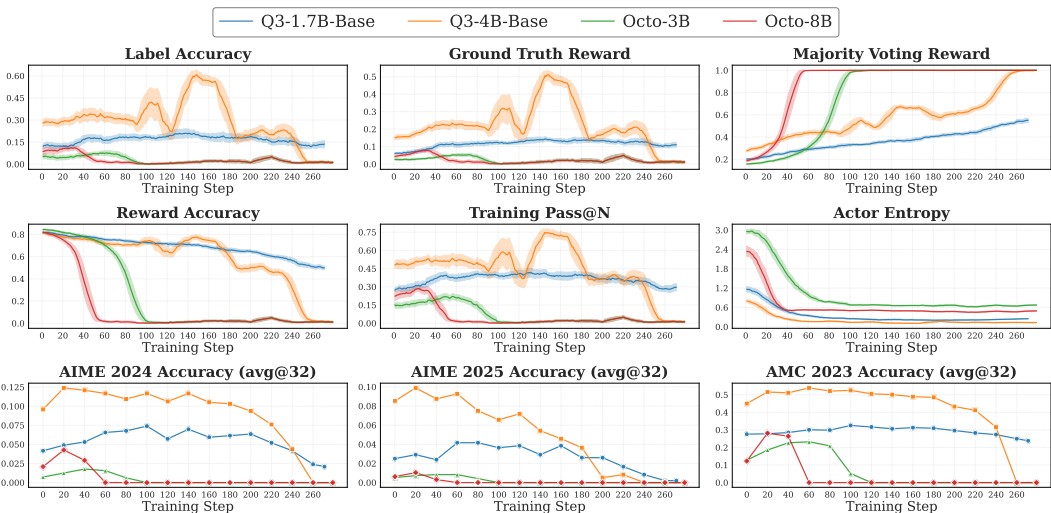

Figure 38: Effect of model size on stability across both Qwen and Llama families.

SFT, then instruct versions (detailed analysis in Figure 37). This architectural difference highlights Qwen's fundamental advantage in providing genuine stability.

### C.4.2 VERTICAL ANALYSIS: SCALE AND GENERATION EFFECTS

**Model size analysis** (Figure 38) reveals counterintuitive scaling effects: smaller models consistently outperform larger variants. Q3-1.7B maintains stability significantly longer than Q3-4B, while Octo-3B outlasts Octo-8B by about 40 steps. This suggests larger models' increased capacity amplifies sensitivity to noisy pseudo-rewards, accelerating convergence toward degenerate solutions and challenging conventional scaling assumptions.

**Architectural generation comparison** shows clear improvements in newer versions. Qwen3 models exhibit superior stability compared to Qwen2.5 counterparts, with Q3-1.7B-Base demonstrating more controlled **Majority Voting Reward** progression (comprehensive comparison in Figure 39). These improvements likely stem from better-calibrated uncertainty estimates and enhanced representation learning supporting more reliable pseudo-reward computation.

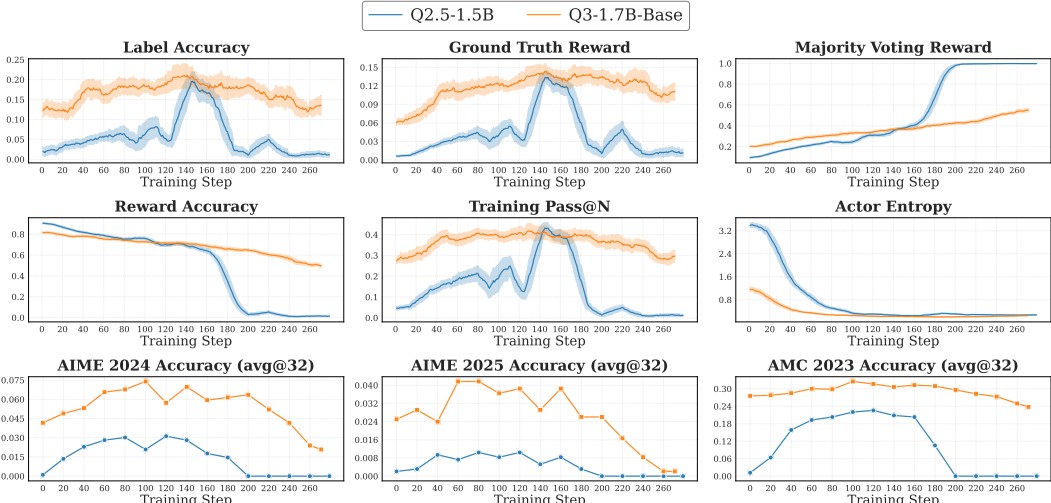

Figure 39: Comparison of Qwen 2.5 and Qwen 3 generations across comprehensive training metrics. Results reveal improved stability in the newer generation, with Qwen3 models demonstrating more gradual and controlled training dynamics compared to Qwen2.5 counterparts.

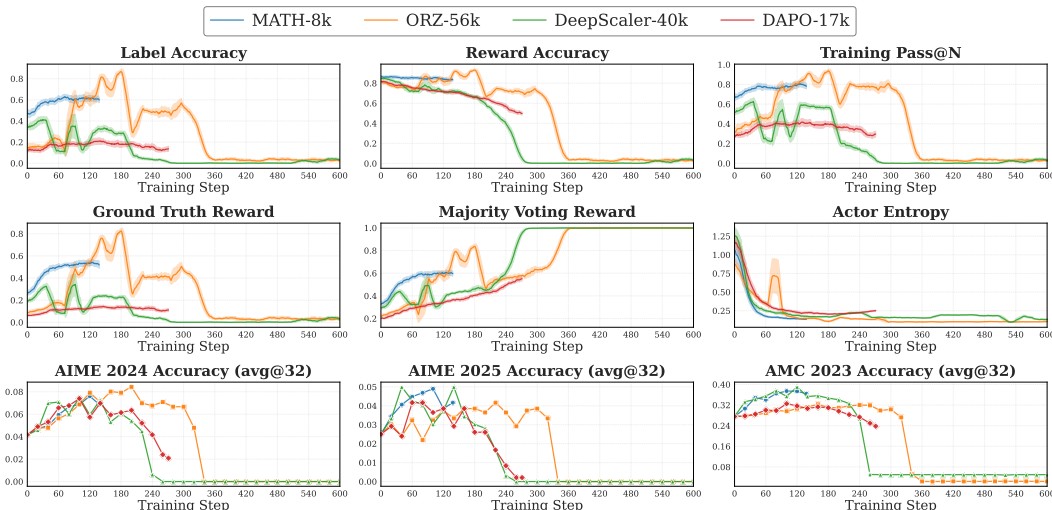

Figure 40: Comparison of different training data sources.

## C.5 IMPACT OF TRAINING DATASET

**Setup.** We investigate how different training dataset influence training stability and performance, focusing on math reasoning, utilizing MATH-8k (Hendrycks et al., 2021), DeepScaleR-40k (Luo et al., 2025), DAPO-17k (Yu et al., 2025) and ORZ-56k (Hu et al., 2025), all settings are trained on Qwen3-1.7B-Base with 1 epoch using optimal hyperparameters from Section B.3, and also evaluated on three validation benchmarks.

**Results.** We can see from Figure 40, much larger datasets (DeepScaler-40k and ORZ-56k) exhibits clear reward hacking trend, while smaller datasets settings are on its steady or rise stage, indicating that current intrinsic methods may see its short-sighted incremental improvements at the early stage, while extending it much larger training corpora, it inevitably encounter the reward hacking.

## C.6 DOES INTRINSIC REWARD METHODS TRULY IMPROVE CAPABILITIES?

Section 4 demonstrates that prevailing intrinsic reward approaches predominantly leverage uncertainty reduction as a mechanism for enhancing performance. This observation motivates a critical question: do such approaches truly enhance a model's capability, or do they merely improve the self-consistency of its outputs? Take the TTRL method as an example. TTRL explicitly models the self-consistency across $m$ model outputs through majority voting and leverages it as a supervisory signal. This design seems to suggest that TTRL may simply push the model toward the performance upper bound implied by the base model under majority voting. In other words, while TTRL might steadily improve the $pass@1$ metric, it would be unlikely to surpass the base model's $maj@m$ performance, thereby failing to deliver substantive gains beyond consistency alignment.

Table 9: Comparisons before and after TTRL. The results show that TTRL-trained models significantly surpass the base models' majority-vote performance in accuracy.

| Metric | Qwen2.5-Math-1.5B | Qwen2.5-Math-7B |
|---|---|---|
| *maj@2* | 28.09 | 33.23 |
| *maj@4* | 33.68 | 41.20 |
| *maj@8* | 37.23 | 45.73 |
| *maj@16* | 38.10 | 47.98 |
| *maj@32* | 38.43 | 49.19 |
| *maj@64* | 38.17 | 49.87 |
| *maj@128* | 37.86 | 50.14 |
| *maj@256* | 37.55 | 50.40 |
| *maj@512* | 37.41 | 50.60 |
| *maj@1024* | 37.30 | 50.79 |
| *avg@32* (w/ TTRL) | 48.90 | 68.10 |
| $\Delta$ | +11.60 | +17.31 |

However, our experiments reveal the opposite, as shown in Table 9. Specifically, we applied TTRL to Qwen2.5-Math-1.5B and Qwen2.5-Math-7B on AIME 2024 (30 samples) with a train batch size of 30 for 100 epochs, and directly compared the base models' $maj@1024$ with the $pass@1$ ($avg@32$) of the TTRL-trained models. Since the majority voting performance converges rapidly once the sample size reaches 32, $maj@1024$ can be reasonably regarded as a close approximation to $maj@\infty$. Strikingly, our results show that even the $pass@1$ metric of the TTRL-trained models significantly exceeds the $maj@\infty$ performance of the base models. This finding demonstrates that TTRL does far more than enforce internal self-consistency: it genuinely enhances the model's ability to generate accurate predictions. Put differently, TTRL enables the model to solve a broader range of problems than the base model, thereby delivering meaningful improvements in real-world performance.

