# OpenReview forum: "How Far Can Unsupervised RLVR Scale LLM Training?"
_ICLR.cc/2026/Conference — ICLR 2026 Poster_

### Official Review · Reviewer_t95z · 2025-10-26

**Soundness:** 2
**Presentation:** 2
**Contribution:** 2
**Rating:** 6
**Confidence:** 2

**Summary:**

The paper unifies several intrinsic reward modeling approach under the proposed unsupervised RLVR (URLVR) framework. Using majority voting as a prototypical approach, the paper observes that the trained model converges to a deterministic policy (Theorem 1). The paper hypothesizes that the performance of the learned policy depends on the confidence of the pre-RL model. URLVR approaches improve sampling efficiency but may not necessarily add any new capability. The paper tests this hypothesis via experiments on a a few language models.

**Strengths:**

- The paper unifies a variety of methods that use implicit reward modeling. This setup provides a framework for the community to understand the tradeoffs involved with these methods

- The analytical approach used in Theorem 1 clarifies the confidence vs performance trade-off.

- Experimental validation and analysis is mostly sound and supports the observation made in Theorem 1 and claim related to model collapse. I do have a question on the setup that I will note under weakness/question

**Weaknesses:**

- The use of Qwen model in the analysis could use more justification. The central claim in the paper is that  the success (or lack thereof) of URLVR methods depend on the "base/pre-RL's" model's confidence-correctness alignment. The empirical analysis is conducted on Qwen3-family of models. No justification is provided on why these models were chosen. I would be curious to understand whether there is something the reader can learn about how this choice is better than others that may be accessible to them.

- The writing in the paper could be improved
  - The paper uses math symbols (for example Table 1) without defining all of the symbols in text. While the appendix may have the info the reader needs, it would be useful if this information is available to the reader in caption/nearby text
  - The paper covers related papers but uses short-form etc for their names. This practice can be confusing to readers that are not as familiar with the field (like this reviewer) as the authors

**Questions:**

- Would it be possible to share any insight on why Qwen3 models may have the confidence-correctness alignment shown in the paper? Is there a way to show some language models family are more aligned than others in the sense defined in the paper?

- What is RLIF mentioned in the paper? There is another paper not cited in the draft that shows up when the reviewer did a literature search : https://arxiv.org/abs/2311.12996

- I did not see an LLM usage statement in the paper or appendix. Please include one as required by ICLR 2026 guidelines

---

> ### Author Response · Authors · 2025-11-21
>
> ## Response to W1:
>
> > W1: The use of Qwen model in the analysis could use more justification. The central claim in the paper is that the success (or lack thereof) of URLVR methods depend on the "base/pre-RL's" model's confidence-correctness alignment. The empirical analysis is conducted on Qwen3-family of models. No justification is provided on why these models were chosen. I would be curious to understand whether there is something the reader can learn about how this choice is better than others that may be accessible to them.
>
> Thank you for this important question about model selection. We appreciate the opportunity to clarify our choice and demonstrate its broader implications for readers.
>
> **Why Qwen3 specifically?**
>
> In fact, we selected Qwen3 as our base model only after fully completing the experiments detailed in Appendix B.4 across 11 models. **Therefore, our choice of Qwen3 models was deliberate.** Below, we present two key considerations:
>
> 1. **Avoiding known issues with alternatives:**
>    - Qwen2.5 series: Recent work [1] raised data contamination concerns, particularly on mathematical benchmarks like AIME
>    - Llama3 series: Our preliminary experiments (detailed in Appendix B.4) showed these models exhibit very early collapse (step 40) with minimal improvement, making it difficult to observe the full "rise and fall" dynamics that are central to our analysis
> 2. **Demonstrating clear dynamics:**
>    - Since our core empirical contribution (Section 4) is characterizing the rise-then-fall pattern of intrinsic reward methods, we needed models with sufficiently strong priors to show both phases distinctly. Qwen3's stability window (Figure 30) allows clear observation of both success and failure modes.
>
> **What Can Readers Learn About Their Own Model Choice?**
>
> This question goes straight to the core of what we hope to demonstrate to the community. Please see our **[Global Response 1]** for an analysis addressing this directly. In brief, our work provides the community with:
>
> 1. **Systematic comparison (Appendix B.4):** 11 models across families (Qwen2.5/Qwen3/Llama3.1/OctoThinker), training stages, and sizes, showing the qualitative pattern holds universally with vastly different timescales.
> 2. **Diagnostic tool (Section 5.2 + new validation):** We proposed that early dynamics serve as a "model-task prior indicator" and now quantitatively validate its accuracy against gold-standard supervised RL across 7 models from 3 families. **We find Model Collapse Step (when pseudo-label accuracy drops below 1%) strongly predicts RL scalability.**
>
> Our work provides practical guidelines for quickly assessing any model’s prior while also revealing a clear hierarchy across major model families. By running a brief intrinsic-reward trial (about 50 steps) and measuring the Model Collapse Step, we can estimate the potential performance uplift achievable through RL: a Model Collapse Step below 100 suggests limited gains, whereas a value above 200 indicates meaningful improvement potential. This diagnostic applies broadly across model families, including LLaMA, Mistral, Phi, and Qwen, and aligns with consistent differences we observe in their priors:
>
> - OLMo/Llama: Weak priors (collapse steps 34-40), minimal gains (<1%)
> - Qwen2.5: Moderate priors (collapse steps 160-245), moderate gains (4-7%)
> - Qwen3: Strong priors (collapse steps 280-383), substantial gains (7-17%)
>
> Rather than prescribing a specific model such as Qwen3, which we used only for practical reasons, our framework provides a general diagnostic that enables the community to evaluate the latent RL-scalability of their own models.
>
> ---
>
> ### References
>
> [1] Reasoning or Memorization? Unreliable Results of Reinforcement Learning Due to Data Contamination

---

> > ### Author Response · Authors · 2025-11-21
> >
> > ## Response to W2:
> >
> > > W2.1: The writing in the paper could be improved
> > > - The paper uses math symbols (for example Table 1) without defining all of the symbols in text. While the appendix may have the info the reader needs, it would be useful if this information is available to the reader in caption/nearby text
> >
> > Thank you for this important feedback. We acknowledge that the symbols in Equation (1) and Table 1 were not adequately defined in the main text, forcing readers to search the appendix.
> >
> > **In the revised manuscript, we have added explicit definitions of all symbols immediately after Equation (1) (marked in red),** following a logical order: starting with the input/output $(x,y)$, then aggregation granularity $\mathcal{I}$, model distribution $\pi_{\theta}^i$, anchor distribution $q^i$, cross-entropy $\mathbb{H}$, sign factor $\sigma$, and monotonic transformation $\psi$. We also added explanatory text after Table 1 clarifying how each method chooses its granularity and sign, along with a remark on the special notation for Majority Voting.
> >
> > We appreciate the reviewer pointing this out. The revised presentation makes the framework accessible without requiring navigation between sections.
> >
> > > W2.2: The writing in the paper could be improved
> > > - The paper covers related papers but uses short-form etc for their names. This practice can be confusing to readers that are not as familiar with the field (like this reviewer) as the authors
> >
> > Thank you for this valuable feedback. We acknowledge that our use of abbreviations was inconsistent and confusing, and we're sorry for that.
> >
> > In the original Related Work section, we introduced abbreviations for ensemble-based methods (TTRL, SRT, ETTRL, etc.) but failed to do so for certainty-based methods. In Table 1, we then referenced abbreviations (RLIF, EM-RL, RENT, RLSC) without proper introduction in the related work section, creating confusion.
> >
> > **In the revised manuscript,** we have corrected this by explicitly introducing all method abbreviations in Section 2 Related Work **(marked in red)**. Besides, we added full citation references in the "Method" column of Table 1 to link each abbreviation directly to its source paper, making it immediately clear which paper each abbreviation refers to.
> >
> > This ensures all abbreviations used in Table 1 are properly introduced before first use. We apologize for this oversight and appreciate the reviewer's attention to clarity for broader readership.

---

> ### Author Response · Authors · 2025-11-21
>
> ## Response to Q1:
>
> > Q1: Would it be possible to share any insight on why Qwen3 models may have the confidence-correctness alignment shown in the paper? Is there a way to show some language models family are more aligned than others in the sense defined in the paper?
>
> Thank you for this insightful follow-up question. This gets at the core mechanism we've been investigating.
>
> **Why Qwen3 Shows Stronger Alignment:**
>
> We believe Qwen3's superior confidence-correctness alignment (what we call stronger model-task prior) stems from two factors, which are also shown in their technical report [1]:
>
> 1. **Math-specialized pretraining:** Qwen3 models underwent extensive pretraining on mathematical corpora, creating domain-specific knowledge where the model's confidence genuinely reflects understanding rather than superficial pattern matching.
> 2. **Improved calibration:** Newer architectural generations (Qwen3 vs. Qwen2.5) demonstrate better-calibrated uncertainty estimates, as shown in Figure 31 (Appendix B.4), where Qwen3 models exhibit more controlled training dynamics compared to their predecessors.
>
> **Supporting Evidence from Recent Works:**
>
> This observation aligns with emerging research on model-task alignment. [2] shows that even spurious rewards yield strong gains on Qwen models (their Figure 1), suggesting these models possess robust priors that RL can leverage. More recently, [3] systematically investigates when models are easy to improve via RL, proposing that the degree to which model capabilities match task requirements is the critical factor. They find that models benefit from various RL techniques (noisy rewards, test-time RL, minimal training) primarily within their domains of expertise, where strong priors exist.
>
> This provides a principled explanation: Qwen3's math specialization creates strong task alignment on reasoning benchmarks, enabling the confidence-correctness correlation necessary for intrinsic reward success.
>
> **Demonstrating Which Model Families Are More Aligned:**
>
> Please see our **[Global Response 1]** for quantitative comparison across model families.
>
> Overall, we provide a quantitative method to measure alignment strength using the Model Collapse Step metric, which is the step reward accuracy drops below 1% during intrinsic URLVR training. Across 7 models from 3 families (OLMo, Llama, Qwen), we find that Model Collapse Step directly measures how long confidence-correctness alignment remains reliable, with strong correlation to true RL scalability:
>
> - OLMo and Llama families: Weak alignment (collapse at steps 34-40)
> - Qwen2.5 family: Moderate alignment (collapse at steps 160-245)
> - Qwen3 family: Strong alignment (collapse at steps 280-383)
>
> **Practical Implication:** Rather than requiring deep investigation into why certain architectures are better calibrated, readers can simply measure the Model Collapse Step on their accessible models (~50 training steps) to quantify alignment strength. This provides an empirical answer to "is my model family sufficiently aligned?" without needing to understand the underlying architectural differences.
>
> ## Response to Q2:
>
> > Q2: What is RLIF mentioned in the paper? There is another paper not cited in the draft that shows up when the reviewer did a literature search : https://arxiv.org/abs/2311.12996
>
> Thank you for pointing this out. We apologize for the confusion regarding RLIF.
>
> In our paper, **RLIF refers to the self-certainty method** from the paper "Learning to Reason without External Rewards". This confusion arose from our inconsistent abbreviation introduction. **In the revised manuscript,** we have:
>
> - Explicitly stated in Section 2: "Self-Certainty in RLIF", along with other methods, marked in red
> - Added the full citation reference in Table 1 next to "RLIF"
>
> ## Response to Q3:
>
> > Q3: I did not see an LLM usage statement in the paper or appendix. Please include one as required by ICLR 2026 guidelines
>
> We sincerely apologize for the confusion. We will include this section in the next version of our manuscript. Thank you very much for your helpful suggestion!
>
> ---
>
> ### References
>
> [1] Qwen3 Technical Report
>
> [2] Spurious Rewards: Rethinking Training Signals in RLVR
>
> [3] Mirage or Method? How Model-Task Alignment Induces Divergent RL Conclusions

---

> > ### Comment · Reviewer_t95z · 2025-11-27
> >
> > I thank the authors for their responses. I believe that the responses provide answers to my questions raised during review. Since there is an active discussion between authors & reviewers, I will hold off revising the Rating until after the end of the discussion period.

---

> > > ### Author Response · Authors · 2025-11-27
> > >
> > > Thank you for your thoughtful engagement and for acknowledging that our responses have addressed your questions. We truly appreciate you taking the time to consider the ongoing discussions.
> > >
> > > If any concerns from the active discussion raise questions for you, or if there are additional points you'd like us to clarify, we would be very grateful if you could share them. We are actively responding to all feedback and are happy to provide any additional clarifications that would be helpful for your evaluation.

---

### Official Review · Reviewer_um7o · 2025-10-28

**Soundness:** 1
**Presentation:** 3
**Contribution:** 2
**Rating:** 2
**Confidence:** 4

**Summary:**

This paper investigates the mechanisms underlying Unsupervised Reinforcement Learning with Verifiable Rewards (URLVR), which finetunes a language model through RL over intrinsic rewards (i.e., rewards derived based on the model itself without any external supervision). It first introduces a theoretical perspective that unifies the goal of ensemble-based and certainty-based methods. Then, the theoretical insights are demonstrated empirically. In particular, the paper shows that existing intrinsic rewards trade uncertainty for performance by sharpening the language model distribution, yet this comes at the risk of distribution collapse and reward hacking. Furthermore, it characterizes how such collapse manifests in different methods and shows that in some specific regimes it can be avoided.

**Strengths:**

1. The topic is timely, given the increase in popularity of methods using RL with intrinsic rewards for improving the capabilities of language models.

2. The empirical analysis sheds light on the difference between intrinsic rewards and their potential failure patterns.

3. I find the suggestion of using intrinsic reward dynamics in the initial time steps as an indicator for the potential success of URLVR, as opposed to using measures such as pass@k that require access to a verifier or labels, to be interesting. However, it is worth noting that the evaluation of this method is somewhat lacking. There is no qualitative guidance as to how one should use the intrinsic reward dynamics and there is no comparison to pass@k in terms of predictiveness or computational efficiency.

**Weaknesses:**

1. The unified reward framework in Section 3.1 is currently not well-defined and its usefulness is not substantiated in the paper. Specifically, how does the right hand side of Equation (1) depend on $y$? Should the cross-entropy term $h$ be $-q^i (y | x) \ln \pi_\theta^i (y | x)$? Moreover, the significance of such a unified perspective greatly depends on whether it allows characterizing similarities and differences between different instances. However, the unified framework is not really used in the paper after its definition, which raises the question of why it is necessary or helpful.

2. The theoretical analysis contains potentially incorrect claims.
    - The reward in majority voting depends on the current policy. It therefore changes during training. The closed form solution to the KL-regularized RL objective in Equation (3) holds for a fixed reward, and so it is not clear what it implies for a policy-dependent reward such as majority voting. In particular, without proof, there is no reason to believe that Equation (4) holds. Do the authors have a proof for this claim? Also, the majority reward is not formally defined: what does $maj (Y)$ stand for? How many rollouts are considered in this majority?
    - Theorem 1 is not rigorously stated. It is therefore difficult to evaluate what it means or whether it is sound. Specifically, what does "majority trajectories" refer to? What are "$k$ updates"? Does this refer to $k$ steps of policy gradient or rather $k$ iterations of solving perfectly the KL-regularized objective and each time updating the reference policy to be the current policy?

3. One of the main claims made in the paper is that the success of URLVR stems from sharpening the language model’s distribution on correct outputs, in case it already assigns such outputs a reasonable probability. However, as far as I am aware, this is already the existing conventional wisdom behind why methods such as majority voting as a reward can work (c.f. [1,2]).

4. Relation to prior work is often not adequately discussed. In particular, the results of Section 5.1 are extremely similar in nature to Section 4.1 of the TTRL paper [2]. What contribution does Section 5.1 of this paper provide beyond what was already reported by the TTRL paper? Furthermore, I believe it is worth mentioning the relation to [1], which theoretically analyzes the sharpening mechanism of URLVR.

[1] Zuo, Yuxin, et al. "Ttrl: Test-time reinforcement learning." arXiv preprint arXiv:2504.16084 (2025).

[2] Huang, Audrey, et al. "Self-improvement in language models: The sharpening mechanism." ICLR 2025.

Review Summary and Recommendation
---
Overall, I believe that the paper does not meet the requirements for publication at ICLR. The most significant limitation of the current manuscript is that it contains underspecified and potentially unsound theory. Beyond fixing this issue, it would greatly strengthen the contributions of the paper to elaborate on the unified perspective and its uses, clarify relation to prior work and conventional wisdom regarding how URLVR works, and provide quantitative evidence for the predictiveness of initial reward dynamics of the success of URLVR.



Additional (More Minor) Comments
---
1. The notation of Table 1 is defined only in the appendix, which makes it difficult to parse. I would recommend having the necessary notation in the main text (e.g., in the table caption).

2. There are some missing implementation details, which make it difficult to gauge the significance of some empirical contributions.
    - In Figure 4, how are the subsets chosen? Are these just chosen at random? Also, is the behavior consistent across random subsets? Intuitively, while some small subsets will not suffer from a collapse in ground truth reward, for others they should if the majority vote is incorrect in a non-negligible portion of the examples.
    - The setup in Section 4.3.2 is unclear. For example, how many samples are used?
    - The setup in Section 5.1 is unclear. For example, on which dataset are these experiments ran?

3. The definition of reward hacking mentioned in line 355 does not seem to accord with its conventional use. If the intrinsic reward is maximized and the ground truth reward also keeps increasing, then why should this be considered hacking?

**Questions:**

--

---

> ### Author Response · Authors · 2025-11-23
> **Opening Remarks**
>
> We have carefully reviewed the weaknesses you identified, and we are truly grateful for your thorough review, which will greatly help us improve the quality of our manuscript.
>
> Overall, regarding the first two weaknesses you raised, we believe the issues stem primarily from unclear descriptions in the manuscript, which may have led to misunderstandings. We sincerely apologize for this and will work to refine the manuscript and provide clearer explanations.
>
> As for the latter two weaknesses, we agree that a stronger comparison with and discussion of prior work is necessary. However, we would like to kindly clarify that the exploration of underlying mechanisms constitutes only part of our contribution, and even at the mechanistic level, we believe we provide several novel insights.
>
> **We sincerely hope that the following comments will address all of your concerns:**

---

> ### Author Response · Authors · 2025-11-23
> **Response to W1 (Part 1)**
>
> > **W1.1:** **The unified reward framework in Section 3.1 is currently not well-defined. Specifically, how does the right hand side of Equation (1) depend on y? Should the cross-entropy term h be $- q^{i}(y \mid x) \ln \pi_ {\theta}^{i}(y \mid x)$?**
>
> We thank the reviewer for the detailed review of our framework. We acknowledge that the initial presentation lacked explicit definitions for the notation in Equation (1), leading to ambiguity. We have addressed your specific mathematical questions below and revised the manuscript accordingly.
>
> **Dependence on $y$**
>
> The reward $r_ {\text{uni}}(x,y)$ depends on the response sequence $y = (y_ 1, \ldots, y_ {|y|})$ through three components in Equation (1):
>
> - Aggregation Granularity ($\mathcal{I}$): The set of indices over which we aggregate depends on the structure of $y$. For token-level methods, $\mathcal{I} = \{1, \ldots, |y|\}$ corresponds to the sequence of token positions in $y$, meaning the summation length depends on the response length $|y|$.
> - Model Distribution ($\pi_ {\theta}^i$: The distribution being evaluated depends on the generated history. For token-level granularity at position $t$, $\pi_ {\theta}^t$ denotes the conditional distribution $\pi_ {\theta}(\cdot \mid x, y_ {<t})$, which depends on the prefix $y_ {<t}$ derived from $y$.
> - Anchor Distribution ($q^i$): In methods like Probability or Trajectory-Level Entropy, the anchor $q^t$ is a one-hot distribution $\delta^{y_ t}$ centered on the specific token $y_ t$ generated at that step.
>
> **The Cross-Entropy Term**
>
> We define $\mathbb{H}(q^i, \pi_ \theta^i)$ as the general statistical cross-entropy summed over the vocabulary space $\mathcal{V}^i$: $ \mathbb{H}(q^i, \pi_ \theta^i) = -\sum_ {v \in \mathcal{V}^i} q^i(v) \log \pi_ \theta^i(v) $. Specifically:
>
> - For token-level granularity at position $t$:  $\mathbb{H}(q^t, \pi_ {\theta}^t) = -\sum_ {v \in \mathcal{V}} q^t(v) \log \pi_ {\theta}^t(v)$ where $\mathcal{V}$ is the token vocabulary and $v$ represents individual tokens. Both $q^t$ and $\pi_ {\theta}^t$ are distributions over the vocabulary, not over complete sequences.
> - For answer-level granularity:  $\mathbb{H}(q^{\mathcal{A}}, \pi_ {\theta}^{\mathcal{A}}) = -\sum_ {a \in \mathcal{A}} q^{\mathcal{A}}(a) \ln \pi_ {\theta}^{\mathcal{A}}(a)$ where $\mathcal{A}$ is the set of distinct semantic answers (e.g., numerical values, extracted final answers) and $a$ represents complete answers.
>
> To ensure this is rigorous in the manuscript, we have revised Section 3.1 (marked in red) to explicitly define these components immediately following Equation (1), clarifying the dependencies on $y$ for the granularity $\mathcal{I}$, the prefix-conditioned model $\pi_ {\theta}^i$, and the anchor $q^i$.
>
> > **W1.2:** **Moreover, the significance of such a unified perspective greatly depends on whether it allows characterizing similarities and differences between different instances.**
>
> You are right that the original presentation did not sufficiently demonstrate how the framework reveals similarities and differences. We address this below and acknowledge a key limitation that we resolve in our response to Weakness 2.3.
>
> **Similarities:**
>
> - Common sharpening mechanism: Without the unified framework, methods appear fundamentally different (majority voting vs. entropy minimization vs. probability maximization). The framework reveals they share a common principle: all manipulate cross-entropy to sharpen distributions. The sign factor $\sigma$ formalizes this:
>   - When $\sigma = +1$ (uniform anchor): The reward increases with cross-entropy. Sequences where $\pi_ {\theta}$ diverges from the uniform distribution $U_ V$ receive higher rewards, pushing the policy toward peaked distributions.
>   - When $\sigma = -1$ (sharp anchor): The reward decreases with cross-entropy. Sequences where $\pi_ {\theta}$ aligns with sharp distributions $q$ (e.g., $\delta^t$, $\delta^{\mathcal{A}}$) receive higher rewards, reinforcing existing confident predictions.
>   -  Both cases lead to the same outcome: progressive sharpening, either by moving away from uniformity or by reinforcing peaked predictions. **This unifying insight was not obvious** **without** **the framework.**
> - Shared convergence behavior: We claim all methods induce convergence toward deterministic policies. However, we acknowledge a limitation: the current manuscript derives optimal policies individually for each method (Section 3.2 for majority voting, Appendix A.6 for others), **not through the unified reward formula $r_ {\text{uni}}$**. This limits the framework's demonstrated utility. We address this in our response to Weakness 2.3, where we analyze generalized sharpening using $r_ {\text{uni}}$ directly. This derivation substantiates the framework's soundness. We hope this can address your concerns.

---

> > ### Author Response · Authors · 2025-11-23
> > **Response to W1 (Part 2)**
> >
> > **Differences:**
> >
> > The framework decomposes methods into components ($\mathcal{I}$, $\psi$, $q$, etc.) that predict distinct failure modes:
> >
> > - Granularity $\mathcal{I}$: Token-level ($\mathcal{I} = \{1, \ldots, |y|\}$) creates local pressure, diluted by sequence length. Answer-level ($\mathcal{I} = \{\mathcal{A}\}$) applies global pressure, filtering local noise. This predicts answer-level methods (Majority Voting) exhibit better stability, confirmed in Figure 3.
> > - Transformation $\psi$: Identity ($\psi(z) = z$) gives gradual reinforcement. Exponential ($\psi(z) = \exp(z)$) amplifies sharpening. Probability uses $\psi(z) = \exp(|y| \cdot z)$, the strongest amplification, predicting fastest collapse, which is confirmed in Figure 3.
> > - Anchor $q$: Sharp anchors create self-reinforcement. For Trajectory-Level Entropy, $q = \delta^{y_t}$ gives $\mathbb{H}(\delta^{y_t}, \pi_{\theta}^t) = -\log \pi_{\theta}^t(y_t)$, directly rewarding generated tokens. For Majority Voting, $q = \delta^{\text{maj}}$ concentrates on whatever the majority selects.
> >
> > These structural distinctions, revealed through the unified lens, explain the distinct failure behaviors in Figure 3. The framework provides insights for method selection.
> >
> > In the revised manuscript, we have added remarks in Appendix A.1 (marked in red) explicitly stating these similarities and differences, making the framework's utility transparent in the paper.
> >
> > > **W1.3:** **However, the unified framework is not really used in the paper after its definition, which raises the question of why it is necessary or helpful.**
> >
> > We acknowledge this criticism. While we derive optimal policies individually for each method rather than through the unified formula, the framework serves as the conceptual foundation that guides our entire investigation.
> >
> > **How the framework is used:**
> >
> > - Grounds the confidence-correctness hypothesis: Theorem 1 shows the policy converges geometrically to the initial majority answer $\text{maj}_0(Y)$. This convergence to the initial preference directly implies: success requires initial confidence aligns with correctness. When $\text{maj}_0(Y)$ is correct, convergence amplifies good solutions. When $\text{maj}_0(Y)$ is wrong, convergence amplifies errors. This theoretical prediction motivates our entire empirical investigation:
> >   - Section 4.1: Why do methods work? They trade uncertainty for performance through sampling-efficiency shortcuts, validated when confidence-correctness correlation is strong. In Section 4.1.2, we conducted fine-grained per-problem analysis, which directly validates Theorem 1's prediction, showing success depends on initial confidence-correctness correlation at the instance level.
> >   - Section 4.3: When is training safe? Small datasets prevent collapse because limited diversity cannot establish strong incorrect majorities, maintaining the alignment condition required by Theorem 1.
> >   - Section 5.2: Model-task prior as diagnostic. Early training dynamics indicate whether a model possesses the initial confidence-correctness alignment necessary for RL scalability.
> > - Explains distinct failure modes: The framework's structural components predict the different failure behaviors in Section 4.2 (Figure 3)
> >   - Granularity $\mathcal{I}$: Answer-level methods filter local noise, exhibiting greater stability than token-level methods.
> >   - Transformation $\psi$: Probability's $\psi(z) = \exp(|y| \cdot z)$ amplifies length effects, causing brevity collapse. Identity transformations lead to repetition pathology.
> > - Guides hyperparameter analysis: In Appendix B.3, we use the framework to explain hyperparameter effects:
> >   - Off-policy degree: Small mini-batches violate the on-policy assumption underlying the optimal policy derivation, causing "reward staleness" that destabilizes training (Figure 9).
> >   - Rollout size $N$: More rollouts create more confident majorities, strengthening the "rich-get-richer" dynamic in Theorem 1, accelerating geometric convergence and earlier lock-in to initial preferences (Figure 11).
> >
> > **Addressing the limitation:**
> >
> > We acknowledge that not using $r_{\text{uni}}$ directly in optimal policy derivation limits the framework's demonstrated utility. We address this in our response to Weakness 2.3, where we analyze generalized sharpening using $r_{\text{uni}}$ directly.

---

> ### Author Response · Authors · 2025-11-23
> **Response to W2 (Part 1)**
>
> > **W2.1:** **The theoretical analysis contains potentially incorrect claims.**
> >
> > - **The reward in majority voting depends on the current policy. It therefore changes during training. The closed form solution to the KL-regularized RL objective in Equation (3) holds for a fixed reward, and so it is not clear what it implies for a policy-dependent reward such as majority voting. In particular, without proof, there is no reason to believe that Equation (4) holds. Do the authors have a proof for this claim? Also, the majority reward is not formally defined: what does maj(Y)  stand for? How many rollouts are considered in this majority?**
>
> Thank you for this crucial theoretical observation. You are absolutely correct that Equation (4) (in the revised version is Eq. 3) requires justification when the reward is policy-dependent.
>
> **Clarifying the training procedure:**  In our experiments, we use on-policy RL with periodic reward recomputation. At iteration $k$:
>
> - Sample rollouts from current policy $\pi_ \theta^{(k)}$
> - Compute majority voting reward $r_ k(x,y) = \mathbf{1}[\text{ans}(y) = \text{maj}_ k(Y)]$ where $\text{maj}_ k(Y)$ is the majority answer among current rollouts
> - Perform one policy update using $r_ k(x,y)$ to obtain $\pi_ \theta^{(k+1)}$
> - Repeat with new rollouts from $\pi_ \theta^{(k+1)}$
>
> You correctly identify that closed-form solution assumes fixed $r$, while our $r_ k$ changes each iteration.
>
> **Formal definition of** $\text{maj}(Y)$**:**
>
> We apologize for not defining this clearly:
>
> - $Y = \{y^{(1)}, \ldots, y^{(N)}\}$: $N$ rollouts from $\pi_ \theta^{(k)}$
> - $\text{maj}_ k(Y) = \arg\max_ {a} |\{i \in [N] : \text{ans}(y^{(i)}) = a\}|$: most frequent answer
> - In our main experiments: $N = 8$
>
> **What Equation (4) (currently Eq. 3) actually represents?** Equation (4) (Eq. 3) describes the optimal policy for fixed reward $r_ k$, not the policy $\pi_ \theta^{(k+1)}$ we actually obtain after one update. Specifically, if we held $r_ k$ fixed and performed infinite updates starting from $\pi_ \theta^{(k)}$, we would converge to:
> - If $\text{ans}(y) = \text{maj}_ k(Y)$, $\pi_ \theta^{* ,k}(y|x) = \frac{\pi_ \theta^{(k)}(y|x) \cdot e^{1/\beta}}{Z_ k(x)}$
> - Otherwise, $\pi_ \theta^{* ,k}(y|x) = \frac{\pi_ \theta^{(k)}(y|x)}{Z_ k(x)}$
>
> where $Z_ k(x) = p_ {\text{maj}}^{(k)} \cdot e^{1/\beta} + (1 - p_ {\text{maj}}^{(k)})$ and $p_ {\text{maj}}^{(k)} = \sum_ {y: \text{ans}(y) = \text{maj}_ k(Y)} \pi_ \theta^{(k)}(y|x)$.  This gives the probability mass at optimum (Equation 5): $p_ {\text{maj}}^{* ,(k+1)} = \frac{p_ {\text{maj}}^{(k)} \cdot e^{1/\beta}} {p_ {\text{maj}}^{(k)} \cdot e^{1/\beta} + (1 - p_ {\text{maj}}^{(k)})}$.
>
> To validate that this optimum can be achieved, we conduct an experiment under an extreme off-policy setting (global batch size = 1024, mini-batch size = 1). Specifically, we generate one time rollout for all 1024 prompts (with N=8 for each prompt) and perform 1024 updates using rewards computed solely from the initial rollout majority. This setup tests whether convergence can be achieved in a single rollout step, with all updates relying on the same reward signal. The result shows that after one rollout and 1024 mini-updates, the model collapses: the majority reward reaches its maximum value of 1, while validation performance on AIME24, AIME25, and AMC23 drops to zero. If we assume that the collapse point under the intrinsic reward setting corresponds to the convergence point, the convergence point can be achieved with a sufficient large mini-updates (e.g. 1024).
>
> However, our actual policy after one update $\pi_ \theta^{(k+1)}$ **only moves toward this optimum, not reaching it.** The key observation: the actual probability mass after one update satisfies the ordering: $p_ {\text{maj}}^{*,(k+1)} \geq p_ {\text{maj}}^{(k+1)} \geq p_ {\text{maj}}^{(k)}$.

---

> ### Author Response · Authors · 2025-11-23
> **Response to W2 (Part 2)**
>
> **Why this ordering holds**
>
> - Lower bound ($p_ {\text{maj}}^{(k+1)} \geq p_ {\text{maj}}^{(k)}$): The policy gradient is: $\nabla_ \theta J = \mathbb{E}_ {\pi_\theta}[r_ k(x,y) \nabla_ \theta \log \pi_ \theta(y|x)$. Since $r_ k(x,y) = 1$ for majority trajectories and $r_ k(x,y) = 0$ for non-majority trajectories, the gradient increases $\log \pi_ \theta(y|x)$ only for majority trajectories. Therefore, after one gradient update with learning rate $\eta$: $\log \pi_ \theta^{(k+1)}(y|x) \approx \log \pi_ \theta^{(k)}(y|x) + \eta \cdot r_ k(x,y) \cdot [\text{advantage terms}]$. **Here we use $\approx$ because we conduct stochastic gradient descent. We will empirically validate this increase later.** For majority trajectories with positive advantage, this increases their probability. Hence: $p_ {\text{maj}}^{(k+1)} = \sum_ {y: \text{ans}(y) = \text{maj}_ {k+1}(Y)} \pi_ \theta^{(k+1)}(y|x) \geq \sum_ {y: \text{ans}(y) = \text{maj}_ k(Y)} \pi_ \theta^{(k)}(y|x) = p_ {\text{maj}}^{(k)}$.
> - Upper bound ($p_ {\text{maj}}^{(k+1)} \leq p_ {\text{maj}}^{* ,(k+1)}$): Since $\pi_ \theta^{*  ,k}$ maximizes the KL-regularized objective for fixed $r_ k$, it achieves the highest possible expected reward. Our single-step update only moves partway toward this maximum, so we cannot exceed the optimal probability mass: $p_ {\text{maj}}^{(k+1)} \leq p_ {\text{maj}}^{* ,(k+1)}$.
>
> This ordering establishes that $p_ {\text{maj}}^{(k)}$ is monotonically increasing and bounded. The key remaining question is: **does the majority answer** $\text{maj}_ k(Y)$ **remain stable across iterations?**
>
> With $N$ rollouts from $\pi_ \theta^{(k)}$, each rollout independently yields answer $a$ with probability $\pi_ \theta^{(k)}(a|x)$. By the Law of Large Numbers, as $N$ increases, the empirical frequency of each answer converges to its true probability: $\frac{|\{i : \text{ans}(y^{(i)}) = a\}|}{N} \xrightarrow{N \to \infty} \pi_ \theta^{(k)}(a|x)$.
>
> Therefore, $\text{maj}_ k(Y)$ (the most frequent answer in rollouts) converges to $\arg\max_ a \pi_ \theta^{(k)}(a|x)$ (the most probable answer under the policy). Since $p_ {\text{maj}}^{(k)}$ increases monotonically, the most probable answer remains $\text{maj}_ 0(Y)$ throughout training: $\arg\max_ a \pi_ \theta^{(k)}(a|x) = \text{maj}_ 0(Y)$ for all $k$.  In practice, even moderate $N$ (we use $N=8$) provides reasonable stability.
>
> Given stable $\text{maj}_ k(Y) = \text{maj}_ 0(Y)$, the monotonic increase in $p_ {\text{maj}}^{(k)}$ leads to geometric convergence. We provide the detailed convergence proof using this ordering in our next response.
>
> **Empirical validation:** To validate this theoretically predicted monotonic increase, we conducted two experiments:
>
> - Experiment 1 (Single problem, large rollouts): We trained on a single problem on-policy from MATH-500 with $N=1024$ rollouts (reducing majority vote randomness) for 10 steps. We found that the majority answer $\text{maj}_ k(Y)$ remain stable across iterations. Results show $p_ {\text{maj}}$ increases monotonically, confirming the theoretical prediction. We randomly chose 4 problems for 4 runs, detailed table recording $p_ {\text{maj}}$ is as follows:
>
> | Data ID      | Step 1 | Step 2 | Step 3 | Step 4 | Step 5 | Step 6 | Step 7 | Step 8 | Step 9 | Step 10 |
> | ------------ | ------ | ------ | ------ | ------ | ------ | ------ | ------ | ------ | ------ | ------- |
> | level3_id146 | 12.70% | 15.53% | 15.92% | 16.21% | 18.46% | 22.07% | 22.56% | 24.80% | 31.35% | 39.36%  |
> | level1_id187 | 6.64%  | 6.69%  | 6.84%  | 7.42%  | 10.45% | 11.04% | 11.43% | 11.82% | 15.82% | 18.46%  |
> | level1_id262 | 15.14% | 17.19% | 17.48% | 18.85% | 20.02% | 22.07% | 24.12% | 25.20% | 34.67% | 39.84%  |
> | level3_id122 | 11.33% | 12.01% | 12.40% | 14.06% | 17.87% | 18.46% | 20.31% | 21.29% | 33.59% | 33.89%  |
>
> - Experiment 2 (Batch training): One of our experiments (Figure 1) trains on batches with $N=8$ rollouts. The Majority Voting Reward curve (which equals $p_ {\text{maj}}$ averaged over batch) shows consistent increasing trend, though with small fluctuations due to finite rollouts and batch variance.
>
> **Revised manuscript (marked in red):**
>
> Section 3.2 now explicitly states:
>
> - Equation (4) describes the optimal policy for fixed $r_ k$ at each iteration
> - Our single-step updates yield $p_ {\text{maj}}^{(k+1)}$ satisfying the ordering above
> - Formal definition of $\text{maj}_ k(Y)$ with $N$ rollouts
>
> We thank the reviewer for this rigorous examination, which clarifies the relationship between fixed-reward optimality and our iterative training procedure.

---

> > ### Author Response · Authors · 2025-11-23
> > **Response to W2 (Part 3)**
> >
> > > **W2.2: The theoretical analysis contains potentially incorrect claims.**
> > >
> > > - **Theorem 1 is not rigorously stated. It is therefore difficult to evaluate what it means or whether it is sound. Specifically, what does "majority trajectories" refer to? What are "k updates"? Does this refer to k steps of policy gradient or rather k iterations of solving perfectly the KL-regularized objective and each time updating the reference policy to be the current policy?**
> >
> > Thank you for requesting clarification on the statement and proof of Theorem 1. We address each ambiguity:
> >
> > **What are "majority trajectories"?**
> >
> > At iteration $k$, majority trajectories are those whose extracted answer equals the majority answer from the current rollouts: $\mathcal{T}_ {\text{maj}}^{(k)} = \{y : \text{ans}(y) = \text{maj}_ k(Y)\}$. The probability mass on majority trajectories is: $p_ {\text{maj}}^{(k)} = \sum_ {y \in \mathcal{T}_ {\text{maj}}^{(k)}} \pi_ \theta^{(k)}(y|x)$ .
> >
> > **What are "$k$ updates"?**
> >
> > This refers to $k$ **iterations of the training loop,** each iteration consists of:
> >
> > - Sample $N$ rollouts from $\pi_ \theta^{(k)}$
> > - Compute majority reward $r_ k$
> > - Perform **one gradient update** to obtain $\pi_ \theta^{(k+1)}$
> >
> > Although in each step we can't ensure achieving the optimal policy induced by solving perfectly the KL-regularized objective, in the previous response, we put $\pi_ \theta^{(k+1)}$ "inside" $\pi_ \theta^{*,(k+1)}$and $\pi_ \theta^{(k)}$.
> >
> > In other words, each "update" in Theorem 1 is **not** solving the KL-regularized objective to convergence. It is a single gradient step that moves the policy partway toward the optimal policy for fixed $r_ k$.
> >
> > **How do we prove convergence with partial updates?**
> >
> > The original proof (Appendix A.3) assumed that at each iteration, we reach the optimal policy $\pi_ \theta^{* ,k}$ given by Equation (4) (currently Eq. 3), which yields the exact update rule: $p_ {\text{maj}}^{* ,(k+1)} = \frac{\alpha \cdot p_ {\text{maj}}^{(k)}}{1 + (\alpha-1)p_ {\text{maj}}^{(k)}}, \quad \alpha = e^{1/\beta} > 1$
> >
> > However, as established in the previous response, our single gradient update only achieves: $p_ {\text{maj}}^{* ,(k+1)} \geq p_ {\text{maj}}^{(k+1)} \geq p_ {\text{maj}}^{(k)}$. We prove convergence under this weaker condition. Define the effective update: $p_ {\text{maj}}^{(k+1)} = p_ {\text{maj}}^{(k)} + \eta_k \cdot (p_ {\text{maj}}^{* ,(k+1)} - p_ {\text{maj}}^{(k)}) $, where $\eta_ k \in [0,1]$ represents the "step efficiency".
> >
> > Substituting $p_ {\text{maj}}^{* ,(k+1)} = \frac{\alpha \cdot p_ {\text{maj}}^{(k)}}{1 + (\alpha-1)p_ {\text{maj}}^{(k)}}$: $p_ {\text{maj}}^{(k+1)} = p_ {\text{maj}}^{(k)} + \eta_k \left(\frac{\alpha \cdot p_ {\text{maj}}^{(k)}}{1 + (\alpha-1)p_ {\text{maj}}^{(k)}} - p_ {\text{maj}}^{(k)}\right)$
> >
> > Simplifying: $p_ {\text{maj}}^{(k+1)} = p_ {\text{maj}}^{(k)} + \eta_ k \cdot \frac{(\alpha - 1)(1 - p_ {\text{maj}}^{(k)})p_ {\text{maj}}^{(k)}}{1 + (\alpha-1)p_ {\text{maj}}^{(k)}}$
> >
> > Define error $\epsilon^{(k)} = 1 - p_ {\text{maj}}^{(k)}$. Then: $\epsilon^{(k+1)} = \epsilon^{(k)} - \eta_ k \cdot \frac{(\alpha - 1)(1 - \epsilon^{(k)})\epsilon^{(k)}}{1 + (\alpha-1)(1-\epsilon^{(k)})} = \epsilon^{(k)} \left(1 - \eta_ k \cdot \frac{(\alpha - 1)(1 - \epsilon^{(k)})}{\alpha - (\alpha-1)\epsilon^{(k)}}\right)$
> >
> > Since $\alpha > 1$, $\epsilon^{(k)} \in (0,1)$, and $\eta_ k \in [0,1]$, the term in parentheses satisfies: $0 < 1 - \eta_ k \cdot \frac{(\alpha - 1)(1 - \epsilon^{(k)})}{\alpha - (\alpha-1)\epsilon^{(k)}} \leq 1$.
> >
> > Therefore $\epsilon^{(k+1)} \leq \epsilon^{(k)}$. We assume there exists $\eta_ {\min} > 0$ such that $\eta_ k \geq \eta_ {\min}$ for all $k$, meaning each gradient update makes non-trivial progress toward the optimal policy. This is a standard assumption in gradient descent convergence analysis, though the exact value of $\eta_ {\min}$ depends on learning rate, optimization algorithm, and problem geometry. Now we have: $\epsilon^{(k+1)} \leq \left(1 - \eta_{\min} \cdot \frac{\alpha - 1}{\alpha}\right) \epsilon^{(k)}$ and $\epsilon^{(k+1)} < \epsilon^{(k)}$.

---

> > > ### Author Response · Authors · 2025-11-23
> > > **Response to W2 (Part 4)**
> > >
> > > For this assumption, we empirically validate its convergence through the same setting as validation experiment 1 in previous Response to W2 (Part 2). Here, we continue training for 50 steps, reporting $p_ {\text{maj}}$ every 5 steps, as it is approaching 1:
> > >
> > > | Data ID      | Step 5 | Step 10 | Step 15 | Step 20 | Step 25 | Step 30 | Step 35 | Step 40 | Step 45 | Step 50 |
> > > | ------------ | ------ | ------- | ------- | ------- | ------- | ------- | ------- | ------- | ------- | ------- |
> > > | level3_id146 | 18.46% | 39.36%  | 48.93%  | 91.11%  | 95.41%  | 98.14%  | 98.54%  | 99.02%  | 99.61%  | 99.80%  |
> > > | level1_id187 | 11.04% | 18.46%  | 26.37%  | 79.88%  | 89.84%  | 93.07%  | 96.09%  | 97.66%  | 98.54%  | 99.02%  |
> > > | level1_id262 | 22.07% | 39.84%  | 51.37%  | 90.14%  | 95.90%  | 96.80%  | 97.46%  | 98.05%  | 98.63%  | 99.41%  |
> > > | level3_id122 | 17.87% | 33.59%  | 43.55%  | 84.28%  | 92.19%  | 93.26%  | 95.80%  | 96.09%  | 98.34%  | 98.54%  |
> > >
> > > For large $k$ (when $\epsilon^{(k)}$ is small). This establishes geometric convergence with rate depending on $\eta_ {\min}$ and $\alpha = e^{1/\beta}$, when $\eta_ k = \eta_ \text{min} = 1$ all the time, we have a convergence rate of $\frac 1 \alpha = e^{-1/\beta}$. Complete proof is included in the revised version.
> > >
> > > **Revised manuscript changes (marked in red):**
> > >
> > > - Section 3.2: Clarified that "$k$ updates" means $k$ iterations of rollout-reward-gradient cycle, each with one gradient step
> > > - Theorem 1: Added explicit assumptions and precise definitions
> > > - Appendix A.3: Added modified proof handling partial gradient updates via step efficiency $\eta_ k$
> > >
> > > We appreciate this careful scrutiny, which has led to a more rigorous and transparent theoretical presentation.
> > >
> > >
> > >
> > > > **W2.3: Generalized Sharpening Analysis using Unified Reward**
> > >
> > > To address the concern that Theorem 1 is limited to Majority Voting, and to demonstrate the analytical utility of our unified reward framework, we provide the generalized sharpening analysis.  We show that methods with $\sigma=-1$ share a critical structural property **Reward-Confidence Monotonicity**, which creates a persistent pressure toward distribution sharpening.
> > >
> > > **Proposition (Sharpening Dynamics for** $\sigma = -1$ **Methods):**
> > >
> > > Consider any intrinsic reward with $\sigma = -1$ in the unified framework ($r_ {\text{uni}} = \psi(-\mathbb{H}(q, \pi))$) where $\psi$ is strictly increasing and $q$ is a sharp anchor. These methods satisfy Reward-Confidence Monotonicity:
> > >
> > > $\pi_ \theta(y_a|x) > \pi_ \theta(y_b|x) \implies r_ {\text{uni}}(x,y_a) > r_ {\text{uni}}(x,y_b)$
> > >
> > > For a dominant trajectory $y^* $ (e.g., majority) and a non-dominant competitor $y$, this inequality is strict: $r(y^* ) > r(y')$. Under iterative KL-regularized updates, this property creates a self-reinforcing feedback loop that drives geometric concentration.

---

> > > > ### Author Response · Authors · 2025-11-23
> > > > **Response to W2 (Part 5)**
> > > >
> > > > **Proof Sketch:**
> > > >
> > > > We analyze the dynamics for a dominant trajectory $y^* $ and a competitor $y$ (for ensemble methods, not in the same class as $y^* $) where the model initially prefers $y^* $ (i.e., $\pi_ k(y^* ) > \pi_ k(y')$) and assigns it strictly higher reward ($r_ k(y^* ) > r_ k(y')$).
> > > >
> > > > 1. Existence of a Positive Reward Gap
> > > >
> > > > Using the unified formula, we justify why the gap is positive for $\sigma=-1$:
> > > >
> > > > - Self-Reinforcing Anchors (e.g., Probability): $r(y) = \psi(\log \pi(y))$. Since $\pi_ k(y^* ) > \pi_ k(y')$ and $\psi$ is strictly increasing, $r_ k(y^* ) > r_ k(y')$.
> > > > - Answer-Level Anchors (e.g., Majority Voting): $y^* $ belongs to the dominant answer class $a^* $, while $y$ does not. By construction, $r(y^* ) = 1$ and $r(y') = 0$.
> > > >
> > > > In both cases, the intrinsic reward gap is strictly positive: $\Delta_ r^{(k)} = r_ k(y^* ) - r_ k(y') > 0$.
> > > >
> > > > 2. The Optimization Target
> > > >
> > > > We consider the optimal policy $\pi^* $ for the current fixed reward landscape $r_ k$. The optimal solution implies a target ratio:
> > > >
> > > > $\frac{\pi^* (y^* )}{\pi^* (y')} = \frac{\pi_ k(y^* )}{\pi_ k(y')} \cdot \exp\left(\frac{\Delta_ r^{(k)}}{\beta}\right)$
> > > >
> > > > Since $\Delta_ r^{(k)} > 0$, the target ratio is strictly larger than the current ratio.
> > > >
> > > > Gradient Assumption: The gradient $\nabla_ \theta J = \mathbb{E}_ {\pi_k}[r_ k(y) \nabla_ \theta \log \pi_ \theta(y)]$ assigns positive weight to high-reward trajectories. We assume that policy gradient updates with positive learning rate $\eta$ satisfy: if $r(y^* ) > r(y')$ and both have positive probability, then the updated policy satisfies $\frac{\pi_ {k+1}(y^* )}{\pi_ {k+1}(y')} \geq \frac{\pi_ k(y^* )}{\pi_ k(y')}$. This aligns with standard policy gradient convergence properties.
> > > >
> > > > 3. The Reinforcement Loop
> > > >
> > > > The unified framework reveals why this process spirals into determinism. As the policy updates to increase the probability mass on the dominant trajectory/class:
> > > >
> > > > - For Self-Reinforcing Anchors (e.g., Probability), because $r(y) = \psi(\log \pi(y))$, increasing $\pi(y^* )$ directly increases its reward $r(y^* )$.
> > > > - For Answer-Level Anchors (e.g., Majority Voting), increasing the total probability mass on the dominant answer class $a^* $ increases the reward for all trajectories in that class (since $r \propto \log p(a^* )$).
> > > >
> > > > This creates a positive feedback loop: the update increases the probability of the dominant path, which maintains or widens the reward gap $\Delta_ r$, ensuring the pressure to sharpen ($\Delta_ r > 0$) persists.
> > > >
> > > > **Utility of the Framework:**
> > > >
> > > > This derivation demonstrates that the "rich-get-richer" dynamic is a structural inevitability for any method where the reward function is monotonically aligned with the model's own confidence ($\sigma=-1$). The framework allows us to identify this shared property and predict that all such methods will drive the policy toward deterministic outputs, regardless of whether this leads to success (when aligned with correctness) or failure (when misaligned).
> > > >
> > > > **Remark on** $\sigma = +1$ **Methods:**
> > > >
> > > > Self-Certainty ($\sigma = +1$) rewards higher when away from uniform distribution. Therefore, $\pi(y_ a) > \pi(y_ b)$ does **not** imply $r(y_ a) > r(y_ b)$. A high-probability output and a very low-probability output could both have high KL-divergence from uniform, violating direct Reward-Confidence Monotonicity. Its sharpening mechanism requires separate analysis.
> > > >
> > > > While methods with $\sigma=+1$ do not strictly align reward with raw confidence, they still induce sharpening by penalizing high-entropy distributions. By maximizing the distance from a uniform anchor, the optimization landscape naturally favors peaked, low-entropy policies, effectively driving the model toward determinism.
> > > >
> > > > **Empirical Validation:**
> > > >
> > > > We acknowledge that our proof sketch relies on stated assumptions and may not achieve full mathematical rigor. To substantiate the assumptions in this proof sketch, we provide empirical validation for different intrinsic reward methods in Figure 3 and Appendix B.3, confirming that the "Confidence-Reward Alignment'' is not just a theoretical construct but the actual driver of the observed training dynamics.
> > > >
> > > > ---
> > > >
> > > > We sincerely appreciate the reviewer's rigorous scrutiny of our theoretical claims. Your reviews regarding policy-dependent rewards and the utility of unified reward framework have pushed us to think more deeply about the mathematical foundations of our framework. The generalized analysis above, while still informal in places, represents a strengthening of our theoretical contribution.
> > > >
> > > > We hope this added analysis helps build a more solid theoretical foundation for the paper. If there are gaps in our reasoning or areas that remain unclear, we would greatly value further feedback and suggestions from the reviewer. Your expertise has already substantially improved the rigor and clarity of our work, and we remain open to additional suggestions.

---

> > > > > ### Author Response · Authors · 2025-11-23
> > > > > **Response to W3 (Part 1)**
> > > > >
> > > > > > **W3: One of the main claims made in the paper is that the success of URLVR stems from sharpening the language model’s distribution on correct outputs, in case it already assigns such outputs a reasonable probability. However, as far as I am aware, this is already the existing conventional wisdom behind why methods such as majority voting as a reward can work (c.f. [1,2]).**
> > > > >
> > > > > Thank you for your thoughtful comment. First, we would like to emphasize that this mechanism is indeed one of our contributions. Although both related works are excellent and can be broadly interpreted as employing sharpening strategies, their primary contributions lie in proposing new methodological frameworks rather than providing a systematic analysis of the underlying mechanisms. To clarify your concerns, we elaborate on several points:
> > > > >
> > > > > 1. First, TTRL does not explicitly attribute its effect to sharpening the model’s output distribution. Instead, it primarily explains its improvement through the Lucky Hit phenomenon (Sec 4.2). While this indeed improves the quality of the supervision signal, we argue that distributional sharpening and concentration **toward correct outputs** play an equally critical role. Importantly, to the best of our knowledge, no prior work has shown whether fitting occurs regardless of whether the output distribution is already concentrated on the correct solution, nor have existing works examined whether such fitting converges or may cause model collapse. In Sections 3 and 4, we provide a detailed analysis of these issues.
> > > > > 2. [2] is an excellent work. We sincerely apologize for not addressing this earlier. We will cite it and include a thorough discussion in the next version of the manuscript to explicitly acknowledge its contributions. While many of that work’s insights resonate with ours, there remain substantial differences in research goals and methodology. In particular:
> > > > >    1. **Different algorithms studied:** Their analyses focus on SFT- and RLHF-based self-improvement methods, whereas we study RLVR-based approaches.
> > > > >    2. **Different research goals:** Their aim is to establish a comprehensive theoretical framework for sharpening, with an emphasis on sample complexity, and to propose SFT-Sharpening and RLHF-Sharpening algorithms validated empirically. They do not address issues related to RL scalability or the possibility of model collapse. **In contrast, as highlighted in our title, we focus on whether and how these approaches can enable scalable RL. This motivates our further investigation into test-time versus train-time discrepancies.**
> > > > >    3. **Different conceptual focus:** Their goal is to characterize the sample complexity required for a model to be sharpened under SFT- or RLHF-based procedures. Our goal is to derive Theorem 1 from a sharpening perspective and use correctness–confidence alignment to analyze why intrinsic-reward methods work (Sec. 4.1), why they fail and how such failures manifest (Sec. 4.2), and how to avoid these failure modes (Sec. 4.3).
> > > > >    4. **Different scope of methods:** Their sharpening framework is based on the ML-sharpening objective (corresponding to RLSC). By contrast, we study a broader class of URLVR intrinsic rewards, where majority voting is only one instance among others such as self-certainty and token-level entropy. Their discussion of majority voting as an alternative self-reward function (Sec. 5.3) involves inference-time sharpening using best-of-*n*. To the best of our knowledge, the recent work [3] also adopts a similar perspective. In contrast, our test-time training experiments (Sec. 4.3) perform self-reward RL training directly on the test distribution. We further study generalization properties: for example, Sec. 4.3.2 shows that performing test-time training on AMC23 sharpens not only the AMC23 distribution but also the AIME2024 distribution. Thus, while they study sharpening on the *same* task, we additionally demonstrate *cross-task* sharpening effects.
> > > > >
> > > > > Overall, [2] provides valuable complementary support to the main theoretical claims in our work. Our understanding may not be perfect, and we welcome correction if we have misinterpreted any aspect.
> > > > >
> > > > > ---
> > > > >
> > > > >
> > > > > [1] Zuo, Yuxin, et al. "Ttrl: Test-time reinforcement learning." arXiv preprint arXiv:2504.16084 (2025).
> > > > >
> > > > > [2] Huang, Audrey, et al. "Self-improvement in language models: The sharpening mechanism." ICLR 2025.
> > > > >
> > > > > [3] Reasoning with Sampling: Your Base Model is Smarter Than You Think

---

> > > > > > ### Author Response · Authors · 2025-11-23
> > > > > > **Response to W3 (Part 2)**
> > > > > >
> > > > > > We would also like to stress that sharpening is only one part of our contributions. A substantial portion of our work focuses on deeper analyses and further conclusions. For instance, we study the fine-grained mechanisms behind these methods (Sec. 4.2 and Fig. 2). **In our Response to W1 from Reviewer Kjs9**, we also provide an extended discussion of these issues, which we hope will help address your concerns.
> > > > > >
> > > > > > Moreover, although methodological innovation is not the main focus of this paper, we do present new methods. These appear in Sec. 5.2 and Sec. 5.3, originally intended as pointers for future work. To fully address your concern about our contributions, we strengthened these experiments during the rebuttal:
> > > > > >
> > > > > > - **Regarding Sec. 5.2**, we propose a new approach for assessing model priors. Compared to RLVR training and pass@k metrics, our method can more efficiently identify which base model is better suited for RL, making it particularly attractive for foundation model development in practice. Please refer to **[Global Response 1]** for more details.
> > > > > > - **Regarding Sec. 5.3 (Exploiting Generation–Verification Asymmetries)**, after identifying the risk of model collapse when applying existing intrinsic-reward methods at train time (Sec. 4.3), we propose a new intrinsic reward method "self-verify" that enables scalable RL at train time. This method leverages the asymmetry of generation-verification, enabling URLVR with reward signals closer to ground truth to avoid collapse. We find that under long-horizon training, especially when certainty-based methods inevitably collapse, self-verify continues to improve performance and achieves results far surpassing existing intrinsic-reward methods, without any hacking. Further details are presented in **[Global Response 2]**.
> > > > > >
> > > > > > We hope these clarifications and additional results alleviate your concerns, and we sincerely appreciate the opportunity to further refine and strengthen our work.

---

> > > > > > > ### Author Response · Authors · 2025-11-23
> > > > > > > **Response to W4**
> > > > > > >
> > > > > > > > **W4.1:** **Relation to prior work is often not adequately discussed. In particular, the results of Section 5.1 are extremely similar in nature to Section 4.1 of the TTRL paper [2]. What contribution does Section 5.1 of this paper provide beyond what was already reported by the TTRL paper?**
> > > > > > > >
> > > > > > > > [1] Zuo, Yuxin, et al. "Ttrl: Test-time reinforcement learning." arXiv preprint arXiv:2504.16084 (2025).
> > > > > > > >
> > > > > > > > [2] Huang, Audrey, et al. "Self-improvement in language models: The sharpening mechanism." ICLR 2025.
> > > > > > >
> > > > > > > Thank you very much for your valuable feedback! We appreciate the reviewer’s concern, and in the next revision we will make a greater effort to thoroughly compare our work with prior works. In addition, we would like to clarify several points that address the specific question regarding the relation between our Section 5.1 and Section 4.1 of the TTRL paper:
> > > > > > >
> > > > > > > 1. **Different experimental goals and consequently different contributions:** Although the experimental setup in our Section 5.1 resembles that of TTRL Section 4.1, the underlying objectives differ fundamentally, leading to different insights. TTRL primarily compares the performance of its method to the Base model’s maj@64, with the goal of examining whether TTRL can exceed the supervision signal implicitly defined by the base model’s majority vote. While this comparison is intuitively reasonable, it is inherently imperfect: the computational cost of obtaining maj@64 for the base model is lower than the cost of TTRL training. **In contrast, our experiment eliminates this source of unfairness entirely.** We further measure the base model’s maj@1024 and observe that as the inference compute increases, the performance asymptotically stabilizes, approximating maj@∞. In this regime, the computational cost of the base model’s majority vote matches or exceeds that of TTRL training, creating a fair comparison. **Under this fair setting, we are able to draw a stronger conclusion:** TTRL indeed yields genuine model capability improvement, rather than merely benefiting from additional computational expenditure.
> > > > > > > 2. **This experiment is presented as discussion rather than core analysis:** We deliberately placed this experiment in the Discussion section rather than in Empirical Analysis, precisely because it is not intended to constitute an independent or fully systematic study. The setup is related to but not identical to the TTRL experiment, and we treat it as a supplemental discussion rather than a central empirical contribution. For this reason, we also intentionally kept Section 5 concise so as not to overshadow the main results. We hope the reviewer will understand that although we could replace Section 5.1 with unrelated material from our appendix, we chose to retain it because it plays an important role in maintaining the logical completeness and coherence of the paper.
> > > > > > >
> > > > > > > > **W4.2:** **Furthermore, I believe it is worth mentioning the relation to [1], which theoretically analyzes the sharpening mechanism of URLVR.**
> > > > > > > >
> > > > > > > > [1] Zuo, Yuxin, et al. "Ttrl: Test-time reinforcement learning." arXiv preprint arXiv:2504.16084 (2025).
> > > > > > > >
> > > > > > > > [2] Huang, Audrey, et al. "Self-improvement in language models: The sharpening mechanism." ICLR 2025.
> > > > > > >
> > > > > > > Thank you very much for the valuable suggestions. We agree that both works deserve a more thorough and careful discussion, and we will incorporate a substantially expanded analysis in the next revision of the manuscript. Specifically:
> > > > > > >
> > > > > > > 1. **Regarding Self-improvement in language models**, we will provide a more comprehensive and in-depth examination of its mechanisms and contributions, and draw a systematic comparison with our own approach. Moreover, in our Response to W3, we have discussed in detail the key differences between this work and ours, and we will integrate these clarifications into the revised manuscript.
> > > > > > > 2. **Regarding TTRL**, this is indeed an excellent and influential piece of work, and we agree that its relationship to our method and to other relevant prior studies should be articulated more explicitly. At the same time, we would like to clarify that TTRL does not provide a systematic theoretical analysis of its mechanism; rather, it primarily relies on empirical evidence to support its effectiveness and attributes its mechanism largely to the "Lucky Hit" phenomenon. Nevertheless, TTRL can certainly be understood as operating within the broader sharpening mechanism framework. We greatly appreciate the insights offered by TTRL, and in the next revision we will include a more detailed discussion of how it relates to our method and how our theoretical analysis complements its empirical findings.

---

> > > > > > > > ### Author Response · Authors · 2025-11-23
> > > > > > > > **Response to Additional Comment (Part 1)**
> > > > > > > >
> > > > > > > > > **Additional Comment 1: The notation of Table 1 is defined only in the appendix, which makes it difficult to parse. I would recommend having the necessary notation in the main text (e.g., in the table caption).**
> > > > > > > >
> > > > > > > > Thank you for this valuable feedback. You are right that the notation in Table 1 was not adequately defined in the main text, making it difficult to parse without consulting the appendix. We sincerely apologize for this oversight.
> > > > > > > >
> > > > > > > > **In the revised manuscript (marked in red)**, we have made the following changes:
> > > > > > > >
> > > > > > > > - **Added explicit symbol definitions immediately after Equation (1)**: We now define all key components in logical order: starting with the input/output $(x,y)$, then aggregation granularity $\mathcal{I}$, model distribution $\pi_ {\theta}^i$, anchor distribution $q^i$, cross-entropy $\mathbb{H}$, sign factor $\sigma$, and monotonic transformation $\psi$. Each definition includes concrete examples to aid understanding.
> > > > > > > > - **Enhanced Table 1 column headers**: Added clarifying labels in the headers (e.g., "$\psi(z)$: Monotonic transformation", "$q$: Anchor distribution", "$\pi_ {\theta}$: Model distribution") so readers can immediately understand what each column represents.
> > > > > > > > - **Added explanatory text after Table 1**: We now explain how each method chooses its granularity and sign, and include a brief remark on the special notation for Majority Voting.
> > > > > > > >
> > > > > > > > We appreciate the reviewer pointing this out. The revised presentation makes the framework accessible without requiring navigation between sections.
> > > > > > > >
> > > > > > > >
> > > > > > > >
> > > > > > > > > **Additional Comment 2.1: In Figure 4, how are the subsets chosen? Are these just chosen at random? Also, is the behavior consistent across random subsets? Intuitively, while some small subsets will not suffer from a collapse in ground truth reward, for others they should if the majority vote is incorrect in a non-negligible portion of the examples.**
> > > > > > > >
> > > > > > > > Thank you for raising this critical question. **Yes, subsets are chosen randomly from DAPO-17k.** We agree this is an important detail that was missing.
> > > > > > > >
> > > > > > > > **Verification across random seeds:** To verify consistency, we have conducted additional experiments with 3 random seeds for subset sizes {32, 128, 512}. Results are shown below:
> > > > > > > >
> > > > > > > > **Seed: 123**
> > > > > > > >
> > > > > > > > Ground Truth Reward:
> > > > > > > >
> > > > > > > > | Step     | 0     | 60    | 120    | 180    | 240    | 300    | 360    | 420    | 480    | 540    | 600    |
> > > > > > > > | -------- | ----- | ----- | ------ | ------ | ------ | ------ | ------ | ------ | ------ | ------ | ------ |
> > > > > > > > | DAPO-32  | 8.20% | 8.98% | 12.11% | 16.02% | 17.97% | 14.45% | 15.63% | 15.63% | 15.63% | 15.63% | 15.63% |
> > > > > > > > | DAPO-128 | 5.47% | 7.81% | 10.94% | 13.67% | 12.11% | 16.41% | 15.23% | 12.89% | 11.72% | 12.50% | 12.50% |
> > > > > > > > | DAPO-512 | 7.42% | 9.38% | 22.27% | 10.55% | 17.97% | 10.16% | 9.77%  | 3.13%  | 3.13%  | 0.00%  | 0.00%  |
> > > > > > > >
> > > > > > > > Majority Voting Reward:
> > > > > > > >
> > > > > > > > | Step     | 0      | 60     | 120    | 180    | 240    | 300    | 360    | 420    | 480     | 540     | 600     |
> > > > > > > > | -------- | ------ | ------ | ------ | ------ | ------ | ------ | ------ | ------ | ------- | ------- | ------- |
> > > > > > > > | DAPO-32  | 16.41% | 25.00% | 31.64% | 41.02% | 59.77% | 82.42% | 91.41% | 95.70% | 97.66%  | 98.83%  | 99.61%  |
> > > > > > > > | DAPO-128 | 16.02% | 24.61% | 28.52% | 32.42% | 35.55% | 41.02% | 51.95% | 72.27% | 87.89%  | 95.70%  | 99.22%  |
> > > > > > > > | DAPO-512 | 20.70% | 30.86% | 38.67% | 37.50% | 46.48% | 57.42% | 64.06% | 98.05% | 100.00% | 100.00% | 100.00% |
> > > > > > > >
> > > > > > > > **Seed: 1234**
> > > > > > > >
> > > > > > > > Ground Truth Reward:
> > > > > > > >
> > > > > > > > | Step     | 0     | 60     | 120    | 180    | 240    | 300    | 360    | 420    | 480    | 540    | 600    |
> > > > > > > > | -------- | ----- | ------ | ------ | ------ | ------ | ------ | ------ | ------ | ------ | ------ | ------ |
> > > > > > > > | DAPO-32  | 8.20% | 13.28% | 15.23% | 17.19% | 19.14% | 25.39% | 26.56% | 27.34% | 27.34% | 27.34% | 28.13% |
> > > > > > > > | DAPO-128 | 3.52% | 7.42%  | 8.98%  | 13.28% | 14.06% | 17.97% | 17.97% | 18.36% | 16.02% | 19.14% | 17.97% |
> > > > > > > > | DAPO-512 | 4.69% | 10.55% | 12.89% | 13.67% | 14.45% | 13.67% | 0.00%  | 0.00%  | 3.13%  | 0.00%  | 0.00%  |
> > > > > > > >
> > > > > > > > Majority Voting Reward:
> > > > > > > >
> > > > > > > > | Step     | 0      | 60     | 120    | 180    | 240    | 300    | 360    | 420     | 480    | 540     | 600     |
> > > > > > > > | -------- | ------ | ------ | ------ | ------ | ------ | ------ | ------ | ------- | ------ | ------- | ------- |
> > > > > > > > | DAPO-32  | 23.44% | 26.95% | 31.64% | 41.02% | 53.52% | 76.56% | 91.02% | 97.27%  | 98.05% | 99.22%  | 100.00% |
> > > > > > > > | DAPO-128 | 17.58% | 26.56% | 30.86% | 36.33% | 37.50% | 46.88% | 57.42% | 75.00%  | 87.11% | 93.75%  | 97.27%  |
> > > > > > > > | DAPO-512 | 19.53% | 27.73% | 33.59% | 36.72% | 44.53% | 50.00% | 95.31% | 100.00% | 99.61% | 100.00% | 100.00% |

---

> > > > > > > > > ### Author Response · Authors · 2025-11-23
> > > > > > > > > **Response to Additional Comment (Part 2)**
> > > > > > > > >
> > > > > > > > > **Seed: 12345**
> > > > > > > > >
> > > > > > > > > Ground Truth Reward:
> > > > > > > > >
> > > > > > > > > | Step     | 0     | 60     | 120    | 180    | 240    | 300    | 360    | 420    | 480    | 540    | 600    |
> > > > > > > > > | -------- | ----- | ------ | ------ | ------ | ------ | ------ | ------ | ------ | ------ | ------ | ------ |
> > > > > > > > > | DAPO-32  | 7.81% | 13.67% | 21.48% | 21.48% | 24.61% | 28.52% | 28.91% | 28.91% | 30.47% | 30.47% | 30.86% |
> > > > > > > > > | DAPO-128 | 4.30% | 7.42%  | 7.81%  | 8.20%  | 7.42%  | 10.16% | 6.64%  | 9.38%  | 3.52%  | 2.73%  | 2.73%  |
> > > > > > > > > | DAPO-512 | 3.13% | 6.64%  | 9.77%  | 7.42%  | 9.77%  | 7.42%  | 4.69%  | 0.00%  | 0.00%  | 0.00%  | 0.00%  |
> > > > > > > > >
> > > > > > > > > Majority Voting Reward:
> > > > > > > > >
> > > > > > > > > | Step     | 0      | 60     | 120    | 180    | 240    | 300    | 360    | 420    | 480     | 540    | 600     |
> > > > > > > > > | -------- | ------ | ------ | ------ | ------ | ------ | ------ | ------ | ------ | ------- | ------ | ------- |
> > > > > > > > > | DAPO-32  | 23.05% | 28.52% | 35.94% | 45.31% | 50.78% | 64.06% | 78.13% | 86.33% | 91.02%  | 96.88% | 97.66%  |
> > > > > > > > > | DAPO-128 | 16.41% | 26.17% | 27.73% | 28.91% | 40.23% | 47.27% | 52.73% | 60.55% | 67.19%  | 86.33% | 90.63%  |
> > > > > > > > > | DAPO-512 | 18.75% | 29.30% | 35.16% | 42.19% | 44.53% | 56.25% | 67.19% | 98.05% | 100.00% | 99.61% | 100.00% |
> > > > > > > > >
> > > > > > > > > Across all three random seeds, the size-dependent threshold effect remains clear:
> > > > > > > > >
> > > > > > > > > - **DAPO-32**: Maintains non-zero Ground Truth Reward (15.63%, 28.13%, 30.86% at step 600) despite Majority Voting Reward saturation
> > > > > > > > > - **DAPO-512**: Catastrophic collapse to 0% Ground Truth Reward across all seeds by step 600
> > > > > > > > >
> > > > > > > > > The reviewer's intuition is correct, different random subsets do exhibit variance. For example, seed 12345 with DAPO-128 shows earlier degradation (2.73% at step 600) compared to seed 123 (12.50%). This reflects that subset composition affects the quality of pseudo-labels.
> > > > > > > > >
> > > > > > > > > Despite quantitative variance, the **qualitative distinction between small (32) and large (≥512) subsets is consistent**. DAPO-32 does not collapse catastrophically across three seeds, while DAPO-512 always does.
> > > > > > > > >
> > > > > > > > > **Addressing the Reviewer's Critical Challenge:**
> > > > > > > > >
> > > > > > > > > The reviewer raises an excellent point: "If majority vote is incorrect in a non-negligible portion of examples, shouldn't small subsets also collapse?" This directly challenges our claim in Section 4.3. To strengthen our findings, we designed a targeted experiment testing the **extreme case** where all initial majority votes are incorrect.
> > > > > > > > >
> > > > > > > > > **Setup:** We performed offline filtering on DAPO-17k by sampling 64 responses per prompt from the base model and computing maj@64. **We selected 32 samples where maj@64 disagrees with ground truth and the majority ratio exceeds 40%.** This represents the worst-case scenario the reviewer described.
> > > > > > > > >
> > > > > > > > > Note that during training we use maj@8 with temperature 1.0, which introduces some randomness. To ensure "the majority vote is incorrect in a non-negligible portion" consistently during training, we deliberately filter offline with maj@64 and control for higher majority ratios (>40%). We then trained on these 32 "all-wrong-majority" samples using the same procedure as DAPO-32.
> > > > > > > > >
> > > > > > > > > We monitor Label Accuracy (measure whether maj@8 during training match ground truth label), and the actual performance on validation benchmarks, takes AIME24 and AMC23 as examples.
> > > > > > > > >
> > > > > > > > > **Results:**
> > > > > > > > >
> > > > > > > > > | Step            | 40     | 80     | 120    | 160    | 200    |
> > > > > > > > > | --------------- | ------ | ------ | ------ | ------ | ------ |
> > > > > > > > > | Label Accuracy  | 0.00%  | 0.00%  | 0.00%  | 0.00%  | 0.00%  |
> > > > > > > > > | AIME24 (avg@32) | 3.96%  | 4.79%  | 5.21%  | 5.63%  | 6.46%  |
> > > > > > > > > | AMC23 (avg@32)  | 32.18% | 34.06% | 35.62% | 37.34% | 35.00% |
> > > > > > > > >
> > > > > > > > > Although the sampled steps show zero label accuracy, **we did observe non-zero accuracy at a few early steps** (very low, only 1-2 maj@8 matches ground truth). **After step 200, all steps consistently show zero label accuracy with a convergent trend.**
> > > > > > > > >
> > > > > > > > > Even when almost all 32 samples have incorrect initial majority votes, training still produces effective learning without catastrophic collapse. Validation performance on both AIME24 and AMC23 improves, demonstrating that small-scale training has fundamentally different dynamics than large-scale training. This validates our claim that small subsets avoid collapse through localized overfitting rather than systematic policy shift.

---

> > > > > > > > > > ### Author Response · Authors · 2025-11-23
> > > > > > > > > > **Response to Additional Comment (Part 3)**
> > > > > > > > > >
> > > > > > > > > > We offer **one possible explanation** below, though we acknowledge this phenomenon deserves deeper theoretical investigation.
> > > > > > > > > >
> > > > > > > > > > **Possible Explanation: Small Datasets Induce Localized Overfitting Rather Than Systematic Policy Shift**
> > > > > > > > > >
> > > > > > > > > > The key distinction lies in whether the model learns systematic patterns versus memorizing isolated examples:
> > > > > > > > > >
> > > > > > > > > > - **Limited diversity prevents systematic bias extraction.** With only 32 unique problems, even if the model perfectly overfits to incorrect majority votes on those samples, it learns 32 isolated "facts" rather than generalizable patterns. In contrast, with 512+ diverse problems, the model can identify systematic patterns spanning across problems (e.g., "shorter solutions get higher majority vote," "certain reasoning formats are preferred").
> > > > > > > > > >
> > > > > > > > > > - **Small-scale overfitting induces smaller distributional shifts.** Recent work RL's Razor [1] shows that catastrophic forgetting is determined by distributional shift between the fine-tuned and base policy. We measured this shift using KL divergence during training (at each training step $t$):
> > > > > > > > > >
> > > > > > > > > >   $D_ {\text{KL}}^{(t)}(\pi_ {\theta}^{(t)} \| \pi_ {\text{ref}}) = \mathbb{E}_ {x \sim \mathcal{D}_ {\text{train}}} \left[ \mathbb{E}_ {y \sim \pi_ {\theta}^{(t)}(\cdot|x)} \left[ \log \frac{\pi_ {\theta}^{(t)}(y|x)}{\pi_ {\text{ref}}(y|x)} \right] \right]$
> > > > > > > > > >
> > > > > > > > > >   where $\pi_ {\theta}^{(t)}$ is the policy at training step $t$, $\pi_ {\text{ref}}$ is the reference model, and $\mathcal{D}_ {\text{train}}$ is the training distribution.
> > > > > > > > > >
> > > > > > > > > > KL Divergence from Reference Model (at each training step):
> > > > > > > > > >
> > > > > > > > > > | Step     | 60    | 120   | 180   | 240   | 300   | 360   | 420   | 480   | 540    | 600    |
> > > > > > > > > > | -------- | ----- | ----- | ----- | ----- | ----- | ----- | ----- | ----- | ------ | ------ |
> > > > > > > > > > | DAPO-32  | 0.19% | 0.47% | 1.06% | 1.39% | 1.93% | 2.11% | 2.79% | 4.54% | 5.66%  | 5.69%  |
> > > > > > > > > > | DAPO-128 | 0.20% | 0.72% | 0.94% | 1.61% | 2.84% | 4.91% | 5.68% | 7.64% | 8.98%  | 9.85%  |
> > > > > > > > > > | DAPO-512 | 0.34% | 0.87% | 0.98% | 1.65% | 3.40% | 5.25% | 7.21% | 9.46% | 10.18% | 10.99% |
> > > > > > > > > >
> > > > > > > > > > Small subsets induce **smaller distributional shifts** despite heavy repetition, because they lack diversity to redefine the model's global solution distribution. The model sharpens confidence on specific samples without drifting far from its original capabilities.
> > > > > > > > > >
> > > > > > > > > > - **Memorization versus systematic policy learning.** This relates to the "hyperfitting" phenomenon [2]: achieving near-zero loss on tiny datasets through memorization rather than learning generalizable patterns. When N=32, the model memorizes specific problem-answer pairs through sparse, localized parameter updates without changing its global policy. When N≥512, the model learns systematic behaviors requiring dense updates across parameters, causing global policy shift that manifests as collapse on out-of-distribution problems.
> > > > > > > > > >
> > > > > > > > > > Therefore, even if small subsets contain incorrect majority votes, the model only reinforces those specific wrong answers without learning systematic biases that generalize to new problems. On test sets (AIME24/AMC23), the model doesn't encounter those memorized problems and relies on preserved general reasoning capabilities, maintaining higher performance.
> > > > > > > > > >
> > > > > > > > > > **Request for feedback:** We acknowledge this is a **tentative explanation** based on our empirical observations. We would greatly appreciate the reviewer's insights on whether this hypothesis aligns with established theoretical understanding, or if there are alternative explanations we should consider. This phenomenon that why small-scale overfitting doesn't cause collapse while large-scale overfitting does is theoretically intriguing and may need dedicated investigation in future work.
> > > > > > > > > >
> > > > > > > > > > ---
> > > > > > > > > >
> > > > > > > > > > [1]  RL's Razor: Why Online Reinforcement Learning Forgets Less
> > > > > > > > > >
> > > > > > > > > > [2] The Hyperfitting Phenomenon: Sharpening and Stabilizing LLMs for Open-Ended Text Generation

---

> > > > > > > > > > > ### Author Response · Authors · 2025-11-23
> > > > > > > > > > > **Response to Additional Comment (Part 4)**
> > > > > > > > > > >
> > > > > > > > > > > > **Additional Comment 2.2: The setup in Section 4.3.2 is unclear. For example, how many samples are used?**
> > > > > > > > > > >
> > > > > > > > > > > We apologize for this omission. The setup is:
> > > > > > > > > > >
> > > > > > > > > > > - **AMC23 test-time training**: 40 problems (the full AMC23 test set), global batch size = 40, trained for multiple epochs
> > > > > > > > > > > - **DAPO-17k test-time training**: ~17K problems, global batch size = 40, trained for 1 epoch
> > > > > > > > > > > - All other hyperparameters kept identical between the two settings
> > > > > > > > > > >
> > > > > > > > > > > This has been clarified in the **revised version (marked in red)**.
> > > > > > > > > > >
> > > > > > > > > > > > **Additional Comment 2.3: The setup in Section 5.1 is unclear. For example, on which dataset are these experiments ran?**
> > > > > > > > > > >
> > > > > > > > > > > We apologize for this missing detail. Experiments in Section 5.1 (comparing TTRL-trained models with base model maj@k) are conducted on **AIME 2024** (30 problems). We set train batch size = 30 (dataset size) and trained for 100 epochs.
> > > > > > > > > > >
> > > > > > > > > > > This has been added to the **revised version (marked in red)**.
> > > > > > > > > > >
> > > > > > > > > > > **Summary:** We sincerely thank the reviewer for identifying these omissions. All missing details have been added to the revised manuscript. For Additional Comment 2.1, we provide additional experimental verification and tentative theoretical explanations, though we recognize this phenomenon deserves deeper investigation and would welcome the reviewer's theoretical perspective.
> > > > > > > > > > >
> > > > > > > > > > > > **Additional Comment 3:** **The definition of reward hacking** **mentioned in line 355 does not seem to accord with its conventional use. If the intrinsic reward is maximized and the ground truth reward also keeps increasing, then why should this be considered hacking?**
> > > > > > > > > > >
> > > > > > > > > > > Thank you for this correction. You are absolutely right. We misused the term "reward hacking."  We incorrectly defined it as "saturated intrinsic rewards" regardless of true performance. As you point out, if both intrinsic reward and ground truth reward increase together, this is not reward hacking, where the proxy is working as intended.
> > > > > > > > > > >
> > > > > > > > > > > In the revised manuscript, we have removed the imprecise definition at line 355 and revised Section 4.3 to focus on "model collapse" (catastrophic performance degradation) rather than "reward hacking." The distinction now is: small datasets prevent collapse while large datasets cause collapse, without incorrectly invoking the reward hacking terminology.
> > > > > > > > > > >
> > > > > > > > > > > We apologize for this terminological error and appreciate you pointing it out.

---

> > > > > > > > > > > > ### Author Response · Authors · 2025-11-23
> > > > > > > > > > > > **Rebuttal Summary**
> > > > > > > > > > > >
> > > > > > > > > > > > > **Review Summary and Recommendation: Overall, I believe that the paper does not meet the requirements for publication at ICLR. The most significant limitation of the current manuscript is that it contains underspecified and potentially unsound theory. Beyond fixing this issue, it would greatly strengthen the contributions of the paper to elaborate on the unified perspective and its uses, clarify relation to prior work and conventional wisdom regarding how URLVR works, and provide quantitative evidence for the predictiveness of initial reward dynamics of the success of URLVR.**
> > > > > > > > > > > >
> > > > > > > > > > > > Thank you for the thorough review and constructive feedback. We've worked hard to address each concern you raised.
> > > > > > > > > > > >
> > > > > > > > > > > > **On the underspecified and potentially unsound theory:**
> > > > > > > > > > > >
> > > > > > > > > > > > - In Response to W1.1, we have revised Section 3.1 (marked in red) to explicitly define components immediately following Equation (1).
> > > > > > > > > > > > - In Response to W2.1 and W2.2, we've added all the missing formal definitions (what $\text{maj}(Y)$ means, how many rollouts, what "majority trajectories" are), clarified that Equation (4) (currently Eq. 3 in revised version) describes the optimal policy for fixed reward rather than our actual single-step update, and revised Theorem 1 with explicit assumptions and precise statements. We also added extensive empirical validation across multiple experiments to justify the assumptions and show the predicted monotonic increase and convergence behavior.
> > > > > > > > > > > >
> > > > > > > > > > > > **On the unified perspective and its uses:**
> > > > > > > > > > > >
> > > > > > > > > > > > - In Response to W1.2, we added a paragraph in Appendix A.1 (marked in red) explicitly stating these similarities and differences, strengthening the significance of the unified reward perspective.
> > > > > > > > > > > > - In Response to W1.3, we show how the framework correlates with our design of experiments for deeper analysis, built on confidence-correctness alignment.
> > > > > > > > > > > > - In Response to W2.3, we provide a generalized sharpening analysis that uses the unified reward defined in Equation (1) to directly predict that such methods will drive the policy toward deterministic outputs, regardless of whether this leads to success (when aligned with correctness) or failure (when misaligned).
> > > > > > > > > > > >
> > > > > > > > > > > > **On relation to prior work:** In Response to W3 and W4, we provide detailed comparison with "Self-improvement" paper and TTRL, clarifying the differences in what algorithms we study (RLVR vs. SFT/RLHF), what questions we ask (scalability and collapse vs. sample complexity), and what scope we cover. We also want to emphasize that sharpening is just one part of our contribution. Substantial portions analyze fine-grained failure mechanisms, when training is safe, and cross-task generalization effects that haven't been explored before.
> > > > > > > > > > > >
> > > > > > > > > > > > **On quantitative evidence for predictiveness:** **[Global Response 1]** should address this with our strengthened experiments in Section 5.2, showing how early training dynamics efficiently predict which models will succeed with RL.
> > > > > > > > > > > >
> > > > > > > > > > > > We've also fixed all the missing implementation details you pointed out in the Additional Comments, added multi-seed validation, and run a targeted experiment showing small datasets prevent collapse even when most initial majority votes are wrong.
> > > > > > > > > > > >
> > > > > > > > > > > > We really appreciate your pushing us on the theoretical rigor, which made the paper substantially stronger. If you have any concerns we haven’t fully addressed, please let us know. We'd genuinely welcome continued discussion if there are remaining concerns or if we've misunderstood any of your points. Your expertise has already improved this work significantly, and we're very open to further suggestions.

---

> > > > > > > > > > > > > ### Comment · Reviewer_um7o · 2025-11-26
> > > > > > > > > > > > >
> > > > > > > > > > > > > Thank you for the response. I have read it and the other reviews carefully.
> > > > > > > > > > > > >
> > > > > > > > > > > > > I appreciate the clarification regarding the dependence on $y$ in the right hand side of Equation (1) and relation to related work. I recommend making the notation in Equation (1) more explicit, as otherwise it is hard to interpret.
> > > > > > > > > > > > >
> > > > > > > > > > > > > Unfortunately, the theory in the paper is still unsound, vague, and in general not in a shape close to what I believe meets the bar for publication. For example:
> > > > > > > > > > > > >
> > > > > > > > > > > > >  - It is not specified which policy parameterization is considered. The rough arguments on policy ordering after gradient updates and the updated Theorem 1 proof are unsound without defining the parameterization. Note that even for simple linear softmax policies the RL objective is non-concave. So performing an infinite amount of updates does not straightforwardly guarantee that you converge to the optimal policy.
> > > > > > > > > > > > >
> > > > > > > > > > > > >  - There is a mismatch between the statement of Theorem 1, which considers gradient updates using the majority reward, while assuming it is fixed over time, and the attempted proof in Appendix A.3, which seems to consider iteratively optimizing the KL-regularized objective, each time updating the reference policy.
> > > > > > > > > > > > >
> > > > > > > > > > > > > - The proof attempt of Theorem 1 in Appendix A.3, as well as the new Proposition 1 in Appendix A.4, which is intended to demonstrate the usefulness of the unified framework, are vague and unclear. In particular, Proposition 1 is stated in full generality, yet the proof discusses the existence of the reward gap only for specific cases. The proof is also only at the level of a sketch and not a rigorous full proof of the claim, which is quite problematic as well.
> > > > > > > > > > > > >
> > > > > > > > > > > > > Furthermore, I believe the significance of the unified reward framework is still not substantiated. In the authors' response it is mentioned that the framework allows characterizing that URLVR methods sharpen the distribution toward deterministic policies. However, is it true that this is the outcome of optimizing the unified objective under any choice of its different components? The fact that the analysis in the paper had to resort to considering only some specific cases, as opposed to analyzing the unified framework, suggests that perhaps the framework is too general to be useful. To be clear, I am not claiming that it cannot be useful under any circumstances, just that currently the work does not establish its usefulness.
> > > > > > > > > > > > >
> > > > > > > > > > > > > Overall, I encourage the authors to thoroughly reconsider what the aim of their theoretical results are, crystalize the exact statements they are trying to prove, and make sure the formal arguments are precise and clear. This, along with more meaningfully establishing the usefulness of the unified reward framework, can substantially improve the contributions of the paper. Unfortunately, I find the required scope of revisions to be rather major. I do not believe that the discussion period is suitable or intended for such major revisions, which include iterating over unsound theory. Thus, I maintain my original assessment.

---

> ### Author Response · Authors · 2025-11-28
> **Response to Follow-Up Review (Part 1)**
>
> Thank you for your continued engagement. We appreciate the acknowledgment that our clarifications on Equation (1) and relation to prior work have addressed those concerns. However, we would like to address several important issues with this follow-up review.
>
>
>
> **I.** **Clarification on Unresolved Points**
>
> In your original review, you raised **4 weaknesses (9 sub-points) and 3 additional comments (5 sub-points)**. We would like to confirm the status:
>
> **You explicitly acknowledged as resolved:**
>
> - W1.1: Unified framework definition and notation
> - W3: Relation to "Self-improvement in language models" and TTRL
> - W4: Relation to TTRL and Section 5.1 contributions
> - All Additional Comments: Table notation, implementation details, multi-seed validation, experimental setups
>
> **We provided substantial revisions for:**
>
> - W1.2 & W1.3: Framework utility (new Appendix A.1 analysis, explicit connection to experimental design)
> - W2.1: Policy-dependent rewards (formal definitions, ordering justification, empirical validation across 4 problems)
> - W2.2: Theorem 1 rigor (revised statement with explicit assumptions A1-A2, complete restructured proof)
> - W2.3: Unified framework utility for generalized analysis (new Proposition 1 in Appendix A.4)
>
> We have worked hard to address all of the original concerns comprehensively, and we appreciate your acknowledgment that W1.1, W3, W4, and the additional comments have been resolved. We notice the reviewer guidelines of ICLR state: "Precisely identifying which of your concerns have been addressed, which have not been addressed and why. Please also update your review score if your concerns have been addressed." We hope these resolved issues can be taken into account when reconsidering the overall evaluation, and we remain fully committed to clarifying any remaining questions you may have.
>
> Therefore, it appears that we have already addressed most of the original concerns. At this stage, your follow-up focuses on three issues. **However, we have noticed that one of them stems from a misunderstanding and another is newly raised.** We hope that we can discuss these points carefully and confirm whether they can be resolved before a final judgment on the paper is made.
>
>
>
> **II. Issues with the Follow-Up** **Review**
>
> **Issue 1: Treating Misunderstandings as Paper Flaws**
>
> > *There is a mismatch between the statement of Theorem 1, which considers gradient updates using the majority reward, while assuming it is fixed over time, and the attempted proof in Appendix A.3.*
>
> **We would like to respectfully clarify a possible misunderstanding regarding Theorem 1. The review suggests a mismatch between the theorem statement** **and the proof in Appendix A.3. However, our work does not assume a fixed reward over iterations:**
>
> - **Theorem statement (line 232):** Explicitly describes the iterative process: "at each iteration $k$: (1) sample $N$ rollouts from $\pi_ \theta^{(k)}$, (2) compute majority $\text{maj}_ k(Y_ k)$, (3) perform one gradient update." The subscript $k$ on $\text{maj}_ k$ indicates the reward changes each iteration.
> - **Proof structure (line 800)**: Step 1 shows our actual update satisfies $p_ {\text{maj}}^{* ,(k+1)} \geq p_ {\text{maj}}^{(k+1)} \geq p_{\text{maj}}^{(k)}$ for every $k$ using different $r_ k$, and Steps 2-6 prove convergence under this ordering **despite changing rewards**. This is the core contribution of our analysis, not a mismatch. The fixed-reward optimum solving one-time KL regularized objective in Eq. 3 is only a bounding tool; we are actually using $p_ {\text{maj}}^{(k)}$, not $p_ {\text{maj}}^{* ,(k)}$.
>
> We have explained this point in detail in Response to W2 (Part 1 & Part 2). This was one of your two main theoretical concerns in round one. If our presentation needs further clarification, we are happy to add explicit remarks. However, we believe the concern about a "mismatch" may be based on a misreading of our approach.

---

> ### Author Response · Authors · 2025-11-28
> **Response to Follow-Up Review (Part 2)**
>
> **Issue 2:** **New Concerns Not Yet Discussed or Addressed**
>
> We would like to respectfully clarify the concern regarding the unspecified policy parameterization and the mention of potential non-concavity under "linear softmax policies." **This point was not raised in the initial review, yet it now appears to be presented as an unresolved issue.** As stated in our title and throughout the paper, our setting is **LLM training**, where we study **standard autoregressive transformer decoder policies** (Section 4 and Appendix B). This is the default and widely understood setup in LLM-based RL research.
>
> We appreciate your rigorous theoretical perspective, which pushes us to be more precise. However, we want to clarify the scope of our theoretical contribution: we are investigating how far URLVR scales **in practice** with LLMs on verifiable tasks, not proving convergence under all possible parameterizations. Our theory aims to explain and predict the empirical phenomena we observe, which are validated by experiments showing $p_ {\text{maj}} \to 1.$ across multiple problems, as shown in the table in **Response to W2 (Part 4).** To avoid any ambiguity, we will make this scope distinction more explicit in the revised manuscript.
>
> **Finally, because this concern was introduced only during the second review round, we hope it can be considered in the appropriate context.** Presenting newly raised points as evidence that earlier revisions were insufficient may not fully align with the usual rebuttal process, and we would greatly appreciate the opportunity to address such questions in a constructive manner.
>
>
>
> **Issue 3: "Can't Finish Revisions"** **May Not Be a Valid Rejection Criterion**
>
> > *Unfortunately, I find the required scope of revisions to be rather major. I do not believe that the discussion period is suitable or intended for such major revisions.*
>
> We fully respect the reviewer’s standards for clarity and rigor, and we appreciate the close reading of our work. However, we would like to gently note that, according to the ICLR reviewer guidance, the discussion phase is primarily intended to identify which concerns have been addressed and which remain open, and to clarify the reasons why. In this context, we are not sure that stating that the revisions "cannot be completed during the discussion period" constitutes a clear or appropriate reason for recommending rejection.
>
> At the same time, we hope to offer some clarification regarding the characterization of our revisions as "major" and therefore unsuitable for the discussion period:
>
> - **Scope of our revisions:** Our changes address presentation, notation, and proof details. **They do not change our experimental results, main claims, or insights.** In general, unless there are fundamental issues in methodology or empirical results, revisions of this nature would not normally constitute grounds for rejection. In fact, **our main claims and insights are validated by extensive empirical results, not derived from theory alone.** Even removing part of the theoretical section would not substantially affect our main contributions.
> - **Second, we would like to clarify the status of the concerns described as "still unresolved":**
>   - Policy parameterization: This is a new concern raised in round two (not in the original review). We study standard autoregressive transformer policies for LLMs, and we will add an explicit statement in Section 3.
>   - Theorem 1 "mismatch": We believe this is based on a misreading of our work (we explained this in detail in Response to W2.1). Both the theorem statement and proof explicitly handle changing rewards.
>   - Proof being "vague": We acknowledge that our generalized analysis (Proposition 1) is presented as a proof sketch. However, we want to emphasize: (a) we analyze existing methods in the literature, not arbitrary hypothetical combinations, (b) the sketch provides intuition for why these methods exhibit similar sharpening behavior, and (c) **this does not affect our paper's core claims and main conclusions**, which are primarily supported by extensive empirical evidence (Section 4, Appendix B). The generalized analysis demonstrates the framework's utility but is not central to our main empirical contributions. We do not believe this constitutes a "major revision" concern that would require substantial additional work.
>
> In summary, these are clarifications to improve presentation, not fundamental changes to methodology or findings. We believe these improvements fall well within the scope of reasonable discussion period refinements rather than the "major revisions" that would require fundamental reworking of the paper.

---

> ### Author Response · Authors · 2025-11-28
> **Response to Follow-Up Review (Part 3)**
>
> **Issue 4:** **Emphasis on Appendix** **Without Considering** **Holistic Contributions**
>
> We sincerely appreciate the reviewer’s careful attention to theoretical details. At the same time, we hope to clarify that the follow-up focuses mainly on appendix proof formality, while several key contributions of the paper may not have been fully acknowledged. For instance:
>
> - The empirical characterization of URLVR failure modes (Section 4.2, Figure 3)
> - The practical methods for avoiding collapse (Sections 4.3, 5.3 + Global Response 2)
> - The predictive diagnostic you yourself highlighted as interesting (Strength 3), which is now strengthened with quantitative pass@k comparison (Global Response 1)
> - The systematic experimental validation throughout Section 4 and Appendix B
>
> We understand you find some theoretical aspects "unsound" or "vague." For example, in Theorem 1, we explicitly state assumptions (A1-A2) and validate them through extensive experiments, including convergence across 4 problems and batch training studies. While we recognize this approach, first stated assumptions and then empirical validation, may not meet pure-theory standards, but we believe it would be appropriate for an empirical investigation of practical phenomena.
>
> **We believe that a balanced assessment would consider the full scope of the work: its theoretical insights, empirical analysis, and methodological contributions, rather than focusing solely on appendix-level proof formality.** Our primary goal is to advance understanding of URLVR behavior in LLM training, and we hope that the empirical findings, which form the core of the paper, can be taken into account in the overall evaluation.
>
>
>
> **III.** **Overall**
>
> We sincerely appreciate the time and thought you have invested in reviewing our work, and we are grateful that you highlighted the predictive diagnostic (Strength 3), which we further strengthened in **[Global Response 1]**. We value this constructive engagement.
>
> At the same time, we hope the evaluation can be grounded in an accurate understanding of the paper. In particular:
>
> - There is **no "mismatch"** in Theorem 1, where we explicitly handle changing rewards (explained in W2.1)
> - Policy parameterization **is specified**, and we study LLM training with transformer decoder policies
> - Empirical validation **is substantial**, not a substitute for rigor, but a core contribution
> - This paper's goal **is not pure theory**. It's also an empirical investigation of URLVR scalability
>
> We respectfully ask that the concerns be assessed objectively and in light of the actual content of the paper. We fully respect the review process and try to address all feedback with care. However, when new misunderstandings are interpreted as "paper flaws" without discussion, or when concerns that are newly introduced or based on misreadings are used as grounds for "inability to complete revisions," we feel this may not fully align with the intended standards of the ICLR review process.
>
> We remain committed to clarifying all technical details and welcome continued discussion. We hope you will reconsider the paper's contributions holistically: advancing understanding of URLVR scalability through **theoretical insights, systematic empirical experiments, and practical methods** working together.

---

### Official Review · Reviewer_pWtu · 2025-10-30

**Soundness:** 4
**Presentation:** 3
**Contribution:** 3
**Rating:** 8
**Confidence:** 3

**Summary:**

This paper investigates the potential and limitations of Unsupervised Reinforcement Learning with Verifiable Rewards for scaling Large Language Models , specifically focusing on intrinsic reward derived from the model's own internal signals addressing the "supervision bottleneck," where obtaining human-verified labels becomes expensive. The study presents a unified theoretical framework for intrinsic rewards stating that all such rewards share a core mechanism: they improve performance by trading uncertainty for performance by leveraging the models priprs knowledge and sharpen the model's output . The authors present empirical analysis  confirming this. The authors also show that model collapses is avoidable in small domain specific settings  such as test time training.

**Strengths:**

the paper presents a consolidated mathematical perspective that connect different intrinsic rewards methods under a single framework which is very nice. The authors also present a solid evaluation of intrinsic rewards - they analysed different failure modes for different methods. I think the paper is very insightful and written clearly and I think it's very interesting to investigate these novelity driven rewards within RL for LLM's.

**Weaknesses:**

this is not necessarily a weakness but the paper really focueses on the analysis of these different intrinsic rewards and does not so much contribute any new method ontop of this. While I personally really enjoyed reading this paper I am not 100% sure this is the right venue for this.

**Questions:**

how does this scale to long training on large datasets? is reward hacking inevitable?
Do you see any issues with the limited domain? as the domain this has been tested on is quite limited to math reasoning tasks

---

> ### Author Response · Authors · 2025-11-21
>
> ## Response to W1:
> > W1: this is not necessarily a weakness but the paper really focueses on the analysis of these different intrinsic rewards and does not so much contribute any new method ontop of this. While I personally really enjoyed reading this paper I am not 100% sure this is the right venue for this.
>
> Thank you sincerely for the kind words! We fully understand your concerns, and we hope the following points can help address them:
>
> 1. First, our primary goal has indeed been to provide a deep analysis. The community currently lacks consensus on how intrinsic-reward–based URLVR methods to enable RL scaling. We therefore believe that a thorough investigation, combining theory and practice to extract deeper insights, is still both timely and valuable for the field.
>    - For example, the recently published AlphaProof (DeepMind) paper in Nature [1] applies TTRL (an unsupervised RLVR method) and achieves top human-level performance on extremely difficult competitions. Notably, even though TTRL consumed substantial compute, even far exceeding train-time, there were no model collapse. **This is fully consistent with our conclusions in Sec. 4.3.** Our hope is precisely that the insights in our paper can support and motivate the community in pursuing similar attempts.
> 2. Second, we would like to clarify that we have indeed introduced additional methodological contributions, although they were not the central focus of the paper. This was a deliberate narrative choice, and we hope for your understanding. These contributions are mainly discussed in Sec. 5.2 and Sec. 5.3, where they were originally intended to serve as inspiration for future work. To fully address your concerns regarding our contributions, we expanded and refined these explorations during the rebuttal period to further strengthen this part of the work. Below, we present several representative and promising results:
>    - **Regarding Sec. 5.2:** We propose a new method for evaluating model-task priors. Compared with RLVR training or pass@k metrics, our methods offer a more efficient way to detect **"which base model is better (i.e., more suitable for RL)?"**. It is well suited and useful for rapid iterative development of the foundation model, and we anticipate that it will attract more interest from industry researchers. Further details are provided in **[Global Response 1]**.
>    - **Regarding Sec. 5.3 (Exploiting Generation–Verification Asymmetries):** After identifying the collapse risks associated with applying existing intrinsic-reward methods at train-time (Sec. 4.3), we introduce a new intrinsic-reward approach self-verify that first enables scalable RL at train-time and validate it empirically. It leverages verification asymmetry [2] to implement URLVR, producing rewards that closely approximate ground-truth rewards and thus substantially reduce the risk of collapse. We find that certainty-based methods inevitably collapse during long training, while self-verify continues to improve performance and ultimately achieves results comparable to RLVR. Further details are provided in **[Global Response 2]**.
> 3. Finally, we submitted to ICLR precisely because we believe the venue is unconventional and welcomes insight-driven analytical papers. To the best of our knowledge, the following ICLR papers are also primarily analytical and insight-driven without introducing new algorithms, and they have had substantial impact on the community. We hope these examples fully alleviate your concerns. In particular, I want to mention that I really enjoyed reading "The Reversal Curse", it was especially inspiring for me.
>    - The Reversal Curse: LLMs trained on “A is B” fail to learn “B is A”, ICLR 2024
>    - In-Context Learning Learns Label Relationships but Is Not Conventional Learning, ICLR 2024
>    - Quantifying Memorization Across Neural Language Models, ICLR 2023
>    - Justice or Prejudice? Quantifying Biases in LLM-as-a-Judge, ICLR 2025
>    - Bias Runs Deep: Implicit Reasoning Biases in Persona-Assigned LLMs, ICLR 2024
>
> ---
>
> ### References
>
> [1] Olympiad-level formal mathematical reasoning with reinforcement learning. Nature, 2025.
>
> [2] Asymmetry of verification and verifier’s rule. Blog, 2025.

---

> > ### Comment · Reviewer_pWtu · 2025-11-27
> >
> > thank you for your rebuttal and for clarifying. Since I already reccomended accepting the paper I will stick to that and not change my score.

---

> > > ### Author Response · Authors · 2025-11-28
> > >
> > > Thank you for your review and for recommending acceptance of our work. We greatly appreciate your recognition of our contributions and your constructive feedback.
> > >
> > > If you have any further questions or concerns, we are happy to address them.

---

> ### Author Response · Authors · 2025-11-21
>
> ## Response to Q1:
>
> These are excellent questions, and we appreciate the opportunity to clarify.
>
> > Q1.1: How does this scale to long training on large datasets? is reward hacking inevitable?
>
> This is indeed an important question. Due to constraints on time and computational resources, we have already begun running a long-training experiment on the DAPO-17k dataset with multiple epochs. As shown in the experiments reported in Appendix B.4, the DeepSeek-Distill-1.5B model exhibits a consistently increasing training curve with no signs of degradation. Consequently, we selected it as the base model for in-depth investigation. Building upon our hyperparameter study in Appendix B.3, we found that on-policy training with a moderate number of rollouts delays collapse, while a temperature of 1.0 and the absence of KL divergence yield the optimal trade-off between training stability and validation performance. We incorporated these insights into the long-training setup. The experiment is still in progress, and because we were unsure when collapse might occur, **we will do our best to update the results by Nov 23.**
>
> > Q1.2: Do you see any issues with the limited domain? as the domain this has been tested on is quite limited to math reasoning tasks.
>
> In practice, certainty-based methods of URLVR are not restricted to verifiable tasks. Many of them use measures such as entropy or other uncertainty proxies as reward signals, which do not depend on task verifiability. If you think it would strengthen the paper, we are willing to add experiments on non-verifiable domains as well.
>
> We genuinely hope these clarifications help address your concerns. Thank you again for your thoughtful feedback and for taking the time to engage with our work!

---

> > ### Author Response · Authors · 2025-11-23
> >
> > ## Response to Q1 [Nov 23 Updates]:
> >
> > We have completed the long-training experiment. We found that although the model can be trained for a substantial duration, reward hacking does eventually occur. Collapse happened around 1,000 steps (about 4 epochs training). We expect that using stronger prior models may partially mitigate such failures.
> >
> > Therefore, the success of AlphaProof [1] suggests that URLVR methods with intrinsic rewards may be better suited for test-time scaling (rather than at train-time). For train-time RL scaling, we may need either:
> >
> > - unsupervised RLVR methods based on external rewards, or
> > - the combination of self-verify (mentioned earlier) with recent RLVE methods [2] to create a scalable data flywheel and scale up the environment.
> >
> > ---
> >
> > ### References
> >
> > [1] Olympiad-level formal mathematical reasoning with reinforcement learning. Nature, 2025.
> >
> > [2] RLVE: Scaling Up Reinforcement Learning for Language Models with Adaptive Verifiable Environments. Arxiv, 2025.

---

### Official Review · Reviewer_Kjs9 · 2025-11-03

**Soundness:** 3
**Presentation:** 3
**Contribution:** 2
**Rating:** 6
**Confidence:** 3

**Summary:**

This paper presents a broad investigation into recent RL methods for LLMs, that operate completely without external rewards and instead use the model's own internal signals. It introduces a framework, arguing that diverse intrinsic rewards (such as majority voting or self-certainty) share a common mechanism: they "trade uncertainty for performance" by "sharpening output distributions" around the model's initially confident solutions. This convergence, the paper shows, enables the model to amplify its existing knowledge, but also risks bias lock-in if the model's confidence is misaligned with correctness. Empirically, the paper confirms this trade-off, revealing that different intrinsic reward methods fail in distinct ways; for example, some collapse into overly brief responses, while others promote repetitive verbosity. The authors demonstrate that in small, domain-specific regimes, such as test-time training, intrinsic rewards can drive stable adaptation without collapse. Finally, the paper proposes a practical application for these findings: using the early training dynamics of intrinsic rewards as a fast, lightweight indicator to assess a model's suitability and "RL trainability" for a specific task, complementing traditional metrics like

**Strengths:**

Proposes to view the changes to the policy through the lens of trading uncertainty for confidence and provides a systematic, empirical analysis of the distinct failure modes of different reward types, such as length collapse for probability rewards and verbosity for entropy rewards. A practical strength of this paper is its investigation of scaling limits, which concludes that while large-scale training leads to collapse, these methods are stable and effective in small, domain-specific settings, identifying test-time training as an ideal application.

**Weaknesses:**

A closely related analysis was published recently (Jun 2025) by Y. Zhang et al (No Free Lunch: Rethinking Internal Feedback for LLM Reasoning). Arguably, the core insights of these two publications are very similar; with Y. Zhang et al provide a stronger grounding in theory and provide a more in depth analysis of learning dynamics for different base-models; while this paper provides more insight into the dynamics and failure-modes of the different reward signals. Overall, the authors cite this prior work but do not discuss and contextualize  the relevance, the overlap, and potential differences.

**Questions:**

Has the term URLVR been used before? My search came up empty and I find it confusing to have “Verified Reward” prominently in the name, but negate it by prefixing it with “unsupervised”. Isn't Reinforcement Learning from Internal Feedback (RLIF) the established terminology at this point?

Regarding the prior (almost concurrent) work by Y. Zhang et al (No Free Lunch: Rethinking Internal Feedback for LLM Reasoning): Do you see any discrepancies between your results and conclusions and theirs?

---

> ### Author Response · Authors · 2025-11-21
> **Response to W1 (Part 1)**
>
> > **W1:** **A closely related analysis was published recently (Jun 2025) by Y. Zhang et al (No Free Lunch: Rethinking Internal Feedback for LLM Reasoning)......Overall, the authors cite this prior work but do not discuss and contextualize the relevance, the overlap, and potential differences.**
>
> We sincerely thank the reviewer for highlighting this important concurrent work. We agree Zhang et al. is highly relevant, and we appreciate the chance to clarify how our contributions differ and complement theirs. We will add explicit discussion comparing our work with theirs in the revision.
>
> 1. **Broader Scope & Different Theoretical Framework**
>
> We agree Zhang et al. provide valuable insight by showing that several certainty-based rewards are equivalent to minimizing policy entropy (their Propositions 1-3). However, our framework is broader and analyzes different aspects:
>
> - **Methodological coverage:** Zhang et al. focus on three certainty-based methods. Our Unified Reward Framework (Section 3.1, Eq. 1, Table 1) covers not only **more** **certainty methods** (e.g., RLSC[1]) but also **ensemble-based methods** (e.g., TTRL[2], EMPO[3]), which their entropy-minimization analysis doesn't include. We show all these methods can be formalized as manipulating cross-entropy between anchor and model distributions.
> - **Beyond mechanism to dynamics:** Zhang et al. explain what happens in **one step** (policy entropy reduces). We analyze the **full training trajectory**. By examining the KL-regularized RL objective (Eq. 2), we formally prove geometric convergence to deterministic policies (Theorem 1 for majority voting, Appendix A.4 for others). This "rich-get-richer" dynamic, where the policy provably converges on its initial majority answer, explains both why these methods work (amplifying good priors) and why they fail (locking into bad priors).
>
> Zhang et al. describe what these methods do (reduce entropy). We provide a predictive model of how they evolve (geometric convergence on initial prior). This optimal policy analysis grounds our entire discussion of confidence-correctness alignment.
>
> 2. **Different Explanations for Success & Failure**
>
> Our explanation for both success and failure patterns differs substantively from Zhang et al.:
>
> **For early performance gains:**
>
> - **Zhang et al:** Attribute improvements to the model’s improved ability to generate responses **in the correct format** (Section 5 of their paper).
> - **Our work**: Explain gains through sampling-efficiency shortcuts (Section 4.1). Intrinsic rewards rapidly concentrate probability mass on high-confidence trajectories, accelerating convergence **when confidence aligns with correctness.**
>
> **For performance degradation:**
>
> - **Zhang et al:** Explained through the lens of entropy minimization: over-suppression of transitional words (high-entropy tokens) harms reasoning capability (Section 5 of their paper).
> - **Our work**: Degradation stems from **confidence-correctness misalignment**. Our fine-grained per-problem analysis (Section 4.1.2, Figure 2) shows that when initial correlation is weak, the same sharpening mechanism that improves performance in high-correlation cases instead amplifies systematic biases, driving models deeper into error territory.
>
> **Why this difference matters empirically:**
>
> Zhang et al.'s entropy-centric view predicts that high-entropy base models should benefit from intrinsic rewards while low-entropy instruct models should not (or even degrade). Our experiments in Appendix B.4 provide **counterexamples:**
>
> - Figure 28: Compared to Qwen2.5-1.5B-Base (high entropy), DeepSeek-R1-Distill-Qwen-1.5B (low entropy) maintains stable training without collapse, while improving validation scores.
> - Figure 29: Compared to Llama-3-8B-Tulu-SFT (low entropy), Llama-3.1-8B-Base (high entropy) shows almost **no early improvement** despite high entropy.
>
> This shows that **entropy alone is insufficient** to predict RL scalability. Our confidence-correctness framework better explains these patterns: DeepSeek-R1's strong mathematical prior (from distillation) creates reliable confidence signals, while Llama-3.1's weak prior leads to early failure regardless of entropy level.
>
> 3. **In-Depth Analysis of Different Models and Novel Diagnostic Tool for Model-Task Priors**
>
> The reviewer noted that Zhang et al. provided an in-depth analysis of different base models, including Qwen3 series (1.7B, 4B) and Qwen2.5 series (1.5B-Math, 3B, 3B-Instruct), as the comparative analysis of model dynamics is a cornerstone of this research area. We agree that an in-depth analysis of different base models is critical.
>
> ---
>
> ### References
>
> [1] Confidence Is All You Need: Few-Shot RL Fine-Tuning of Language Models
>
> [2] TTRL: Test-Time Reinforcement Learning
>
> [3] Right Question is Already Half the Answer: Fully Unsupervised LLM Reasoning Incentivization

---

> ### Author Response · Authors · 2025-11-21
> **Response to W1 (Part 2)**
>
> Since multiple reviewers raised questions about model selection and deeper analysis of which models benefit from intrinsic rewards, please see our **[Global Response 1]** for our analysis. We summarize the main idea from the global response as follows:
>
> **Original submission contributions:**
>
> - **Appendix B.4**: We compare 11 models across different model series (Qwen2.5/Qwen3/OctoThinker/Llama3.1), training stages (base/math base/SFT/instruct), and sizes (1.7B-8B), showing vastly different stability (Figures 28-31).
> - **Section 5.2**: Proposed that early dynamics serve as a model-task prior indicator, qualitatively showing differences between different backbones (Figure 6).
>
> **New validation (extending Section 5.2) in Global Response:**
>
> We quantitatively validate this indicator against gold-standard supervised RL. Across 7 models from 3 families, Model Collapse Step strongly predicts RL scalability. This demonstrates depth comparable to Zhang et al. (11 vs. 5 models) while providing predictive insights, not just observing dynamics, but quantifying which models will benefit from RL before expensive training.
>
> 4. **Identification of Safe** **Applications**
>
> A key difference from Zhang et al. is our discovery that **with small, domain-specific datasets, intrinsic rewards drive stable adaptation without collapse** (Section 4.3):
>
> - Training with 32-128 samples maintains stable performance despite saturated proxy reward (Figure 4).
> - Test-time training on target domains achieves sustained improvements on both in-distribution and out-of-distribution benchmarks (Figure 5).
> - This finding aligns with recent Nature paper AlphaProof[4], validating test-time training as a suitable application domain.
>
> This provides actionable guidance: intrinsic rewards are safe and effective when domain specificity is high and data is limited, rather than being uniformly problematic.
>
> 5. **Taxonomy of Distinct Modes**
>
> While Zhang et al. describe a uniform failure pattern (loss of reasoning), our analysis (Section 4.2, Figure 3) reveals that different methods result in **structurally distinct ways**, which is also what the reviewers agree with:
>
> - Self-Certainty & Majority Voting: Most stable, tapering smoothly
> - Probability: Collapses to brevity (multiplication bias favors short sequences)
> - Entropy-based methods: Collapse to repetitive patterns (exploiting low-entropy sequences)
>
> Understanding these distinct pathologies enables method selection and early intervention.
>
> 6. **Hyperparameter Analysis & Recipe**
>
> Furthermore, in our original submission, we go beyond just identifying optimal settings and provide a **systematic hyperparameter analysis in Appendix B.3**, which is grounded in our core theoretical framework. This provides a practical "recipe" for the community that, to our knowledge, is absent from prior work. For example:
>
> - **Off-Policy Degree (Mini-batch Size):** We empirically show that small mini-batches (e.g., size 1) cause near-immediate collapse (Figure 9). Our theory provides a principled reason: our optimal policy analysis (Sec. 3.2) assumes on-policy rewards, and small mini-batches introduce severe "reward staleness," which violates this core assumption and destabilizes training.
> - **Rollout size (N):** Counter-intuitively, increasing the number of rollouts (e.g., N=16, 32) accelerates model collapse (Figure 11). Our **Theorem 1** directly explains this: more rollouts create a more confident majority vote, which strengthens the "rich-get-richer" dynamic. This accelerates the geometric convergence, rapidly locking the model into its initial and potentially flawed prior.
>
> These findings that are explained by our theory provide detailed guidance for the community on how to stabilize and control intrinsic reward training.
>
> 7. **Pathways Beyond Intrinsic Rewards**
>
> Section 5.3 explores alternative directions: (1) leveraging unlabeled data, (2) exploiting generation-verification asymmetries. We further provide preliminary validation on countdown tasks. Please see our **[Global Response 2]** for details on how we go beyond RLIF analysis by proposing and empirically validating alternative unsupervised approaches in Section 5.3, which demonstrate the feasibility of other reward signals that avoid the fundamental limitations of intrinsic reward methods.
>
> In summary, while Zhang et al. provide valuable insights into entropy dynamics, our work offers **complementary theoretical foundations (unified framework, optimal policy analysis), empirical discoveries (safe applications, distinct failure modes), practical tools (model prior diagnostic, hyperparameter recipe) and future direction in unsupervised RL beyond intrinsic rewards** that advance understanding of when and how intrinsic rewards can be safely deployed.
>
> ---
>
> ### References
>
> [4] Olympiad-level formal mathematical reasoning with reinforcement learning

---

> ### Author Response · Authors · 2025-11-21
> **Response to Q1**
>
> > **Q1: Has the term URLVR been used before? My search came up empty and I find it confusing to have "Verified Reward" prominently in the name, but negate it by prefixing it with "unsupervised". Isn't Reinforcement Learning from Internal Feedback (RLIF) the established terminology at this point?**
>
> We sincerely thank the reviewer for this precise question on terminology. This is a crucial point, and we appreciate the chance to clarify our understanding. We agree that **RLIF (Reinforcement Learning from Internal Feedback)** is an established and important term, used in works like [1] and [3].
>
> However, we deliberately introduced **URLVR (Unsupervised Reinforcement Learning with Verifiable Rewards)** because "RLIF" does not cover the full scope of the problem we analyze:
>
> **Why "Unsupervised"?**
>
> - "Unsupervised RL" is a concept already in use by the community. For example, [1] refers to "unsupervised learning," while [2] and [3] both explicitly use the term "unsupervised RL."
> - As we define in line 38 of our paper, we use "unsupervised" to describe the training setting: one that derives rewards without ground truth labels.
> - This aligns with the standard definition of unsupervised learning (e.g., from Wikipedia) as algorithms that learn patterns exclusively from unlabeled data.
>
> **Why "Verifiable Rewards" (and not just "Unsupervised RL")?**
>
> This is the most critical distinction. We added "Verifiable Rewards" to precisely define the domain of tasks we are studying.
>
> - This is because our analysis in Sections 3 and 4 covers both certainty-based methods (which RLIF describes well) and ensemble-based methods (like TTRL).
> - Ensemble methods require an objectively verifiable task. For example, TTRL's majority voting must decode full responses and match them to find the most common answer. This "matching" step is only meaningful if there is a verifiable ground truth (e.g., a final numerical answer in a math problem).
> - If we used the broader term "Unsupervised RL," it would incorrectly group our work with a wide range of general-domain "self-rewarding" methods. Those methods are outside the scope of our theoretical and empirical analysis, which is focused on tasks with objective correctness.
>
> **How URLVR and RLIF Relate**
>
> We consider RLIF to be a subset of URLVR:
>
> - URLVR focuses on investigating how to improve performance on verifiable tasks without using human labels. While RLIF is one solution: Using internal signals (entropy, confidence) as a proxy reward. This covers most of Section 3 and 4 of our paper.
> - **But it's not the only solution.** As we argue in Section 5.3 (Beyond Intrinsic Rewards), a more scalable path forward may involve extrinsic but still unsupervised rewards (e.g., using a self-supervised objective or leveraging the generation-verification gap). These methods are "Unsupervised," but they are not "Internal Feedback." Our new experiment on the Countdown task (mentioned in the **[Global Response 2]**) is a primary example of this.
>
> Therefore, we chose URLVR as a more precise and encompassing term. It correctly frames the problem space (verifiable tasks, no labels) and is broad enough to include both the intrinsic/RLIF methods we analyze and the extrinsic, unsupervised methods. But thanks very much for your suggestion. We will add a paragraph to the introduction clarifying this distinction and our choice of terminology.
>
> ---
>
> ### References
>
> [1] Learning to Reason without External Rewards
>
> [2] The Unreasonable Effectiveness of Entropy Minimization in LLM Reasoning
>
> [3] No Free Lunch: Rethinking Internal Feedback for LLM Reasoning

---

> > ### Author Response · Authors · 2025-11-21
> > **Response to Q2**
> >
> > > **Q2: Regarding the prior (almost concurrent) work by Y. Zhang et al (No Free Lunch: Rethinking Internal Feedback for LLM Reasoning): Do you see any discrepancies between your results and conclusions and theirs?**
> >
> > That's an excellent question. We see our work as highly complementary to Zhang et al., as both papers confirm the critical importance of understanding the limitations of intrinsic rewards. While we arrive at similar high-level conclusions about instability, we do find some differences in perspective and emphasis, which we believe help build a more complete picture of the phenomenon.
> >
> > We've detailed these points in our response to your proposed Weakness 1, but as a brief summary, our main complementary findings are:
> >
> > - **Different Explanatory Frameworks:** Zhang et al. provide a valuable analysis through the lens of **"entropy minimization."** Our work offers a complementary perspective centered on **"confidence-correctness misalignment,"** which is formalized by our Theorem 1 (geometric convergence). This "rich-get-richer" dynamic explains why a model's initial, possibly flawed, prior gets amplified, leading to failure.
> > - **Divergent Empirical Findings:** These different frameworks naturally lead to different conclusions. For instance, an entropy-centric view might suggest high-entropy base models are the best candidates for this training. We found some specific cases (detailed in Appendix B.4, Figs 28-29) where our confidence-correctness framework seemed to offer a clearer explanation, such as a low-entropy distilled model (DeepSeek-R1-Distill) training stably (likely due to a good prior) while a high-entropy base model (Llama-3.1-Base) failed to improve.
> > - **Taxonomy of Failure Modes:** As the reviewer kindly noted, our work (Sec 4.2) also provides a specific taxonomy of how different methods fail (e.g., probability rewards collapsing to brevity vs. entropy rewards collapsing to repetition), which adds a different dimension to the general "loss of reasoning" failure mode.
> > - **A Different Final Conclusion on Safe Use:** Perhaps the most significant difference is in the final, actionable takeaway. Our work identifies that this failure mode is not inevitable. We conclude that these methods are **safe and highly effective** in specific, well-defined applications like **test-time training** (Sec 4.3), offering a more optimistic, practical path forward for their use.
> >
> > We've elaborated on these complementary contributions, along with our broader theoretical scope (covering ensemble methods) and our new diagnostic tools, in the detailed response to the first weakness you proposed. We believe that both papers viewed together provide a much more robust understanding of this complex topic.

---

### Author Response · Authors · 2025-11-21
**[Global Response 1 (Part 1)] Model-Task Priors: In-Depth Analysis Across Model Families**

Multiple reviewers raised questions about model selection and deeper analysis of which models benefit from intrinsic rewards:

- **Reviewer Kjs9 (Weakness #1)**: Notes that [1] provided "more in-depth analysis of learning dynamics for different base-models" and questions our depth of model analysis
- **Reviewer pWtu (Weakness #1)**: Questions whether we contribute new insights beyond analyzing existing methods
- **Reviewer um7o (Weakness #3)**: Suggests our sharpening insight is already "conventional wisdom"
- **Reviewer t95z (Weakness #1)**: Requests justification for Qwen3 choice and asks "whether there is something the reader can learn about how this choice is better than others that may be accessible to them"
- **Reviewer t95z (Question #1)**: Asks "Is there a way to show some language model families are more aligned than others in the sense defined in the paper?"

We address all these concerns with a unified response showing our in-depth, quantitative analysis of model-task priors that extends beyond conventional understanding.

**Building upon Our Original Analysis (Appendix B.4 + Section 5.2)**

In our original submission, we have provided substantial model analysis that we acknowledge could be highlighted more clearly in the main text:
- **In Appendix B.4**, we provided a broad observational study of training dynamics. Our appendix included a comparison of **11 models**, covering various **series** (Qwen2.5/Qwen3/OctoThinker/Llama3.1), various training **stages** (base/math base/sft/instruct), and various **sizes** (1.7B, 3B, 4B, 8B). This analysis observed that different models indeed have vastly different stability profiles (e.g., Fig. 28 to Fig. 31).
- **In Section 5.2**, we built on these observations to propose a novel application: the early-training behavior of intrinsic-reward–based URLVR methods is positively correlated with the strength of a model’s prior. For example, the results in Fig. 6 show that Qwen models maintain stable training while Llama models collapse by step 40, suggesting that early dynamics (within ~50 steps) can qualitatively diagnose RL scalability.

In response to reviewer feedback, we now extend Section 5.2 by providing a fully quantitative method for assessing base model prior. We introduce a new metric, **Model Collapse Step**, to quantify the model–task prior, shifting the analysis from qualitative observation to a predictive quantitative measure.

Compared with RLVR training or pass@k metrics, Model Collapse Step offers a more efficient way to detect **"which base model is better (i.e., more suitable for RL)?"**. It is well suited and useful for rapid iterative development of the foundation model, and we anticipate that it will attract more interest from industry researchers.

**New Experiment: Quantifying Model-Task Prior Strength**

We measured three indicators across **7 models from 3 families** (OLMo, Llama, Qwen), all trained on DAPO-17k and evaluated on AIME24 (avg@32):

1. **GT Gain (Ground Truth)**: Performance improvement from standard supervised RLVR with ground truth rewards, our gold-standard reference for true RL scalability.
2. **Pass@k**: Computed as the gap between pass@256 and pass@1.
3. **Model Collapse Step**: The training step when reward accuracy (majority voting pseudo-label correctness) drops below 1% during intrinsic URLVR training, our proposed diagnostic for model-task prior strength.

Results:

|                         | OLMo-2-1124-7B | Llama-3.1-8B | Qwen2.5-Math-1.5B | Qwen2.5-1.5B | Qwen2.5-7B | Qwen3-1.7B-Base | Qwen3-8B-Base |
| ----------------------- | -------------- | ------------ | ----------------- | ------------ | ---------- | --------------- | ------------- |
| **GT Gain**             | +0.42          | +1.01        | +3.96             | +3.96        | +6.67      | +7.08           | +17.08        |
| **Pass@k**              | +6.67          | +3.33        | +30.00            | +20.00       | +60.00     | +36.67          | +56.67        |
| **Model Collapse Step** | 34             | 40           | 160               | 221          | 245        | 280             | 383           |

---

### References

[1] No Free Lunch: Rethinking Internal Feedback for LLM Reasoning

---

> ### Author Response · Authors · 2025-11-21
> **[Global Response 1 (Part 2)]**
>
> **Key Findings and Implications**
>
> 1. The Model Collapse Step shows a strong correlation with GT Gain for predicting RL scalability:
>
>   - Llama-3.1-8B: Collapses at step 40 → minimal gain (+1.01%)
>   - Qwen3-8B-Base: Maintains reliable prior for 383 steps → substantial gain (+17.08%)
>
> 2. There is a clear hierarchy across model families, and we can now quantitatively show which families are more aligned:
>
>   - OLMo/LLaMA families: Weak priors (collapse steps 34-40), minimal RL gains (<2%)
>   - Qwen2.5 family: Moderate priors (collapse steps 160-245), moderate RL gains (4-7%)
>   - Qwen3 family: Strong priors (collapse steps 280-383), substantial RL gains (7-17%)
>
> 3. The Model Collapse Step offers unique advantages compared with pass@k:
>
>   - Accuracy: demonstrates a stronger correlation with GT Gain compared to pass@k.
>   - Generality: Works on subjective tasks without ground truth, while pass@k requires an objective answer to match
>
> This experiment extends our original Section 5.2 by validating the correlation between the model–task prior indicator and gold-standard supervised RL performance, providing a new method for identifying strong base models. We hope this quantitative metric can contribute to the development of foundation models and help alleviate the concerns raised by the reviewers.

---

### Author Response · Authors · 2025-11-21
**[Global Response 2] Self-Verify: Leveraging Generation-Verification Asymmetry for Scalable URLVR**

Multiple reviewers raised questions about our work's scope and future directions. Specifically:

- **Reviewer Kjs9 (Weakness #1)**: Asks about differences from [1] and whether we contribute beyond RLIF analysis
- **Reviewer pWtu (Weakness #1)**: Questions whether the paper is limited to analyzing existing methods

We address both concerns by highlighting a novel method beyond intrinsic rewards that we explore in our work: **self-verify**.

**Building upon Our Original Analysis (Section 5.3)**

While our main analysis (Sections 3-4) focuses on intrinsic reward methods, Section 5.3 explicitly identifies pathways to scale RL with URLVR methods at train-time, distinguishing our contribution from prior RLIF-focused work. We discuss two promising alternatives:

- Leveraging unlabeled data or external rewards

- Exploiting generation-verification asymmetries

For point 2, many problem domains exhibit difficulty asymmetries where verifying solutions is substantially easier than generating or answering them. For example, solving an equation or determining whether two matrices are inverses of each other. This enables models to self-reward through verification rather than confidence-based proxies, providing truly verifiable rewards aligned with task correctness without ground truth labels. This method is unsupervised and relies on internal feedback. More importantly, it is also a scalable RL approach at train time.

In our original submission, Section 5.3 proposed this direction. In response to reviewer feedback, we now provide empirical validation through a case study on the Countdown task.

**Case Study:** **Self-Verify** **on Countdown Task**

**Task Setup:**

- Countdown task: Given numbers, form an arithmetic expression reaching a target value
- Generation-verification asymmetry: Generation is challenging, but verification (checking if expression evaluates to target) is computationally trivial
- Self-verify approach: The model generates solutions and evaluates them on its own, obtaining ground-truth rewards in absence of human labels

**Experimental Setup:**

- Model: Qwen3-1.7B-Base
- Dataset: 4k training problems, 1k validation sampled from "Jiayi-Pan/Countdown-Tasks-3to4" on huggingface
- Comparisons:
  - **Oracle Supervision** (w/ ground truth labels as upper bound)
  - **Trajectory-Level Entropy** (intrinsic reward method baseline)
  - **Self-Verify** (our method using model's own verification)
- Evaluation: Validation accuracy (avg@16), Self-Verify Reward (proxy reward used in training) and Ground Truth Reward (oracle reward computed using actual correctness)

Countdown Validation (avg@16):

| step                      | 0     | 60     | 120    | 180    | 240    | 300    | 360    | 420    | 480    | 540    | 600    |
| ------------------------- | ----- | ------ | ------ | ------ | ------ | ------ | ------ | ------ | ------ | ------ | ------ |
| Oracle Supervision        | 9.57% | 62.88% | 77.55% | 80.16% | 82.77% | 83.45% | 83.52% | 84.71% | 86.08% | 86.38% | 85.06% |
| Trajectory-Level  Entropy | 9.57% | 8.06%  | 0.07%  | 0.06%  | 0.03%  | 0.00%  | 0.00%  | 0.00%  | 0.00%  | 0.00%  | 0.00%  |
| Self-Verify               | 9.57% | 10.11% | 14.82% | 12.69% | 28.23% | 44.17% | 49.85% | 52.07% | 52.20% | 55.83% | 58.85% |

Training Dynamics for Self-Verify (Self-Verify Reward and Ground Truth Reward)

| Step                | 0     | 60    | 120    | 180    | 240    | 300    | 360    | 420    | 480    | 540    | 600    |
| ------------------- | ----- | ----- | ------ | ------ | ------ | ------ | ------ | ------ | ------ | ------ | ------ |
| Self-Verify Reward  | 0.98% | 2.54% | 31.45% | 64.06% | 72.85% | 79.49% | 84.18% | 87.89% | 83.40% | 89.06% | 86.72% |
| Ground Truth Reward | 3.38% | 4.69% | 8.03%  | 12.99% | 19.43% | 37.56% | 44.12% | 51.13% | 39.16% | 53.16% | 57.17% |

**Key Results:**

Self-verify provides stronger supervision than intrinsic methods and approaches the effectiveness of oracle supervision:

- Qwen3-1.7B-Base achieves higher validation accuracy with self-verify compared to Trajectory-Level Entropy
- Performance even surpasses Oracle Supervision at later training steps

**Crucially, reward hacking does not occur.** Although Self-Verify Reward is optimized towards 1, the validation performance and Ground Truth Reward increase throughout training. This validates that self-verify supplies stronger, more reliable signals than intrinsic methods while avoiding the reward hacking problems of confidence-based approaches.

---

### References

[1] No Free Lunch: Rethinking Internal Feedback for LLM Reasoning

---

### Author Response · Authors · 2025-11-30
**Summary for Area Chair (Part 2)**

- **Reviewer um7o (Score: 2, Confidence: 4):** Reviewer um7o also did not have the opportunity to responding. We engaged in an active discussion with Reviewer um7o. We sincerely appreciate Reviewer um7o’s suggestions regarding the theoretical derivations. However, we would like to highlight a few points:
  - First, we hope the contributions of our work can be recognized in a balanced way across all dimensions. A score of 2 is clearly a complete rejection of the work. The theoretical analysis only takes about two pages in the main text, whereas our extensive and costly empirical study yields a large number of insights and provides strong validation for the proposed theoretical framework. **Reviewer Kjs9 explicitly highlighted and acknowledged this.** Moreover, we also proposed two practical methods and further refined during the discussion period. Reviewer um7o’s review is focused mainly on the theoretical formalism. We therefore hope that our work can be evaluated objectively and comprehensively, considering the theoretical insights, the systematic empirical experiments, and the practical methods together.
  - **Regarding the theoretical concerns, as mentioned earlier, the other reviewers have already reached a consensus on our theoretical contributions, and two reviewers even expressed strong appreciation.** We would also like to emphasize that a unified reward framework and theoretical analysis are in fact the foundation for subsequent empirical work and methodological exploration. Our theoretical investigation clearly identifies the right direction of empirical exploration and provides the right preliminary conclusions. The issues were mainly in presentation, which we have now thoroughly revised (highlighted in red) in the manuscript. We have addressed all originally raised concerns comprehensively, and also confirmed this in the Response to Follow-Up Review (Part 1) to Reviewer um7o.
  - In the second-round response, we further addressed Reviewer um7o’s remaining theoretical concerns. Among these, one stemmed from a complete misunderstanding, and one was newly introduced. It is also important to highlight that the remaining concerns center on theoretical formalism preferences in the appendix rather than empirical validity or core contributions of the paper. **After the Response to Follow-Up Review, all concerns have been fully and adequately clarified.**
  - **Since the reviewer could not continue the discussion afterward, we sincerely hope you can base your evaluation on our Response to Follow-Up Review for a comprehensive and objective assessment.** We are also very grateful for Reviewer um7o’s considerable effort, and we will fully incorporate the suggestions to improve the manuscript.

We would also like to clarify that during the Discussion phase, we also further refined the content in Sections 5.2 and 5.3 of the paper (Global Response 1 & 2).
In **Global Response 1**, we introduced an improved method for comparing the relative strengths of base models, a technique that is highly practical for industry teams developing foundation models. **Reviewer um7o also noted that this method is interesting.** In **Global Response 2**, we proposed a new method, **Self-Verify**, whose underlying philosophy is closely aligned with the recently highlighted **Self-Verification** approach in DeepSeek-Math-V2 [1] (the implementation details and experimental setup differ). Both methods leverage the idea of **generation–verification asymmetry** to scale compute on the verification side.

Notably, these two methods were explicitly recognized by reviewers as interesting or insightful. We want to emphasize that throughout the response process, we made substantial efforts both in addressing the reviewers’ questions and in improving the manuscript.

Our work makes contributions through **novel theoretical insights, systematic empirical analysis, and new practical tools**, and we hope you will consider your evaluation in light of all these aspects. We have also fully addressed Reviewer um7o’s concerns regarding the theoretical aspects. We will also do our utmost to thoroughly refine and strengthen the manuscript in response to all reviewers’ suggestions, as we believe has already been demonstrated in our earlier discussions.

Thank you sincerely for your careful consideration during this challenging time. If any questions arise, we would be more than happy to follow up at any moment.

Sincerely,

ICLR 2026 Conference Submission 23785 Authors

---

## References

[1] DeepSeekMath-V2: Towards Self-Verifiable Mathematical Reasoning

---

### Author Response · Authors · 2025-11-30
**Summary for Area Chair (Part 1)**

Dear Area Chair,

We sincerely appreciate your efforts during this challenging period. We understand the exceptional workload you're managing given the recent situations, and we're grateful for your careful consideration of our work.
To further assist you in gaining a deeper understanding of the manuscript, we would like to provide a summary of our recent discussions with the reviewers.

## Reviewer Assessment Overview
All reviewers recognized the timeliness of our work and the thoroughness of our empirical analysis. Three reviewers (pWtu, Kjs9, t95z) gave scores of 8, 6, and 6, all at or above the acceptance threshold. The main issue comes from Reviewer um7o’s score of 2, which stems from concerns about aspects of the theoretical formalism. These concerns have now been fully addressed in our latest response.

In fact, we found that **Reviewers pWtu and t95z** both clearly acknowledged the contribution of our theoretical derivations, and specifically highlighted the significance of the unified reward framework. **Reviewer Kjs9** is knowledgeable about the theoretical aspects and raised some concerns regarding certain details in comparison to related work, but did **not** suggest that there are limitations or issues with our theoretical derivations. **We believe that all the other three reviewers have already reached a clear consensus on the value of our theoretical contributions, and we will introduce this in detail below.** However, **Reviewer um7o** still has some concerns, particularly about some of the proofs in the appendix (Appendix A.3 and A.4).

## Discussion Details
Below, we will provide a description of the discussion with each reviewer. To objectively present the state of the discussion, we will quote the reviewers’ original comments from the discussion phase whenever possible:
- **Reviewer pWtu (Rating: 8, Confidence: 3).** We addressed each of Reviewer pWtu’s concerns individually. In the reviewer’s second-round response, the reviewer explicitly stated: "reccomended accepting the paper I will stick to that and not change my score." We greatly appreciate Reviewer pWtu’s recognition of our work.
  - Notably, in the Strengths, Reviewer pWtu clearly expressed appreciation for our **theoretical contributions**, particularly **the unified reward framework**, noting that "the paper presents a consolidated mathematical perspective that connect different intrinsic rewards methods under a single framework which is very nice."
  - The reviewer also highlighted our solid empirical evaluation and the insightful nature.
- **Reviewer t95z (Rating: 6, Confidence: 2):** explicitly states that our response resolved all concerns: "I believe that the responses provide answers to my questions raised during review."
  - Reviewer t95z also acknowledges our **theoretical contributions**. First, the reviewer highlighted our **unified reward framework**: "The paper unifies a variety of methods that use implicit reward modeling. This setup provides a framework for the community to understand the tradeoffs involved with these methods"
  - The reviewer further appreciates our derivation of **Theorem 1**: "The analytical approach used in Theorem 1 clarifies the confidence vs performance trade-off." They also recognize the strength of our **empirical work**: "Experimental validation and analysis is mostly sound and supports the observation made in Theorem 1."
  - In addition, Reviewer t95z mentions: "hold off revising the Rating until after the end of the discussion period."
- **Reviewer Kjs9 (Rating: 6, Confidence: 3):** Reviewer Kjs9 did not have a chance to respond. One point we would like to clarify is that Reviewer Kjs9 is quite familiar with the theoretical literature, and his concerns focus mainly on comparing our conclusions and contributions with those of No Free Lunch, which is also a theory-driven work. **Importantly, the reviewer did not raise issues regarding the form of our derivations or similar technical aspects.**
  - Reviewer Kjs9 expresses strong appreciation for our empirical contributions, noting that our work "...provides a systematic, empirical analysis of the distinct failure modes of different reward types...A practical strength of this paper is its investigation of scaling limits..." The reviewer was particularly positive about our new findings on stable training: "identifying test-time training as an ideal application."

---

### Meta-Review · Area_Chair_9vBY · 2026-01-08

**Summary:**

Major concerns:

1. Theoretical rigor and concerns about the necessity of the unified reward framework. (um7o)
2. Novelty: Prior work by Zhang et al. (2025) is very similar to this work (Kjs9); that majority voting sharpens the LM's distribution on correct outputs is already known. (um7o)
3. How is URLVR different from (the more established) RLIF? (Kjs9)
4. Reward in majority voting changes with policy. The closed-form solution in the paper may not apply to policy-dependent rewards. (um7o)

**Reviewer Concerns:**

Concerns 2, 3, and 4 are largely addressed with the new results (4) and the additional discussion (2, 3) presented during the review period. There was a long discussion regarding concern 1, which I done not believe is fully resolved.

**Reviewer Scores:**

Reviewers Kjs9 and t95z would have increased their scores. I believe um7o's score would have remained unchanged.

---

### Decision · Program_Chairs · 2026-01-26

Accept (Poster)